# On Linear Convergence in Smooth Convex-Concave Bilinearly-Coupled Saddle-Point Optimization: Lower Bounds and Optimal Algorithms

Ekaterina Borodich [1 2 3]  Alexander Gasnikov [4 3]  Dmitry Kovalev [2 3]

## Abstract

We revisit the smooth convex-concave bilinearly-coupled saddle-point problem of the form $\min_x \max_y f(x) + \langle y, \mathbf{B}x \rangle - g(y)$. In the highly specific case where function $f(x)$ is strongly convex and function $g(y)$ is affine, or both functions are affine, there exist lower bounds on the number of gradient evaluations and matrix-vector multiplications required to solve the problem, as well as matching optimal algorithms. A notable aspect of these algorithms is that they are able to attain linear convergence, i.e., the number of iterations required to solve the problem is proportional to $\log(1/\epsilon)$. However, the class of bilinearly-coupled saddle-point problems for which linear convergence is possible is much wider and can involve general smooth non-strongly convex functions $f(x)$ and $g(y)$. Therefore, *we develop the first lower complexity bounds and matching optimal linearly converging algorithms for this problem class*. Our lower complexity bounds are much more general, but they cover and unify the existing results in the literature. On the other hand, our algorithm implements the separation of complexities, which, for the first time, enables the simultaneous achievement of both optimal gradient evaluation and matrix-vector multiplication complexities, resulting in the best theoretical performance to date.

[1] Higher School of Economics [2] Yandex Research [3] Ivannikov Institute for System Programming, Trusted AI Research Center [4] Innopolis University. Correspondence to: Dmitry Kovalev <dakovalev1@gmail.com>, Ekaterina Borodich <borodich.ed@phystech.edu>, Alexander Gasnikov <gasnikov@yandex.ru>.

*Proceedings of the 42$^{nd}$ International Conference on Machine Learning*, Vancouver, Canada. PMLR 267, 2025. Copyright 2025 by the author(s).

## 1. Introduction

In this paper, we consider the following saddle-point optimization problem with a bilinear coupling function:

$$\min_{x \in \mathcal{X}} \max_{y \in \mathcal{Y}} \left[ F(x,y) = f(x) + \langle y, \mathbf{B}x \rangle - g(y) \right], \quad (1)$$

where $\mathcal{X} = \mathbb{R}^{d_x}$ and $\mathcal{Y} = \mathbb{R}^{d_y}$ are finite-dimensional Euclidean spaces, $\mathbf{B} \in \mathbb{R}^{d_y \times d_x}$ is a coupling matrix, and $f(x) \colon \mathcal{X} \to \mathbb{R}$ and $g(y) \colon \mathcal{Y} \to \mathbb{R}$ are continuous functions. We aim to solve problem (1) in the fundamental setting where both functions $f(x)$ and $g(y)$ are convex and smooth.[1]

Saddle-point problems of the form (1) appear in various fields such as economics (Von Neumann & Morgenstern, 1947), game theory (Roughgarden, 2010), and statistics (Berger, 2013). Moreover, these problems have a wide range of applications in machine learning, including supervised learning (Zhang & Xiao, 2017; Wang & Xiao, 2017; Xiao et al., 2019), reinforcement learning (Du et al., 2017), computer vision (Chambolle & Pock, 2011), robust optimization (Ben-Tal & Nemirovski, 2002; Liu et al., 2017), distributed optimization (Lan et al., 2020; Scaman et al., 2018; Kovalev et al., 2021; Yarmoshik et al., 2024; Kovalev et al., 2024), and the training of generative adversarial networks (Mescheder et al., 2017; Nagarajan & Kolter, 2017).

### 1.1. First-Order Methods and Linear Convergence

The majority of machine learning applications of problem (1) involve high-dimensional spaces $\mathcal{X}$ and $\mathcal{Y}$. In this scenario, the most widely used and often the only scalable optimization algorithms are *first-order methods*. These methods implement an iterative process to find an approximate solution to the problem using the evaluation of the gradients of the functions $f(x)$ and $g(y)$, as well as matrix-vector multiplication with the matrices $\mathbf{B}$ and $\mathbf{B}^\top$. More specifically, they perform iterative updates of the current estimate of the solution until it converges to the exact solution up to a given accuracy. One of the main goals of our paper is to develop efficient first-order optimization methods for

---

[1] A function is called smooth if it is differentiable and has a Lipschitz-continuous gradient. See Section 2 for an equivalent formal definition.

solving problem (1).

In this paper, we are interested in first-order methods for solving problem (1) that are able to achieve *linear convergence*. That is, we are interested in algorithms that can find an $\epsilon$-approximate solution to the problem using at most $\mathcal{O}(\mathcal{K} \cdot \log(1/\epsilon))$ gradient evaluations and matrix-vector multiplications, where $\mathcal{O}(\cdot)$ hides universal constants, $\epsilon > 0$ is an arbitrary precision, and $\mathcal{K} \geq 1$ is a constant that possibly depends on the internal properties of the problem such as condition numbers, etc.

In this work, we also intend to consider problem classes where linear convergence is possible in principle. A typical and one of the most fundamental examples of such a class is problem (1) with strongly convex functions $f(x)$ and $g(y)$. There are plenty of linearly converging first-order optimization methods in this *strongly-convex-strongly-concave* setting, which include the gradient descent ascent (Zhang et al., 2022a), the extragradient method (Korpelevich, 1976), and the optimistic gradient method (Gidel et al., 2018). Moreover, there is an array of algorithms that enjoy improved, or accelerated, convergence rates (Kovalev et al., 2022b; Thekumparampil et al., 2022; Jin et al., 2022; Du et al., 2022; Li et al., 2023) with the help of the Nesterov momentum trick (Nesterov, 2013). Another fundamental problem class where linear convergence is possible is the class of bilinear min-max games, which is a special case of problem (1) with affine functions $f(x)$ and $g(y)$. Such problems can be solved using the corresponding algorithms with linear convergence rates (Azizian et al., 2020; Li et al., 2022). Finally, Kovalev et al. (2022b) developed an algorithm for solving the general smooth convex-concave problem (1) and provided a set of sufficient conditions under which the proposed algorithm attains linear convergence.

### 1.2. Optimal Algorithms and Complexity Separation

In this paper, we are concerned with the task of developing *optimal algorithms*, which is one of the ultimate goals in optimization research. This task can be divided into two key parts. The first part involves finding lower bounds on the *oracle complexity* of solving the optimization problem, i.e., the number of oracle calls, such as gradient evaluations or matrix-vector multiplications, required to find an approximate solution to the problem. The second part is to find optimization algorithms that match these lower bounds. Such algorithms are called optimal because their oracle complexity cannot be improved due to the lower complexity bounds. For example, in the case of bilinear min-max games, lower bounds were proposed by Ibrahim et al. (2020) and matching optimal algorithms were developed by Azizian et al. (2020); Li et al. (2022). Similarly, Salim et al. (2022) developed lower bounds and optimal algorithms for smooth and strongly convex minimization with affine constraints,

which is a special case of problem (1) with a strongly convex function $f(x)$ and an affine function $g(y)$.

Unfortunately, apart from the cases of bilinear min-max games and affinely constrained minimization, the question of finding optimal algorithms for solving problem (1) is far from being resolved, even in the fundamental *strongly-convex-strongly-concave* setting. Although separate lower bounds on the gradient evaluation and matrix-vector multiplication complexities have already been developed by Nesterov (2013) and Zhang et al. (2022b), respectively, the existing state-of-the-art algorithms (Kovalev et al., 2022b; Thekumparampil et al., 2022; Jin et al., 2022; Du et al., 2022; Li et al., 2023) cannot simultaneously reach these bounds. The main reason is that these algorithms perform the same number of evaluations of the gradients $\nabla f(x)$ and $\nabla g(y)$ and matrix-vector multiplications with the matrices $\mathbf{B}$ and $\mathbf{B}^\top$ at each iteration while solving the problem, whereas the lower bounds on these numbers (Nesterov, 2013; Zhang et al., 2022b) can be significantly different. Thus, to reach the desired lower bounds, an optimal algorithm would have to implement *the separation of complexities* by skipping gradient evaluations and/or matrix-vector multiplications from time to time. Borodich et al. (2023); Alkousa et al. (2020); Sadiev et al. (2022); Lan & Ouyang (2021) attempted to develop efficient first-order methods with the complexity separation for solving the problem; however, these algorithms are not able to achieve optimal complexities by a substantial margin, see Section 5 and Table 1 for details.

The situation is even worse in other cases, such as the *strongly-convex-concave* or *convex-strongly-concave* settings, where only one of the functions $f(x)$ or $g(y)$ is strongly convex, or the *convex-concave setting*, where neither of the functions is strongly convex. To the best of our knowledge, there are no lower complexity bounds that would cover these cases, with the exception of the highly specific cases of bilinear min-max games (Ibrahim et al., 2020) and affinely constrained minimization (Salim et al., 2022). Therefore, the question remains unresolved as to whether the current state-of-the-art linearly converging algorithms for this setting (Kovalev et al., 2022b; Sadiev et al., 2022) are optimal or not.

### 1.3. Main Contributions

The above discussion reveals significant gaps in the current theoretical understanding of smooth convex-concave saddle-point problems with bilinear coupling. In particular, the existing lower complexity bounds are insufficient, and the state-of-the-art optimization algorithms are limited. Summarizing these gaps leads to the following open research question: *Is it possible to develop an optimal linearly converging first-order optimization method for solving the smooth convex-concave bilinearly-coupled saddle-point*

*problem* (1)*?* We provide a positive answer to this question and present the following key contributions:

**(i)** We describe the class of smooth convex-concave saddle-point problems of the form (1) for which it is possible to achieve linear convergence in Section 2. We establish *the first lower complexity bounds* for this class. In particular, we show that to find an $\epsilon$-approximate solution to problem (1), any first-order optimization method requires at least $\tilde{\mathcal{O}}(\kappa_x)$ evaluations of the gradient $\nabla f(x)$, $\tilde{\mathcal{O}}(\kappa_y)$ evaluations of the gradient $\nabla g(y)$, and $\tilde{\mathcal{O}}(\kappa_{xy})$ matrix-vector multiplications with the matrices $\mathbf{B}$ and $\mathbf{B}^\top$, where $\kappa_x$, $\kappa_y$, and $\kappa_{xy}$ denote certain condition numbers associated with functions $f(x)$, $g(y)$, and matrix $\mathbf{B}$.[2]

**(ii)** We show that our lower complexity bounds are tight. That is, we develop *the first optimal algorithm* that matches these lower bounds. This algorithm implements the complexity separation, allowing us to simultaneously achieve both optimal gradient evaluation and matrix-vector multiplication complexities. To the best of our knowledge, such a result has never been established in the literature, even in the fundamental strongly-convex-strongly-concave setting.

**(iii)** As a side contribution, we develop a new algorithm for solving a class of composite monotone variational inequalities. Just like the current state-of-the-art method of Lan & Ouyang (2021), our algorithm implements the separation of complexities, but enjoys substantially improved convergence rates and works in a much broader range of settings. Refer to Section 4 and Section 5.2 for details.

Our lower complexity bounds are much more general than the existing lower bounds for the special cases of strongly-convex-strongly-concave (Nesterov, 2013; Zhang et al., 2022b), bilinear (Ibrahim et al., 2020), and affinely constrained (Salim et al., 2022) optimization. On the other hand, our lower bounds recover and provide unification of these existing results. Besides, our optimal algorithm shows the best theoretical performance "on the market", which, to the best of our knowledge, outclasses all existing methods in the literature, with the exception of the algorithms of Azizian et al. (2020); Li et al. (2022) and Salim et al. (2022); Kovalev et al. (2020), which are already optimal in the aforementioned specific cases of bilinear and affinely constrained optimization, respectively.

---

[2]Here, $\tilde{\mathcal{O}}(\cdot)$ hides the logarithmic factor $\log(1/\epsilon)$, and universal (and possibly additive) constants. The precise definitions of $\kappa_x$, $\kappa_y$, and $\kappa_{xy}$ are provided in Section 2.

## 2. Preliminaries

### 2.1. Main Definitions and Assumptions

In this paper, we use mathematical notation, which is mostly standard and is therefore described in Appendix A. Further, in this section, we provide a formal description of the assumptions that we impose on problem (1). First, we define the (strong) convexity and smoothness properties of a differentiable function.

**Definition 2.1.** A differentiable function $h(x)\colon \mathbb{R}^d \to \mathbb{R}$ is called $\mu$-strongly convex for $\mu \geq 0$ if the following inequality holds for all $x, x' \in \mathbb{R}^d$:

$$\mathrm{D}_h(x, x') \geq \tfrac{1}{2} \cdot \mu \|x - x'\|^2. \tag{2}$$

A differentiable function $h(x)\colon \mathbb{R}^d \to \mathbb{R}$ is called convex if the same inequality holds with $\mu = 0$.

**Definition 2.2.** A differentiable function $h(x)\colon \mathbb{R}^d \to \mathbb{R}$ is called $L$-smooth for $L \geq 0$ if the following inequality holds for all $x, x' \in \mathbb{R}^d$:

$$|\mathrm{D}_h(x, x')| \leq \tfrac{1}{2} \cdot L \|x - x'\|^2. \tag{3}$$

Next, we formalize the (strong) convexity and smoothness assumptions that we impose on functions $f(x)$ and $g(y)$ as Assumptions 2.3 and 2.4. Note that we allow the strong convexity constants $\mu_x$ and $\mu_y$ to be zero, thus covering the case of non-strongly convex functions $f(x)$ and $g(y)$.

**Assumption 2.3.** Function $f(x)\colon \mathcal{X} \to \mathbb{R}$ is $\mu_x$-strongly convex and $L_x$-smooth for $L_x > \mu_x \geq 0$.

**Assumption 2.4.** Function $g(y)\colon \mathcal{Y} \to \mathbb{R}$ is $\mu_y$-strongly convex and $L_y$-smooth for $L_y > \mu_y \geq 0$.

Finally, the next Assumption 2.5 describes the spectral properties of the coupling matrix $\mathbf{B}$.

**Assumption 2.5.** There exist constants $L_{xy} > \mu_{xy}, \mu_{yx} \geq 0$, such that

$$\mu_{xy}^2 \leq \begin{cases} \lambda_{\min}^+(\mathbf{B}^\top \mathbf{B}) & \nabla f(x) \in \operatorname{range} \mathbf{B}^\top \text{ for all } x \in \mathcal{X} \\ \lambda_{\min}(\mathbf{B}^\top \mathbf{B}) & \text{otherwise} \end{cases}$$

$$\mu_{yx}^2 \leq \begin{cases} \lambda_{\min}^+(\mathbf{B}\mathbf{B}^\top) & \nabla g(y) \in \operatorname{range} \mathbf{B} \text{ for all } y \in \mathcal{Y} \\ \lambda_{\min}(\mathbf{B}\mathbf{B}^\top) & \text{otherwise} \end{cases}$$

$$L_{xy}^2 \geq \lambda_{\max}(\mathbf{B}^\top \mathbf{B}) = \lambda_{\max}(\mathbf{B}\mathbf{B}^\top),$$

Additionally, we assume that if $\mu_{xy} > 0$ and $\mu_{yx} > 0$, then $\mu_{xy} = \mu_{yx}$.

Further, to shorten the notation, we gather all the parameters defined in Assumptions 2.3 to 2.5 into a single vector $\pi = (L_x, L_y, L_{xy}, \mu_x, \mu_y, \mu_{xy}, \mu_{yx}) \in \Pi$, where $\Pi \subset \mathbb{R}_+^7$ denotes the parameter set.

## 2.2. Key Assumption for Linear Convergence

As discussed in Section 1, we are interested in algorithms for solving problem (1) that exhibit linear convergence. We introduce the key Assumption 2.6, which will enable us to establish linear lower complexity bounds and devise optimal linearly converging algorithms.

**Assumption 2.6.** Parameters $\pi \in \Pi$ satisfy the inequality $\min\{\delta_x, \delta_y\} > 0$, where $\delta_x$ and $\delta_y$ are defined as follows:

$$\delta_x = \mu_x + \mu_{xy}^2/L_y, \quad \delta_y = \mu_y + \mu_{yx}^2/L_x. \quad (4)$$

To better understand this assumption, consider the standard primal and dual reformulations of problem (1), which are given as follows:

$$\begin{aligned}
\min_{x \in \mathcal{X}} &\left[ P(x) = f(x) + g^*(\mathbf{B}x) \right], \\
\max_{y \in \mathcal{Y}} &\left[ D(y) = -g(y) - f^*(-\mathbf{B}^\top y) \right].
\end{aligned} \quad (5)$$

One can show that the primal objective function $P(x)$ and the dual objective function $-D(y)$ satisfy the *quadratic growth* condition (Anitescu, 2000; Karimi et al., 2016) with constants $\delta_x$ and $\delta_y$, respectively. This fact provides a good starting point for understanding why linear convergence is plausible under Assumption 2.6. On the other hand, Kovalev et al. (2022b) showed that this assumption is sufficient for developing a linearly converging algorithm. Moreover, in Section 3, we obtain Theorem 3.2, which implies that Assumption 2.6 is also necessary for achieving linear convergence, thus making it both a *necessary and sufficient* condition.

We also need to characterize the linear convergence rates of the first-order methods that we consider in this paper. Such rates are typically expressed via the *condition numbers* associated with a given optimization problem. Consequently, we define the following condition numbers for problem (1):

$$\kappa_x = \frac{L_x}{\delta_x}, \quad \kappa_y = \frac{L_y}{\delta_y}, \quad \kappa_{xy} = \frac{L_{xy}^2}{\delta_x \delta_y}. \quad (6)$$

The condition numbers $\kappa_x$ and $\kappa_y$ correspond to the functions $f(x)$ and $g(y)$, respectively. These can be seen as extensions of the standard condition numbers $L_x/\mu_x$ and $L_y/\mu_y$, which are commonly used in smooth and strongly convex optimization (Nesterov, 2013). Similarly, the condition number $\kappa_{xy}$ associated with the bilinear coupling term is a generalization of the standard condition number $L_{xy}^2/(\mu_x\mu_y)$, which is widespread in strongly-convex-strongly-concave saddle-point optimization (Zhang et al., 2022b; Ibrahim et al., 2020).

Further, we would like to ensure that the condition numbers defined in equation (6) are lower-bounded by some small universal constants. This is achieved by the following additional Assumption 2.7 on the parameter set $\Pi$. It allows us

to avoid addressing some corner cases where the condition numbers are small, which are neither theoretically nor practically interesting. It should be noted that Assumption 2.7 does not impose any fundamental restrictions;[3] it is merely introduced to streamline our complex theoretical findings.

**Assumption 2.7.** For all $\pi \in \Pi$ the following additional constraints are satisfied:

$$L_x > 4\mu_x, \; L_y > 4\mu_y, \; \sqrt{L_x L_y} > 4\max\{\mu_{xy}, \mu_{yx}\}$$
$$L_{xy} > 18\max\{\mu_{xy}, \mu_{yx}, \sqrt{\mu_x \mu_y}\}$$

## 2.3. Structure of the Solution Set

In this paper, we denote the solution set of the saddle-point problem (1) as $\mathcal{S} \subset \mathcal{X} \times \mathcal{Y}$. Under Assumptions 2.3 to 2.5, $(x^*, y^*) \in \mathcal{S}$ if and only if the following first-order optimality conditions hold:

$$\nabla f(x^*) + \mathbf{B}^\top y^* = 0, \quad \nabla g(y^*) - \mathbf{B}x^* = 0. \quad (7)$$

Moreover, under Assumption 2.6, the solution set is always non-empty and has an affine structure, as indicated by Lemma 2.8.

**Lemma 2.8.** *Under Assumptions 2.3 to 2.6, the solution set $\mathcal{S}$ of problem (1) is nonempty and is given as $\mathcal{S} = \mathcal{S}_x \times \mathcal{S}_y$, where $\mathcal{S}_x = \operatorname{Arg\,min}_{x \in \mathcal{X}} P(x)$ and $\mathcal{S}_y = \operatorname{Arg\,max}_{y \in \mathcal{Y}} D(y)$. Moreover, the primal and dual solution sets $\mathcal{S}_x \subset \mathcal{X}$ and $\mathcal{S}_y \subset \mathcal{Y}$ have the following affine structure:*

$$\begin{aligned}
\mathcal{S}_x &= x^* + \begin{cases} \{0\} & \mu_x > 0 \\ \ker \mathbf{B} & otherwise \end{cases}, \\
\mathcal{S}_y &= y^* + \begin{cases} \{0\} & \mu_y > 0 \\ \ker \mathbf{B}^\top & otherwise \end{cases},
\end{aligned} \quad (8)$$

*where $(x^*, y^*) \in \mathcal{S}$ is an arbitrary solution to problem (1).*

We also define a weighted squared distance function $\mathcal{R}_{\delta_x \delta_y}^2(x, y)$ as follows:

$$\mathcal{R}_{\delta_x \delta_y}^2(x, y) = \delta_x \operatorname{dist}^2(x; \mathcal{S}_x) + \delta_y \operatorname{dist}^2(y; \mathcal{S}_y). \quad (9)$$

We are going to use this function to measure the quality of a given approximate solution to problem (1) in both lower complexity bounds and the convergence analysis of optimal algorithms.

# 3. Lower Complexity Bounds

## 3.1. First-Order Saddle-Point Optimization Methods

In this section, we present lower bounds on the number of gradient evaluations and matrix-vector multiplications required to solve problem (1). These lower bounds apply to

---

[3]In particular, it is always possible to increase the smoothness constants $L_x$, $L_y$, and $L_{xy}$ to satisfy Assumption 2.7.

a specific class of algorithms that we refer to as *first-order saddle-point optimization methods*. A formal description of this class is provided in Definition 3.1. This definition is mostly inspired by the common *linear span assumption* (Nesterov, 2013; Zhang et al., 2022b; Ibrahim et al., 2020). However, the standard existing definitions focus only on iteration complexity. This is insufficient in our case, as we need to derive more specific lower bounds on the numbers of gradient evaluations and matrix-vector multiplications. Therefore, in Definition 3.1, we introduce a continuous execution time parameter $\tau \geq 0$ and assume that the evaluation of the gradients $\nabla f(x)$ and $\nabla g(y)$ takes time $\tau_f$ and $\tau_g$, respectively, while matrix-vector multiplication with matrices $\mathbf{B}$ and $\mathbf{B}^\top$ takes time $\tau_\mathbf{B}$. A similar approach was previously used in distributed optimization by Scaman et al. (2017; 2018); Kovalev et al. (2024), where they had to ensure a distinction between communication and local computation complexities.

**Definition 3.1.** An algorithm is called a first-order saddle-point optimization method with gradient computation times $\tau_f, \tau_g > 0$, and matrix-vector computation time $\tau_\mathbf{B} > 0$, if it satisfies the following constraints:

**(i) Memory.** At any time $\tau \geq 0$, the algorithm maintains a memory, which is represented by a set $\mathcal{M}(\tau) = \mathcal{M}_x(\tau) \times \mathcal{M}_y(\tau)$, where $\mathcal{M}_x(\tau) \subset \mathcal{X}$ and $\mathcal{M}_y(\tau) \subset \mathcal{Y}$. The memory can be updated by computing the gradients $\nabla f(x)$ and $\nabla g(y)$, and by performing matrix-vector multiplications with matrices $\mathbf{B}$ and $\mathbf{B}^\top$. This is represented by the following inclusions:

$$\begin{aligned} \mathcal{M}_x(\tau) &\subset \mathcal{M}_f(\tau) \cup \mathcal{M}_{\mathbf{B}^\top}(\tau), \\ \mathcal{M}_y(\tau) &\subset \mathcal{M}_g(\tau) \cup \mathcal{M}_\mathbf{B}(\tau), \end{aligned} \quad (10)$$

where sets $\mathcal{M}_f(\tau)$, $\mathcal{M}_g(\tau)$, $\mathcal{M}_{\mathbf{B}^\top}(\tau)$, and $\mathcal{M}_\mathbf{B}(\tau)$ are defined below.

**(ii) Gradient computation.** At any time $\tau \geq 0$, the algorithm can update the memory by computing the gradients $\nabla f(x)$ and $\nabla g(y)$, which take time $\tau_f$ and $\tau_g$, respectively. That is, for all $\tau \geq 0$, sets $\mathcal{M}_f(\tau) \subset \mathcal{X}$ and $\mathcal{M}_g(\tau) \subset \mathcal{Y}$ are defined as follows:

$$\mathcal{M}_f(\tau) = \begin{cases} \mathrm{span}(\{x, \nabla f(x) : x \in \mathcal{M}_x(\tau - \tau_f)\}) & \tau \geq \tau_f \\ \varnothing & \tau < \tau_f \end{cases},$$

$$\mathcal{M}_g(\tau) = \begin{cases} \mathrm{span}(\{y, \nabla g(y) : y \in \mathcal{M}_y(\tau - \tau_g)\}) & \tau \geq \tau_g \\ \varnothing & \tau < \tau_g \end{cases}.$$

**(iii) Matrix-vector multiplication.** At any time $\tau \geq 0$, the algorithm can update the memory by performing matrix-vector multiplication with matrices $\mathbf{B}$ and $\mathbf{B}^\top$, which takes time $\tau_\mathbf{B}$. That is, for all $\tau \geq 0$, sets

$\mathcal{M}_{\mathbf{B}^\top}(\tau) \subset \mathcal{X}$ and $\mathcal{M}_\mathbf{B}(\tau) \subset \mathcal{Y}$ are defined as follows:

$$\mathcal{M}_{\mathbf{B}^\top}(\tau) = \begin{cases} \mathrm{span}(\{x, \mathbf{B}^\top y : (x, y) \in \mathcal{M}(\tau - \tau_\mathbf{B})\}) & \tau \geq \tau_\mathbf{B} \\ \varnothing & \tau < \tau_\mathbf{B} \end{cases},$$

$$\mathcal{M}_\mathbf{B}(\tau) = \begin{cases} \mathrm{span}(\{\mathbf{B}x, y : (x, y) \in \mathcal{M}(\tau - \tau_\mathbf{B})\}) & \tau \geq \tau_\mathbf{B} \\ \varnothing & \tau < \tau_\mathbf{B} \end{cases}.$$

**(iv) Initialization and output.** At time $\tau = 0$, the algorithm must initialize the memory with the zero vector, that is, $\mathcal{M}_x(0) = \{0\}, \mathcal{M}_y(0) = \{0\}$. At any time $\tau \geq 0$, the algorithm must specify a single output vector from the memory, $(x_o(\tau), y_o(\tau)) \in \mathcal{M}(\tau)$.

### 3.2. Lower Bounds

In this section, we present our lower complexity bounds. We start with Theorem 3.2, which shows that it is not possible to obtain a linearly converging algorithm for solving problem (1) if Assumption 2.6 does not hold. The proof can be found in Appendix F. This theorem indicates that there exists a specific "hard" instance of problem (1), such that any first-order saddle-point optimization method fails to converge in terms of the distance to the solution set and converges sublinearly in terms of the primal-dual gap. It is important to clarify that the main purpose of Theorem 3.2 is to demonstrate the impossibility of attaining linear convergence in general if Assumption 2.6 does not hold, rather than to provide tight lower complexity bounds for this setting. Thus, we leave further investigation of the general case of problem (1) under Assumptions 2.3 to 2.5 for future work.

**Theorem 3.2.** *Let $\pi \in \Pi$, $R_x > 0$, and $\epsilon > 0$ be arbitrary parameters, distance, and precision, respectively. Suppose that Assumption 2.6 does not hold, i.e., without loss of generality, $\delta_x = 0$. There exists a problem (1) satisfying Assumptions 2.3 to 2.5 with parameters $\pi$, such that $\mathrm{dist}(0; \mathcal{S}_x) = R_x$, and for any first-order saddle-point optimization method and execution time $\tau > 0$, the following inequality holds:*

$$\mathrm{dist}^2(x_o(\tau); \mathcal{S}_x) > \tfrac{1}{8} R_x^2. \quad (11)$$

*Moreover, to reach precision on the primal-dual gap $P(x_o(\tau)) - D(y_o(\tau)) < \epsilon$ by any first-order saddle-point optimization method, the execution time $\tau$ must satisfy the following inequality:*

$$\tau \geq \Omega\left(\tau_f \cdot \mathrm{dist}(0; \mathcal{S}_x)\sqrt{L_x/\epsilon}\right). \quad (12)$$

Now, we are ready to present lower complexity bounds for problem (1) under Assumptions 2.3 to 2.7 in Theorem 3.3. The proof can be found in Appendix G. The lower bound on the total execution time $\tau$ in equation (13) contains the

terms $\tilde{\Omega}(\sqrt{\kappa_x})$, $\tilde{\Omega}(\sqrt{\kappa_y})$, and $\tilde{\Omega}(\sqrt{\kappa_{xy}})$. These terms can be respectively interpreted as gradient evaluation complexities with respect to the gradients $\nabla f(x)$ and $\nabla g(y)$, and matrix-vector multiplication complexity with respect to the matrices $\mathbf{B}$ and $\mathbf{B}^\top$, as they are respectively multiplied by the corresponding times $\tau_f$, $\tau_g$, and $\tau_{\mathbf{B}}$. In addition, these complexities are proportional to the logarithmic factor $\log(1/\epsilon)$, making them linear as we previously discussed.

**Theorem 3.3.** *Under Assumption 2.7, let $\pi \in \Pi$, $R > 0$, and $\epsilon > 0$ be arbitrary parameters, distance, and precision, respectively. Suppose that Assumption 2.6 holds. There exists a problem* (1) *satisfying Assumptions 2.3 to 2.5 with parameters $\pi$, such that $\mathcal{R}^2_{\delta_x \delta_y}(0,0) = R^2$, and to reach precision $\mathcal{R}^2_{\delta_x \delta_y}(x_o(\tau), y_o(\tau)) < \epsilon$ by any first-order saddle-point optimization method, the execution time $\tau$ must satisfy the following inequality:*

$$\tau \geq \tilde{\Omega}\left(\tau_f \cdot \sqrt{\kappa_x} + \tau_g \cdot \sqrt{\kappa_y} + \tau_{\mathbf{B}} \cdot \sqrt{\kappa_{xy}}\right), \quad (13)$$

*where $\tilde{\Omega}(\cdot)$ hides the multiplicative factor $\log \frac{cR^2}{\epsilon}$, $c > 0$ is a universal constant.*

The result in Theorem 3.3 has two important merits. First, this lower bound is tight, which we prove by developing a matching optimal algorithm in Section 4. Second, by making an appropriate restriction of the parameter set $\Pi$, we can recover the existing lower complexity bounds for the important and fundamental special cases of strongly-convex-strongly-concave saddle-point optimization (Zhang et al., 2022b; Nesterov, 2013), bilinear saddle-point optimization (Ibrahim et al., 2020), and strongly convex minimization with affine constraints (Salim et al., 2022). On the other hand, our result applies to an arbitrary choice of parameters $\pi \in \Pi$. Therefore, Theorem 3.3 and our definition of the condition numbers $\kappa_x$, $\kappa_y$, and $\kappa_{xy}$ in Section 2 provide unification and substantial generalization of the existing results. See Appendix D for additional discussion.

# 4. Optimal Algorithm

## 4.1. Monotone Variational Inequalities

In this section, we develop an optimal algorithm for solving problem (1). To do this, we consider a more general monotone variational inequality problem of finding $z^* \in \mathcal{C}_z$ such that

$$p(z^*) - p(z) + \langle Q(z), z^* - z \rangle \leq 0 \text{ for all } z \in \mathcal{C}_z, \ (14)$$

where $\mathcal{C}_z$ is a closed and convex subset of the finite-dimensional Euclidean space $\mathcal{Z} = \mathbb{R}^{d_z}$, and differentiable convex function $p(z) : \mathcal{Z} \to \mathbb{R}$ and continuous monotone operator $Q(z) : \mathcal{Z} \to \mathcal{Z}$ have the following finite-sum structures:

$$p(z) = \sum_{i=1}^n p_i(z), \qquad Q(z) = \sum_{i=1}^n Q_i(z), \quad (15)$$

where $p_i(z) : \mathcal{Z} \to \mathbb{R}$, $Q_i(z) : \mathcal{Z} \to \mathcal{Z}$. Vector $z^*$ defined in equation (14) is often called a *weak* solution to the monotone variational inequality. In the setting of this paper, it is equivalent to the *strong* solution[4]; refer to Kinderlehrer & Stampacchia (2000) for details.

Further, we assume that the gradients $\nabla p_i(z)$ and operators $Q_i(z)$ are monotone and Lipschitz with respect to the norm $\|\cdot\|_{\mathbf{P}}^2$, where $\mathbf{P} \in \mathbb{S}_{++}^{d_z}$. These assumptions are commonly used in the literature and are formalized through the following Definitions 4.1 and 4.2 and Assumptions 4.3 and 4.4. Note that Assumption 4.3 implies that each function $p_i(z)$ is convex and smooth.

**Definition 4.1.** An operator $G(x) : \mathbb{R}^d \to \mathbb{R}^d$ is called $\mu$-strongly monotone with respect to the norm $\|\cdot\|_{\mathbf{P}}$ for $\mu \geq 0$ if the following inequality holds for all $x, x' \in \mathbb{R}^d$:

$$\langle G(x) - G(x'), x - x' \rangle \geq \mu \|x - x'\|_{\mathbf{P}}^2. \quad (16)$$

An operator $G(x) : \mathbb{R}^d \to \mathbb{R}^d$ is called monotone if the same inequality holds with $\mu = 0$.

**Definition 4.2.** An operator $G(x) : \mathbb{R}^d \to \mathbb{R}^d$ is called $M$-Lipschitz with respect to the norm $\|\cdot\|_{\mathbf{P}}$ for $M \geq 0$ if the following inequality holds for all $x, x' \in \mathbb{R}^d$:

$$\|G(x) - G(x')\|_{\mathbf{P}^{-1}} \leq M \|x - x'\|_{\mathbf{P}}. \quad (17)$$

**Assumption 4.3.** For all $1 \leq i \leq n$, the gradient $\nabla p_i(z)$ is monotone and $L_i$-Lipschitz w.r.t. $\|\cdot\|_{\mathbf{P}}$.

**Assumption 4.4.** For all $1 \leq i \leq n$, operator $Q_i(z)$ is monotone and $M_i$-Lipschitz w.r.t. $\|\cdot\|_{\mathbf{P}}$.

## 4.2. Optimal Sliding Algorithm for Monotone Variational Inequalities

Now, we are ready to present our new algorithm for solving the variational inequality problem (14). One of the key ideas behind the development of this algorithm is our new perspective on the celebrated accelerated gradient method of Nesterov (2013). In particular, a single step of this algorithm, applied to minimizing an $L$-smooth convex function $h(z) : \mathcal{Z} \to \mathbb{R}$, can be seen as applying a single step of the standard gradient method to the function $h_t(z)$ with the fixed stepsize $1/L$, where $h_t(z) : \mathcal{Z} \to \mathbb{R}$ is defined as follows:

$$h_t(z) = \alpha_t^{-2} h(\alpha_t z + (1 - \alpha_t)\bar{z}^t), \quad \text{where} \quad \bar{z}^t \in \mathcal{Z}. \ (18)$$

Indeed, the stepsize $1/L$ is suitable, since one can show that function $h_t(z)$ is $L$-smooth as well. Hence, using the standard recursion for the gradient descent, for all $z \in \mathcal{Z}$, we obtain the following inequality:[5]

$$\tfrac{1}{2}L\|z^{t+1} - z\|^2 + h_t(z^{t+1}) \leq \tfrac{1}{2}L\|z^t - z\|^2 + h_t(z). \ (19)$$

---

[4]Vector $z^* \in \mathcal{C}_z$ is a strong solution to the variational inequality if $p(z^*) - p(z) + \langle Q(z^*), z^* - z \rangle \leq 0$ for all $z \in \mathcal{C}_z$.

[5]Refer to Lan (2020, proof of Theorem 3.3) for the proof.

Next, we can define $\overline{z}^{t+1} = \alpha_t z^{t+1} + (1-\alpha_t)\overline{z}^t$, and use the definition of function $h_t(z)$ in equation (18), the convexity of function $h(z)$, and the definition of $\alpha_t$ in equation (20).[6] This gives the following recursion:

$$\frac{1}{2}L\|z^{t+1} - z\|^2 + \alpha_t^{-2}[h(\overline{z}^{t+1}) - h(z)]$$
$$\leq \frac{1}{2}L\|z^t - z\|^2 + \alpha_{t-1}^{-2}[h(\overline{z}^t) - h(z)],$$

which implies the accelerated convergence rate $[h(\overline{z}^t) - \min_z h(z)] = \mathcal{O}(LR^2/t^2)$, where $R > 0$ is the initial distance to the solution. Overall, the derivations above offer a vast simplification compared to the standard proof of Nesterov (2013).

Inspired by the *sliding* algorithm of Lan & Ouyang (2021; 2016); Kovalev et al. (2022a), we apply a series of transformations of the form (18) to functions $p_i(z)$ in a recursive fashion. This leads, subject to some additional details, to Algorithm 1 for solving problem (14).[7] Moreover, using the considerations above, we obtain the key theoretical result in Theorem 4.5. The proof can be found in Appendix K.

**Theorem 4.5.** *Let Assumptions 4.3 and 4.4 hold, where $M_i, L_i \geq 0$ and $M_i + L_i > 0$. Let $\alpha_t$ be defined recursively as follows:*

$$\alpha_0 = 1, \ \alpha_{t+1} = 2 \cdot \left(1 + \sqrt{1 + 4/\alpha_t^2}\right)^{-1} \text{ for } t \geq 1. \quad (20)$$

*Then the output $z_{out}$ of Algorithm 1 for solving problem (14) satisfies the inclusion $z_{out} \in \mathcal{C}_z$ and the inequality*

$$p(z_{out}) - p(z) + \langle Q(z), z_{out} - z \rangle$$
$$\leq \sum_{i=1}^n \left( \frac{4^i L_i}{\prod_{j=1}^i T_j^2} + \frac{2^i M_i}{\prod_{j=1}^i T_j} \right) \cdot \frac{1}{2}\|z_{in} - z\|_{\mathbf{P}}^2. \quad (21)$$

Furthermore, we can reorder functions $p_i(z)$ and operators $Q_i(z)$ in ascending order of the values of the Lipschitz constants $L_i$ and $M_i$, which leads to the complexity result in Corollary 4.6. The proof can be found in Appendix L.

**Corollary 4.6.** *Under the conditions of Theorem 4.5, to ensure the following inequality*

$$p(z_{out}) - p(z) + \langle Q(z), z_{out} - z \rangle \leq \epsilon n \|z_{in} - z\|_{\mathbf{P}}^2 \quad (22)$$

*for all $z \in \mathcal{C}_z$ and $\epsilon > 0$, it is sufficient to perform no more than*

$$6^n \cdot \max\left\{\sqrt{L_i/\epsilon}, M_i/\epsilon, 1\right\} \quad (23)$$

*computations of the gradient $\nabla p_i(z)$ and operator $Q_i(z)$ for $1 \leq i \leq n$.*

Using Corollary 4.6, we can show that Algorithm 1 achieves the optimal complexity separation for solving the variational inequality problem (14) as long as $n = \mathcal{O}(1)$. Thus, Algorithm 1 matches the existing lower complexity bounds for this problem (Nesterov, 2013; Ouyang & Xu, 2021) and theoretically outperforms the existing state-of-the-art algorithm of Lan & Ouyang (2021), which is designed for the case $n = 2$ with additional restrictions.[8] See Section 5.2 for details.

### 4.3. Application to the Main Saddle-Point Problem

In this section, we show how to adapt Algorithm 1 to solve the main problem (1) and reach the lower complexity bounds in Theorem 3.3. To do this, we define a diagonal matrix $\mathbf{P} \in \mathbb{S}_{++}^{d_z}$ as follows:

$$\mathbf{P} = \mathrm{diag}(\delta_x \mathbf{I}_{d_x}, \delta_y \mathbf{I}_{d_y}), \quad (24)$$

and consider a special instance of problem (14), where $n = 3$, $\mathcal{Z} = \mathcal{C}_z = \mathcal{X} \times \mathcal{Y}$, operators $Q_i(z) = Q_i(x, y)$ are defined as follows:

$$Q_1(x,y) = 0, \qquad Q_3(x,y) = \begin{bmatrix} \mathbf{O}_{d_x} & \mathbf{B}^\top \\ -\mathbf{B} & \mathbf{O}_{d_y} \end{bmatrix} \begin{bmatrix} x \\ y \end{bmatrix}, \quad (25)$$
$$Q_2(x,y) = 0,$$

and functions $p_i(z) = p_i(x, y)$ are defined as follows:

$$p_1(x,y) = f(x), \qquad p_2(x,y) = g(y),$$
$$p_3(x,y) = \frac{\beta_x}{2}\|\mathbf{B}x - \nabla g(y_{in})\|^2 + \frac{\beta_y}{2}\|\mathbf{B}^\top y + \nabla f(x_{in})\|^2, \quad (26)$$

where $z_{in} = (x_{in}, y_{in}) \in \mathcal{Z}$ and $\beta_x, \beta_y \geq 0$ are defined as

$$\beta_x = 1/(4L_y), \qquad \beta_y = 1/(4L_x). \quad (27)$$

From the optimality conditions (7), one can conclude that this variational inequality problem instance is equivalent to the original problem (1) as long as function $p_3(z)$ is replaced with zero. However, the additional quadratic regularizer $p_3(z)$ helps to achieve the optimal linear convergence rates in all cases where $\delta_x > 0$ and $\delta_y > 0$, even when $\mu_x = 0$ and/or $\mu_y = 0$. Moreover, this regularization does not break the convergence to the exact solution $z^*$ of the original problem (1). Indeed, it is easy to show that $\nabla p_3(z^*)$ converges to zero as long as $z_{in}$ converges to $z^*$.[9]

Next, we apply Algorithm 1 to solve the variational inequality problem instance defined in equations (14), (25) and (26). Using Theorem 4.5 and Corollary 4.6, we obtain the following result in Theorem 4.7. The proof can be found in Appendix M.

---

[6]From the definition of $\alpha_t$ in equation (20), it follows that $\alpha_t \in (0, 1]$ and $\alpha_t^{-2} = \alpha_{t-1}^{-2} + \alpha_t^{-1}$.

[7]The pseudocode for the proposed Algorithm 1 is postponed to Appendix B due to space limitations.

[8]The algorithm of Lan & Ouyang (2021) works in the case $n = 2$, where $Q_1(z) \equiv \mathbf{0}_{d_z}$ and $p_2(z) \equiv 0$.

[9]Refer to the proof of Theorem 4.7 in Appendix K for more details.

**Theorem 4.7.** *Under Assumptions 2.3 to 2.7, functions $p_i(z)$, operators $Q_i(z)$, and matrix $\mathbf{P}$ defined in equations* (24), (25) *and* (26) *satisfy the conditions of Theorem 4.5 with the following parameters:*

$$L_1 = \kappa_x, \quad L_2 = \kappa_y, \quad L_3 = \kappa_{xy},$$
$$M_1 = M_2 = 0, \quad M_3 = \sqrt{\kappa_{xy}}. \tag{28}$$

*Moreover, the input $z_{in} = (x_{in}, y_{in})$ and the output $z_{out} = (x_{out}, y_{out})$ of Algorithm 1 satisfy the inequality $\Psi(z_{out}) \leq \frac{2}{3}\Psi(z_{in})$ as long as the numbers of inner iterations $\{T_i\}_{i=1}^3$ are chosen according to the proof of Corollary 4.6 in Appendix L. Here, the Lyapunov function $\Psi(z) = \Psi(x, y)$ is defined as follows:*

$$\Psi(z) = \mathcal{R}^2_{\delta_x \delta_y}(x, y) + 12\mathrm{D}_f(x, x^*) + 12\mathrm{D}_g(y, y^*), \tag{29}$$

*where $z^* = (x^*, y^*) = \mathrm{proj}_{\mathcal{S}}(z_{in}) = \mathrm{proj}_{\mathcal{S}}(z_{out})$.*

Theorem 4.7 implies that we can reduce the value of the Lyapunov function $\Psi(x, y)$ defined in equation (29) by a constant factor with a single run of Algorithm 1. Hence, we can apply the standard restarting technique to this algorithm and obtain the complexity result in Corollary 4.8. The proof can be found in Appendix N.

**Corollary 4.8.** *Under the conditions of Theorem 4.7, to reach precision $\mathcal{R}^2_{\delta_x \delta_y}(x, y) \leq \epsilon$, it is sufficient to perform $\tilde{\mathcal{O}}(\sqrt{\kappa_x})$, $\tilde{\mathcal{O}}(\sqrt{\kappa_y})$, and $\tilde{\mathcal{O}}(\sqrt{\kappa_{xy}})$ computations of the gradients $\nabla f(x)$ and $\nabla g(y)$, and matrix-vector multiplications with the matrices $\mathbf{B}$ and $\mathbf{B}^\top$, respectively. Here, $\tilde{\mathcal{O}}(\cdot)$ hides the multiplicative factor $\log \frac{cR^2}{\epsilon}$, $R^2 = \mathcal{R}^2_{\delta_x \delta_y}(0, 0)$ is the initial distance, $\epsilon \in (0, R^2)$, and $c = 1 + 12\kappa_x + 12\kappa_y$.*

The complexity result in Corollary 4.8 matches the lower complexity bounds in Theorem 3.3 up to universal and/or additive constants. Hence, this result is optimal. Moreover, to the best of our knowledge, this result theoretically outperforms all existing state-of-the-art algorithms, including the algorithms of Kovalev et al. (2022b); Li et al. (2023); Jin et al. (2022); Du et al. (2022); Thekumparampil et al. (2022); Borodich et al. (2023); Alkousa et al. (2020); Sadiev et al. (2022); Chambolle & Pock (2011). See Section 5.1 for additional discussion.

# 5. Comparison with Existing Results

## 5.1. Algorithms for Solving the Main Problem (1)

The theoretical complexity of Algorithm 1 with restarting, applied to solve the smooth bilinearly-coupled saddle-point optimization problem (1), is established in Corollary 4.8 and is proven to be optimal due to the lower complexity bounds in Theorem 3.3. We compare this result with the theoretical complexities of the existing state-of-the-art linearly converging first-order methods. These include the

algorithms for the strongly-convex-strongly-concave case (Kovalev et al., 2022b; Li et al., 2023; Jin et al., 2022; Du et al., 2022; Thekumparampil et al., 2022; Borodich et al., 2023; Chambolle & Pock, 2011; Alkousa et al., 2020), the strongly-convex-concave case (Kovalev et al., 2022b; Sadiev et al., 2022), and the convex-concave case (Kovalev et al., 2022b). This comparison is summarized in Table 1, which is postponed to Appendix C due to the page limit. One can observe that our optimal result is substantially better compared to all the listed algorithms.

Once again, it is worth highlighting that the complexity of our algorithm matches the complexities of the algorithms of Salim et al. (2022) and Azizian et al. (2020); Li et al. (2022), which are optimal in the more specialized cases of affinely constrained minimization and bilinear saddle-point optimization, respectively. We provide a detailed discussion of the lower complexity bounds in these cases in Appendix D.

## 5.2. Algorithm for Solving the Variational Inequality Problem (14)

Here, we consider the variational inequality problem defined in equations (14) and (15) in the case $n = 2$, where function $p_2(z)$ and operator $Q_1(z)$ are zero. This is one of the simplest yet most important special cases of problem (14) as discussed by Lan & Ouyang (2021). We compare Algorithm 1 for solving this problem with the algorithm of Lan & Ouyang (2021). Let $R \geq 0$ be the following distance parameter associated with the constraint set $\mathcal{C}_z$:

$$R = \sup_{z \in \mathcal{C}_z} \|z - z_{\text{in}}\|_{\mathbf{P}}. \tag{30}$$

We compare the numbers of evaluations of the gradient $\nabla p_1(z)$ and operator $Q_2(z)$ required by both algorithms to find a vector $z_{\text{out}} \in \mathcal{C}_z$ that satisfies the following accuracy criterion:

$$\sup_{z \in \mathcal{C}_z} p(z_{\text{out}}) - p(z) + \langle Q(z), z_{\text{out}} - z \rangle \leq \epsilon, \tag{31}$$

where $\epsilon > 0$ is an arbitrary precision. Note that the parameter $R$ is finite only if the constraint set is bounded. However, we can easily tackle this issue by following the standard approach and replacing the constraint set $\mathcal{C}_z$ with its intersection with the ball $\{z \in \mathcal{Z} : \|z - z_{\text{in}}\|_{\mathbf{P}} \leq D\}$, where $D > 0$ is a positive parameter. Refer, for instance, to Nesterov (2007).

The comparison of Algorithm 1 with the algorithm of Lan & Ouyang (2021) is summarized in Table 2, along with the corresponding lower complexity bounds (Nesterov, 2013; Ouyang & Xu, 2021). The table is postponed to Appendix C due to the page limit. One can observe that the theoretical complexities of these algorithms coincide up to universal

constants when $\sqrt{L_1R^2/\epsilon} \leq M_2R^2/\epsilon$. However, Algorithm 1 can significantly outperform the algorithm of Lan & Ouyang (2021) in the case where $\sqrt{L_1R^2/\epsilon} \gg M_2R^2/\epsilon$. It is important to highlight that this case is worth considering, as it plays an essential role in achieving the optimal complexities in Corollary 4.8 for the main problem (1). In addition, the algorithm of Lan & Ouyang (2021) only works in the case $n = 2$ with the additional restrictions described above. On the other hand, using the result in Corollary 4.6, it is easy to verify that our Algorithm 1 can achieve the optimal complexity separation for $n > 2$, provided that $n = \mathcal{O}(1)$.

## Impact Statement

This paper presents work whose goal is to advance the field of Machine Learning. There are many potential societal consequences of our work, none which we feel must be specifically highlighted here.

## Acknowledgements

This work was supported by a grant, provided by the Ministry of Economic Development of the RF in accordance with the subsidy agreement (agreement identifier 000000C313925P4G0002) and the agreement with the Ivannikov Institute for System Programming of the RAS dated June 20, 2025 No. 139-15-2025-011.

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

# Appendix

## A. Notation

In this paper, we use the following notations: $\mathbb{S}^p$ and $\mathbb{S}^p_{++}$ denote the sets of $p \times p$ symmetric and symmetric positive definite matrices, respectively; $\mathbf{I}_p$ denotes the $p \times p$ identity matrix, $\mathbf{J}_{p \times q}$ and $\mathbf{O}_{p \times q}$ denote the $p \times q$ all-ones and all-zeros matrices, respectively, $\mathbf{J}_p = \mathbf{J}_{p \times p}$ and $\mathbf{O}_p = \mathbf{O}_{p \times p}$; $\mathbf{e}_j^p \in \mathbb{R}^p$ denotes the $j$-th unit basis vector, $\mathbf{1}_p = (1, \ldots, 1) \in \mathbb{R}^p$, $\mathbf{0}_p = (0, \ldots, 0) \in \mathbb{R}^p$. In addition, $\|\cdot\|$ denotes the standard Euclidean norm of a vector, and $\langle \cdot, \cdot \rangle$ denotes the standard scalar product of two vectors, $\|\cdot\|_{\mathbf{P}} = \|\mathbf{P}^{\frac{1}{2}}(\cdot)\|$ and $\langle \cdot, \cdot \rangle_{\mathbf{P}} = \langle \mathbf{P}(\cdot), \cdot \rangle$ denote the weighted Euclidean norm and scalar product, respectively, where $\mathbf{P} \in \mathbb{S}^p_{++}$; $\lambda_{\min}(\cdot)$, $\lambda_{\min}^+(\cdot)$, and $\lambda_{\max}(\cdot)$ denote the smallest, smallest positive, and largest eigenvalues of a symmetric matrix, respectively; $\sigma_{\max}(\cdot)$ and $\sigma_{\min}^+(\cdot)$ denote the largest and smallest positive singular values of a matrix.

For a nonempty closed convex set $\mathcal{A} \subset \mathbb{R}^d$ and a vector $x \in \mathbb{R}^d$, we define the standard distance function $\mathrm{dist}(x; \mathcal{A})$ as follows:

$$\mathrm{dist}(x; \mathcal{A}) = \min_{x' \in \mathcal{A}} \|x - x'\|. \tag{32}$$

For a differentiable function $h(x)$, we denote the Bregman divergence associated with $h(x)$ as $\mathrm{D}_h(x, x')$, which is defined as follows:

$$\mathrm{D}_h(x, x') = h(x) - h(x') - \langle \nabla h(x'), x - x' \rangle. \tag{33}$$

For a proper, closed, and convex function $h(x)$, we denote its Fenchel conjugate as $h^*(x)$, its Moreau envelope as $M_{\lambda h}(x)$, and its proximal operator as $\mathrm{prox}_{\lambda h}(x)$. These are respectively defined as follows:

$$h^*(x) = \sup_{x'} \left( \langle x, x' \rangle - h(x') \right),$$

$$M_{\lambda h}(x) = \min_{x'} \left( h(x') + \frac{1}{2\lambda} \|x' - x\|^2 \right), \tag{34}$$

$$\mathrm{prox}_{\lambda h}(x) = \arg\min_{x'} \left( h(x') + \frac{1}{2\lambda} \|x' - x\|^2 \right).$$

## B. Algorithm 1 Pseudocode

---

**Algorithm 1**

---

1: **input:** $z_{\text{in}} \in \mathcal{C}_z$
2: **parameters:** $\mathbf{P} \in \mathbb{S}_{++}^{d_z}$, $\{\alpha_t\}_{t=0}^{\infty} \subset \mathbb{R}_{++}$,
 $\{L_i\}_{i=1}^{n}, \{M_i\}_{i=1}^{n} \subset \mathbb{R}_+$, $\{T_k\}_{k=1}^{n} \in \{1, 2, \ldots\}$
3: **for** $i = 1, \ldots, n$ **do**
4: $\quad z_{0,\ldots,0}^i = z_{\text{in}}$
5: $\quad p_i^0(z) \equiv p_i(z)$
6: **end for**
7: $z_{\text{out}} = \text{RecursiveProcedure}(1)$
8: **output:** $z_{\text{out}} \in \mathcal{Z}$
9: $\triangleright$ Auxiliary Recursive Procedure:
10: **function** RecursiveProcedure$(k, t_1, \ldots, t_{k-1})$
11: $\quad$ **if** $k = n + 1$ **then**
12: $\quad\quad$ **return** $\arg\min_{z \in \mathcal{C}_z} \sum_{i=1}^{n} p_i^{n;t_1,\ldots,t_n}(z)$
13: $\quad$ **else**
14: $\quad\quad \bar{z}_0^k = z_{t_1,\ldots,t_{k-1},0}^k$
15: $\quad\quad$ **for** $t_k = 0, \ldots, T_k - 1$ **do**
16: $\quad\quad\quad$ **for** $i = 1, \ldots, n$ **do**
17: $\quad\quad\quad\quad \hat{p}_i^{k;t_1,\ldots,t_k}(z) \equiv \begin{cases} \alpha_{t_k}^{-1} p_i^{k-1;t_1,\ldots,t_{k-1}}(\alpha_{t_k} z + (1 - \alpha_{t_k})\bar{z}_{t_k}^k) & i \geq k \\ p_i^{k-1;t_1,\ldots,t_{k-1}}(z) & i < k \end{cases}$
18: $\quad\quad\quad\quad L_i^{k;t_1,\ldots,t_k} = L_i \cdot \prod_{l=1}^{k} \alpha_{t_l}$
19: $\quad\quad\quad\quad M_i^{k;t_1,\ldots,t_k} = M_i \cdot \prod_{l=1}^{k} (\alpha_{t_l}/\alpha_{T_l - 1})$
20: $\quad\quad\quad\quad H_i^{k;t_1,\ldots,t_k} = L_i^{k;t_1,\ldots,t_k} + M_i^{k;t_1,\ldots,t_k}$
21: $\quad\quad\quad\quad \Delta_i^{k;t_1,\ldots,t_k} = \nabla \hat{p}_i^{k;t_1,\ldots,t_k}(z_{t_1,\ldots,t_k}^k) + Q_i(z_{t_1,\ldots,t_k}^k)$
22: $\quad\quad\quad\quad p_i^{k;t_1,\ldots,t_k}(z) \equiv \begin{cases} \frac{H_k^{k;t_1,\ldots,t_k}}{2} \|z - z_{t_1,\ldots,t_k}^k\|_{\mathbf{P}}^2 + \langle z, \Delta_k^{k;t_1,\ldots,t_k} \rangle & i = k \\ \hat{p}_i^{k;t_1,\ldots,t_k}(z) & i \neq k \end{cases}$
23: $\quad\quad\quad$ **end for**
24: $\quad\quad\quad z_{t_1,\ldots,t_{k-1},t_k+1/2}^k = \text{RecursiveProcedure}(k+1, t_1, \ldots, t_k)$
25: $\quad\quad\quad \bar{z}_{t_k+1}^k = \alpha_{t_k} z_{t_1,\ldots,t_{k-1},t_k+1/2}^k + (1 - \alpha_{t_k})\bar{z}_{t_k}^k$
26: $\quad\quad\quad \Delta_{Q_k}^{k;t_1,\ldots,t_k} = Q_k(z_{t_1,\ldots,t_k}^k) - Q_k(z_{t_1,\ldots,t_{k-1},t_k+1/2}^k)$
27: $\quad\quad\quad z_{t_1,\ldots,t_{k-1},t_k+1}^k = z_{t_1,\ldots,t_{k-1},t_k+1/2}^k + (H_k^{k;t_1,\ldots,t_k}\mathbf{P})^{-1} \Delta_{Q_k}^{k;t_1,\ldots,t_k}$
28: $\quad\quad\quad$ **for** $l = k + 1, \ldots, n$ **do**
29: $\quad\quad\quad\quad z_{t_1,\ldots,t_{k-1},t_k+1,0,\ldots,0}^l = z_{t_1,\ldots,t_{k-1},t_k,T_{k+1},0,\ldots,0}^l$
30: $\quad\quad\quad$ **end for**
31: $\quad\quad$ **end for**
32: $\quad\quad$ **return** $\bar{z}_{T_k}^k$
33: $\quad$ **end if**
34: **end function**

---

# C. Tables

*Table 1.* Comparison of the optimal complexity of Algorithm 1 for solving the main problem (1) developed in this paper (Theorem 3.3, Corollary 4.8) with the existing state-of-the-art linearly-converging algorithms in the strongly-convex-strongly-concave, strongly-convex-concave, and convex-concave settings.

| Algorithm | Complexity[1] | | |
|---|---|---|---|
| | $\nabla f(x)$ | $\nabla g(y)$ | $\mathbf{B}$ and $\mathbf{B}^\top$ |
| **Strongly-convex-strongly-concave case ($\mu_x, \mu_y > 0$ and $\mu_{xy} = \mu_{yx} = 0$)** | | | |
| Kovalev et al. (2022b); Li et al. (2023) Jin et al. (2022); Du et al. (2022) Thekumparampil et al. (2022) | $\sqrt{\frac{L_x}{\mu_x}} + \sqrt{\frac{L_y}{\mu_y}} + \frac{L_{xy}}{\sqrt{\mu_x\mu_y}}$ | | |
| Borodich et al. (2023) | $\sqrt{\frac{L_x}{\mu_x}} + \sqrt{\frac{L_y}{\mu_y}}$ | | $\sqrt{\frac{L_x}{\mu_x}} + \sqrt{\frac{L_y}{\mu_y}} + \frac{L_{xy}}{\sqrt{\mu_x\mu_y}}$ |
| Chambolle & Pock (2011) | N/A[2] | | $\frac{L_{xy}}{\sqrt{\mu_x\mu_y}}$ |
| Alkousa et al. (2020) | $\sqrt{\frac{L_x}{\mu_x}}$ | $\frac{L_{xy}\sqrt{L_y}}{\sqrt{\mu_x}\mu_y}$ | $\frac{\sqrt{L_{xy}^3}}{\sqrt{\mu_x}\mu_y}$ |
| **Optimal[3] (this paper)[4]** | $\sqrt{\frac{L_x}{\mu_x}}$ | $\sqrt{\frac{L_y}{\mu_y}}$ | $\frac{L_{xy}}{\sqrt{\mu_x\mu_y}}$ |
| **Strongly-convex-concave case ($\mu_x, \mu_{yx} > 0$ and $\mu_{xy} = \mu_y = 0$)[5]** | | | |
| Kovalev et al. (2022b) | $\frac{L_{xy}}{\mu_{yx}}\sqrt{\frac{L_x}{\mu_x}} + \frac{\sqrt{L_xL_y}}{\mu_{yx}} + \frac{L_{xy}^2}{\mu_{yx}^2}$ | | |
| Sadiev et al. (2022) | $\frac{L_{xy}}{\mu_{yx}}\sqrt[4]{\frac{L_x^3}{\mu_x^3}} + \frac{L_{xy}^2}{\mu_{yx}^2}\sqrt[4]{\frac{L_x}{\mu_x}}$ | N/A[2] | $\frac{L_{xy}}{\mu_{yx}}\sqrt{\frac{L_x}{\mu_x}} + \frac{L_{xy}^2}{\mu_{yx}^2}$ |
| **Optimal[3] (this paper)** | $\sqrt{\frac{L_x}{\mu_x}}$ | $\frac{\sqrt{L_xL_y}}{\mu_{yx}}$ | $\frac{L_{xy}}{\mu_{yx}}\sqrt{\frac{L_x}{\mu_x}}$ |
| **Convex-concave case ($\mu_{xy} = \mu_{yx} > 0$ and $\mu_x = \mu_y = 0$)** | | | |
| Kovalev et al. (2022b) | $\frac{L_{xy}\sqrt{L_xL_y}}{\mu_{xy}^2} + \frac{L_{xy}^2}{\mu_{xy}^2}$ | | |
| **Optimal[3] (this paper)** | $\frac{\sqrt{L_xL_y}}{\mu_{xy}}$ | $\frac{\sqrt{L_xL_y}}{\mu_{xy}}$ | $\frac{L_{xy}\sqrt{L_xL_y}}{\mu_{xy}^2}$ |

[1] For brevity, we omit universal constants and logarithmic factors such as $\log\frac{1}{\epsilon}$.

[2] Requires computation of the proximal operators of functions $f(x)$ and/or $g(y)$.

[3] Lower complexity bounds are established in Theorem 3.3. These bounds are matched by Algorithm 1, which is established by Theorem 4.7 and Corollary 4.8.

[4] Here, the lower bounds were also established by Zhang et al. (2022b); Nesterov (2013).

[5] This case is symmetric to the convex-strongly-concave case, which we omit for brevity.

*Table 2.* Comparison of Algorithm 1 with the algorithm of Lan & Ouyang (2021) for solving the variational inequality problem defined in equations (14) and (15) in the case $n = 2$, $p_2(z) \equiv 0$, $Q_1(z) \equiv 0$.

| Algorithm | Complexity | |
|---|---|---|
| | $\nabla p_1(z)$ | $Q_2(z)$ |
| Lan & Ouyang (2021) | $\mathcal{O}\left(\sqrt{\frac{L_1 R^2}{\epsilon}}\right)$ | $\mathcal{O}\left(\sqrt{\frac{L_1 R^2}{\epsilon}} + \frac{M_2 R^2}{\epsilon}\right)$ |
| **Algorithm 1 (this paper)** Lower Bounds (Nesterov, 2013; Ouyang & Xu, 2021) | $\mathcal{O}\left(\sqrt{\frac{L_1 R^2}{\epsilon}}\right)$ | $\mathcal{O}\left(\frac{M_2 R^2}{\epsilon}\right)$ |

## D. Additional Discussion of the Lower Complexity Bounds

As mentioned in Section 3.2, the lower complexity bound in Theorem 3.3 recovers several important problem classes, which are special instances of problem (1). We can recover these problem classes by imposing additional constraints on the parameter set $\Pi$.

(i) The class of smooth strongly-convex-strongly-concave saddle-point optimization problems corresponds to the constraint

$$\mu_{xy} = \mu_{yx} = 0. \tag{35}$$

In this case, the lower bound in equation (13) becomes the following:

$$\tilde{\Omega}\left(\tau_f \cdot \sqrt{\frac{L_x}{\mu_x}} + \tau_g \cdot \sqrt{\frac{L_y}{\mu_y}} + \tau_{\mathbf{B}} \cdot \frac{L_{xy}}{\sqrt{\mu_x \mu_y}}\right), \tag{36}$$

where $\tilde{\Omega}(\cdot)$ hides universal constants and logarithmic factors. This result recovers the existing lower complexity bounds of Zhang et al. (2022b); Nesterov (2013).

(ii) The class of bilinear saddle-point optimization problems is obtained by choosing

$$L_x = L_y = 0, \quad \mu_x = \mu_y = 0, \quad \tau_f = \tau_g = 0, \quad \mu_{xy} = \mu_{yx} > 0, \tag{37}$$

and the lower bound in equation (13) turns into the following:

$$\tilde{\Omega}\left(\tau_{\mathbf{B}} \cdot \frac{L_{xy}}{\mu_{xy}}\right). \tag{38}$$

This result recovers the existing lower complexity bound of Ibrahim et al. (2020). Note that strictly speaking, we cannot choose $L_x = L_y = 0$ due to Assumption 2.7. However, this is not an issue, because Assumption 2.7 allows us to choose arbitrary $L_x, L_y > 0$ such that $\sqrt{L_x L_y} = 5\mu_{xy}$ and still obtain the lower bound (38) from equation (13). In addition, as mentioned in Section 2, this assumption is not a fundamental restriction but is rather used to avoid covering uninteresting corner cases in our theoretical proofs.

(iii) The class of smooth strongly convex optimization problems with affine constraints is obtained by choosing

$$L_y = \mu_y = 0, \quad \mu_{xy} = 0, \quad \tau_g = 0. \tag{39}$$

In this case, the lower bound in equation (13) becomes the following:

$$\tilde{\Omega}\left(\tau_f \cdot \sqrt{\frac{L_x}{\mu_x}} + \tau_{\mathbf{B}} \cdot \frac{L_{xy}}{\mu_{yx}}\sqrt{\frac{L_x}{\mu_x}}\right), \tag{40}$$

which recovers the existing result of Salim et al. (2022). Similarly to the previous case (ii), we can choose $L_y = 17\mu_{yx}^2/L_x$ instead of $L_y = 0$ to satisfy Assumption 2.7 and still obtain the lower bound (40).

It is worth mentioning the work of Ouyang & Xu (2021), who offer sublinear lower complexity bounds for this problem class. However, their result does not contradict ours since they consider the case $\mu_{yx} = \delta_y = 0$, i.e., Assumption 2.6 does not hold. It is also important to highlight that affinely constrained optimization problems, where $\mu_{yx} = 0$, hold limited interest. Indeed, in this setting, it is typically assumed that $\mu_{yx}^2 = \lambda_{\min}^+(\mathbf{B}\mathbf{B}^\top)$, which, by definition, is always nonzero.

# E. Proof of Lemma 2.8

We define linear spaces $\mathcal{L}_x \subset \mathcal{X}$ and $\mathcal{L}_y \subset \mathcal{Y}$ as follows:

$$\mathcal{L}_x = \begin{cases} \{0\} & \mu_x > 0 \\ \ker \mathbf{B} & \text{otherwise} \end{cases}, \qquad \mathcal{L}_y = \begin{cases} \{0\} & \mu_y > 0 \\ \ker \mathbf{B}^\top & \text{otherwise} \end{cases}. \tag{41}$$

One can show that the following identity holds:

$$F(x + d_x, y + d_y) = F(x, y) \quad \text{for all} \quad (x, y) \in \mathcal{X} \times \mathcal{Y} \quad \text{and} \quad (d_x, d_y) \in \mathcal{L}_x \times \mathcal{L}_y. \tag{42}$$

Indeed, for the saddle part, we obviously have $\langle y + d_y, \mathbf{B}(x + d_x) \rangle = \langle y, \mathbf{B}x \rangle$. Furthermore, we can show that $f(x + d_x) = f(x)$ and $g(y + d_y) = g(y)$. Indeed, Assumptions 2.5 and 2.6 and equation (41) imply $\nabla f(x) \in \mathcal{L}_x^\perp$ and $\nabla g(y) \in \mathcal{L}_y^\perp$. Hence, we obtain

$$f(x + d_x) - f(x) = \int_0^1 \langle \nabla f(x + d_x \cdot t), d_x \rangle \mathrm{d}t = 0,$$

$$g(y + d_y) - g(y) = \int_0^1 \langle \nabla g(y + d_y \cdot t), d_y \rangle \mathrm{d}t = 0,$$

which conludes the proof of equation (42). In addition, it is easy to show that

$$\mathrm{dom}\, f^*(\cdot) \subset \mathcal{L}_x^\perp \quad \text{and} \quad \mathrm{dom}\, g^*(\cdot) \subset \mathcal{L}_y^\perp. \tag{43}$$

Consider the following saddle-point problem:

$$\min_{x \in \mathcal{L}_x^\perp} \max_{y \in \mathcal{L}_y^\perp} F(x, y). \tag{44}$$

We can show that this problem has a unique solution $(x^*, y^*) \in \mathcal{L}_x^\perp \times \mathcal{L}_y^\perp$, which, together with equation (42), implies Lemma 2.8. Let us further prove this statement.

One can show that function $P(x)$ is strongly convex on $\mathcal{L}_x^\perp$. Indeed, if $\mu_x > 0$ this statement is obvious. Otherwise, Assumption 2.6 implies $\mu_{xy} > 0$, which in turn implies the strong convexity of function $g^*(\mathbf{B}x)$ on $\mathcal{L}_x^\perp$ thanks to the strong convexity of function $g^*(y)$. The strong convexity of function $g^*(y)$ is implied by the smoothness of function $g(y)$.

Next, we show that $\mathrm{dom}\, P(\cdot) \neq \varnothing$, which immediately implies $\mathrm{dom}\, P(\cdot) \cap \mathcal{L}_x^\perp \neq \varnothing$, thanks to equation (42). Indeed, if $\mu_y > 0$ function $g^*(y)$ is smooth, which implies $\mathrm{dom}\, P(\cdot) = \mathcal{X}$. Otherwise, Assumption 2.6 implies $\mu_{yx} > 0$, which in turn implies $\nabla g(y) \in \mathrm{range}\, \mathbf{B}$ for all $y \in \mathcal{Y}$. Hence, there exists $x \in \mathcal{L}_x^\perp$ such that $\mathbf{B}x = \nabla g(y)$ for some $y \in \mathcal{Y}$. On the other hand, $\nabla g(y) \in \mathrm{dom}\, g^*(\cdot)$, which implies $\mathbf{B}x \in \mathrm{dom}\, g^*(\cdot)$ and $x \in \mathrm{dom}\, P(\cdot)$.

The strong convexity of $P(x)$ on $\mathcal{L}_x^\perp$ and the fact that $\mathrm{dom}\, P(\cdot) \cap \mathcal{L}_x^\perp \neq \varnothing$ imply that there exists a unique solution $x^* \in \mathcal{L}_x^\perp$ to the following problem:

$$\min_{x \in \mathcal{L}_x^\perp} P(x). \tag{45}$$

Similarly, there exists a unique solution $y^* \in \mathcal{L}_y^\perp$ to the following problem:

$$\max_{y \in \mathcal{L}_y^\perp} D(y). \tag{46}$$

Moreover, vectors $x^*$ and $y^*$ are solutions to the primal and dual problems in equation (5), respectively, thanks to equation (42).

Let $h(x) = g^*(\mathbf{B}x)$. Vector $x^*$ is a solution to the problem $\min_{x \in \mathcal{X}}[f(x) + h(x)]$. Hence, standard theory implies $-\nabla f(x^*) \in \partial h(x^*)$, or

$$h(x) \geq h(x^*) - \langle \nabla f(x^*), x - x^* \rangle \quad \text{for all} \quad x \in \mathcal{X}. \tag{47}$$

From this inequality, for arbitrary $x \in x^* + \ker \mathbf{B}$, we obtain

$$0 \leq \langle \nabla f(x^*), x - x^* \rangle,$$

which implies $\nabla f(x^*) \in \text{range } \mathbf{B}^\top$. Hence, there exists $y \in \mathcal{L}_y^\perp$ such that $\nabla f(x^*) = -\mathbf{B}^\top y$, which for all $x \in \mathcal{X}$, implies

$$h(x) \geq h(x^*) + \langle y, \mathbf{B}x - \mathbf{B}x^* \rangle.$$

Hence, for all $z \in \text{range } \mathbf{B}$, we obtain

$$g^*(z) \geq g^*(\mathbf{B}x^*) + \langle y, z - \mathbf{B}x^* \rangle.$$

In addition, this inequality holds for all $z \in \mathcal{Y}$ due to equation (43). Hence, $y \in \partial g^*(\mathbf{B}x^*)$, which implies $\mathbf{B}x^* = \nabla g(y)$. We also have $x^* \in \partial f^*(-\mathbf{B}^\top y)$. Hence, $y$ is a solution of problem (46), which implies $y = y^*$. It remains to observe, that $(x^*, y^*)$ satisfies the first-order optimality conditions (7). Hence, the strong duality holds in both problems (1) and (44), which concludes the proof. □

## F. Proof of Theorem 3.2

Note that the condition $\delta_x = 0$ implies $\mu_x = 0$ and $\mu_{xy} = 0$. Consider the following special instance of problem (1):

$$\min_{u_x \in \mathbb{R}^{d_1}} \min_{v_x \in \mathbb{R}^{d_2}} \max_{y \in \mathbb{R}^{d_2}} f(u_x) + \frac{L_x}{2}\|v_x\|^2 + \mu_{yx}\langle y, v_x \rangle - \frac{\mu_y}{2}\|y\|^2, \tag{48}$$

where $f(x) \colon \mathbb{R}^{d_1} \to \mathbb{R}$ is the $L_x$-smooth function proposed by Nesterov (2013, Theorem 2.1.7). This problem has a single solution $(u_x^*, 0, 0) \in \mathbb{R}^{d_1} \times \mathbb{R}^{d_2} \times \mathbb{R}^{d_2}$, where $u_x^* = \arg\min_{u_x \in \mathbb{R}^{d_1}} f(u_x)$. Moreover, the primal-dual gap is lower-bounded as follows:

$$P(u_x, v_x) - D(y) \geq f(u_x) - f(u_x^*).$$

Thus, the statement of Theorem 3.2 trivially follows from Theorem 2.1.7 of Nesterov (2013). $\qquad\square$

## G. Proof of Theorem 3.3

The proof of Theorem 3.3 relies on the following Lemmas G.1, G.2 and G.3. The proof of Lemma G.1 is available in appendix H. The proof of Lemma G.2 is available in appendix I. The proof of Lemma G.3 is available in appendix J.

**Lemma G.1.** *Under conditions of Theorem 3.3, the execution time can be lower-bounded as follows:*

$$\tau \geq \Omega\left(\tau_f \cdot \sqrt{\kappa_x}\log\frac{cR^2}{\epsilon} + \tau_g \cdot \sqrt{\kappa_y}\log\frac{cR^2}{\epsilon}\right). \tag{49}$$

**Lemma G.2.** *Under conditions of Theorem 3.3, let $\mu_y, \mu_{yx} > 0$ and $\mu_{yx}^2 \geq \mu_x\mu_y$. Then the execution time can be lower-bounded as follows:*

$$\tau \geq \Omega\left(\tau_{\mathbf{B}} \cdot \sqrt{\kappa_{xy}}\log\frac{cR^2}{\epsilon}\right). \tag{50}$$

**Lemma G.3.** *Under conditions of Theorem 3.3, let $\mu_x, \mu_y > 0$ and $\mu_x\mu_y \geq \max\{\mu_{xy}^2, \mu_{yx}^2\}$. Then the execution time can be lower-bounded as follows:*

$$\tau \geq \Omega\left(\tau_{\mathbf{B}} \cdot \sqrt{\kappa_{xy}}\log\frac{cR^2}{\epsilon}\right). \tag{51}$$

It remains to obtain the lower bound $\Omega\left(\tau_{\mathbf{B}} \cdot \sqrt{\kappa_{xy}}\log\frac{cR^2}{\epsilon}\right)$ without the additional assumptions that were made in Lemmas G.2 and G.3. It can be done by considering the following special cases:

(i) **Case $\mu_x = \mu_y = 0$.** In this case, we have $\mu_{xy} = \mu_{yx} > 0$ due to Assumptions 2.5 and 2.6. We can replace $\mu_y = 0$ with a very small value $\mu_y > 0$ and apply Lemma G.2 to obtain the desired result.

(ii) **Case $\mu_x = 0$ and $\mu_y > 0$.**

    (ii.a) **Case $\mu_{yx} > 0$.** We can apply Lemma G.2.

    (ii.b) **Case $\mu_{yx} = 0$.** This case is symmetric to case **(iii.b)**.

(iii) **Case $\mu_x > 0$ and $\mu_y = 0$.**

    (iii.a) **Case $\mu_{xy} > 0$.** This case is symmetric to case **(ii.a)**.

    (iii.b) **Case $\mu_{xy} = 0$.** In this case, we have $\mu_{yx} > 0$ due to Assumption 2.6. We can replace $\mu_y = 0$ with a very small value $\mu_y > 0$ and apply Lemma G.2 to obtain the desired result.

(iv) **Case $\mu_x > 0$ and $\mu_y > 0$.**

    (iv.a) **Case $\mu_x\mu_y \geq \max\{\mu_{xy}^2, \mu_{yx}^2\}$.** We can apply Lemma G.3.

    (iv.b) **Case $\mu_x\mu_y < \mu_{yx}^2$.** We can apply Lemma G.2.

    (iv.c) **Case $\mu_x\mu_y < \mu_{xy}^2$.** This case is symmetric to case **(iv.b)**.

This concludes the proof. $\qquad\square$

# H. Proof of Lemma G.1

**Case $\mu_{xy} > 0$.** We consider a special instance of problem (1), where $\mathcal{X} = \mathcal{Y} = \mathbb{R}^d$, functions $f(x)$ and $g(y)$ and matrix $\mathbf{B}$ are defined as follows:

$$f(x) = \frac{\mu_x}{2}\|x\|^2 + \frac{L_x - \mu_x}{2}\|\mathbf{F}x\|^2 - A\langle \mathbf{e}_1^d, x\rangle,$$

$$g(y) = \frac{L_y}{2}\|y\|^2, \qquad \mathbf{B} = \mu_{xy}\mathbf{I}_d, \tag{52}$$

where matrix $\mathbf{F} \in \mathbb{R}^{(d-1)\times d}$ is defined as follows:

$$\mathbf{F} = \frac{1}{2}\begin{bmatrix} 1 & -1 & & \\ & \ddots & \ddots & \\ & & \ddots & \ddots \\ & & & 1 & -1 \end{bmatrix}. \tag{53}$$

This problem instance has a unique solution $(x^*, y^*) \in \mathcal{X} \times \mathcal{Y}$, which is given as follows:

$$x^* = \arg\min_{x \in \mathcal{X}} \frac{\delta_x}{2}\|x\|^2 + \frac{L_x - \mu_x}{2}\|\mathbf{F}x\|^2 - A\langle \mathbf{e}_1^d, x\rangle, \qquad y^* = \frac{\mu_{xy}}{L_y}x^*. \tag{54}$$

The rest of the proof is similar to the proofs of Lemmas G.2 and G.3.

**Case $\mu_{xy} = 0$.** In this case, we assume $\mu_{yx} > 0$, otherwise we can use the proof of the previous case. We consider a special instance of problem (1), where $\mathcal{X} = \mathbb{R}^{d+1}$ and $\mathcal{Y} = \mathbb{R}$, functions $f(x)$ and $g(y)$ and matrix $\mathbf{B}$ are defined as follows:

$$f(x) = f(u_x, v_x) = \frac{\mu_x}{2}\|x\|^2 + \frac{L_x - \mu_x}{2}\|\mathbf{F}u_x\|^2 - A\langle \mathbf{e}_1^d, u_x\rangle,$$

$$g(y) = \frac{L_y}{2}\|y\|^2, \qquad \mathbf{B} = \mu_{yx}\begin{bmatrix} 0 \cdots\cdots\cdots 0 & 1 \end{bmatrix}, \tag{55}$$

where $x = (u_x, v_x)$, $u_x \in \mathbb{R}^d$, $v_x \in \mathbb{R}$, and matrix $\mathbf{F} \in \mathbb{R}^{(d-1)\times d}$ is defined as follows:

$$\mathbf{F} = \frac{1}{2}\begin{bmatrix} 1 & -1 & & \\ & \ddots & \ddots & \\ & & \ddots & \ddots \\ & & & 1 & -1 \end{bmatrix}. \tag{56}$$

This problem instance has a unique solution $(x^*, y^*) = (u_x^*, v_x^*, y^*) \in \mathcal{X} \times \mathcal{Y}$, which is given as follows:

$$u_x^* = \arg\min_{x \in \mathcal{X}} \frac{\mu_x}{2}\|u_x\|^2 + \frac{L_x - \mu_x}{2}\|\mathbf{F}u_x\|^2 - A\langle \mathbf{e}_1^d, u_x\rangle, \qquad v_x^* = y^* = 0. \tag{57}$$

The rest of the proof is similar to the proofs of Lemmas G.2 and G.3. $\qquad\square$

# I. Proof of Lemma G.2

We consider the following "hard" instance of problem (1):

**(i)** Linear spaces $\mathcal{X}$ and $\mathcal{Y}$ are defined as $\mathcal{X} = (\mathbb{R}^d)^{n_x}$ and $\mathcal{Y} = (\mathbb{R}^d)^{n_y}$, where

$$n_x = 3n \quad \text{and} \quad n_y = \begin{cases} 3n & \mu_{xy} > 0 \\ 3n - 1 & \mu_{xy} = 0 \end{cases}, \quad n \in \{2, 3, \ldots\}. \tag{58}$$

**(ii)** Function $f(x)$ is defined as follows:

$$f(x) = \sum_{i=1}^{n_x} f_i(x_i), \tag{59}$$

where we use the notation $x = (x_1, \ldots, x_{3n}) \in (\mathbb{R}^d)^{3n}$, and functions $f_i(x_i) \colon \mathbb{R}^d \to \mathbb{R}$ are defined as follows:

$$f_i(z) = \begin{cases} \frac{1}{2}\mu_x\|z\|^2 + \frac{1}{2}(L_x - \tilde{\delta}_x)\|\mathbf{F}_1 z\|^2 & i \in \{1, \ldots, n\} \\ \frac{1}{2}\tilde{\delta}_x\|z\|^2 & i \in \{n+1, \ldots, 2n\} \\ \frac{1}{2}\tilde{\delta}_x\|z\|^2 + \frac{1}{2}(L_x - \tilde{\delta}_x)\|\mathbf{F}_2 z\|^2 - A\langle \mathbf{e}_1^d, z\rangle & i \in \{2n+1, \ldots, 3n\} \end{cases}, \tag{60}$$

where $A \in \mathbb{R}$ will be determined later, $\tilde{\delta}_x > 0$ is defined as follows:

$$\tilde{\delta}_x = \mu_x + 4\mu_{xy}^2/L_y, \tag{61}$$

and matrices $\mathbf{F}_1 \in \mathbb{R}^{2\lfloor d/2 \rfloor \times d}$ and $\mathbf{F}_2 \in \mathbb{R}^{2\lfloor (d-1)/2 \rfloor \times d}$ are defined as follows:

$$\mathbf{F}_1 = \frac{1}{\sqrt{2}}\begin{bmatrix} 1 & -1 & 0 \cdot \cdots \cdots \cdots \cdots \cdot \\ 0 & 0 & 1 & -1 & 0 \cdot \cdots \cdot \\ \vdots & & & & \\ \vdots & & & & \end{bmatrix},$$

$$\mathbf{F}_2 = \frac{1}{\sqrt{2}}\begin{bmatrix} 0 & 1 & -1 & 0 \cdot \cdots \cdots \cdots \cdot \\ 0 & 0 & 0 & 1 & -1 & 0 \cdot \cdots \cdot \\ \vdots & & & & \\ \vdots & & & & \end{bmatrix}. \tag{62}$$

**(iii)** Function $g(y)$ is defined as follows:

$$g(y) = \sum_{i=1}^{n_y} g_i(y_i), \tag{63}$$

where we use the notation $y = (y_1, \ldots, y_{n_y}) \in (\mathbb{R}^d)^{n_y}$, and functions $g_i(y_i) \colon \mathbb{R}^d \to \mathbb{R}$ are defined as follows:

$$g_i(y_i) = \begin{cases} \frac{1}{2}\tilde{L}_y\|y_i\|^2 & i = 1 \\ \frac{1}{2}\mu_y\|y_i\|^2 & i \in \{2, \ldots, n_y\} \end{cases}, \quad \text{where} \quad \tilde{L}_y = \begin{cases} L_y & \mu_{xy} > 0 \\ \mu_y & \mu_{xy} = 0 \end{cases}. \tag{64}$$

**(iv)** Matrix $\mathbf{B} \in \mathbb{R}^{n_y d \times n_x d}$ is defined as follows:

$$\mathbf{B} = \begin{cases} \mathbf{E} \otimes \mathbf{I}_d & \mu_{xy} > 0 \\ \mathbf{E}' \otimes \mathbf{I}_d & \mu_{xy} = 0 \end{cases} \tag{65}$$

where matrix $\mathbf{E} \in \mathbb{R}^{3n \times 3n}$ is defined as follows:

$$
\mathbf{E} = \left[
\begin{array}{c|c|c}
\begin{matrix} \gamma \cdots\cdots \gamma \end{matrix} & & \\[2pt]
\hline
\begin{matrix} \beta & & \\ & \ddots & \\ & & \beta \end{matrix} & \begin{matrix} -\beta \\ \vdots \\ -\beta \end{matrix} & \\
\hline
& \begin{matrix} \alpha & -\alpha & \\ & \ddots & \ddots \\ & & \alpha & -\alpha \end{matrix} & \\
\hline
& \begin{matrix} -\beta \\ \vdots \\ -\beta \end{matrix} & \begin{matrix} \beta & & \\ & \ddots & \\ & & \beta \end{matrix}
\end{array}
\right], \tag{66}
$$

and matrix $\mathbf{E}' \in \mathbb{R}^{(3n-1) \times 3n}$ is defined as follows:

$$
\mathbf{E}' = \left[
\begin{array}{c|c|c}
\begin{matrix} \beta & & \\ & \ddots & \\ & & \beta \end{matrix} & \begin{matrix} -\beta \\ \vdots \\ -\beta \end{matrix} & \\
\hline
& \begin{matrix} \alpha & -\alpha & \\ & \ddots & \ddots \\ & & \alpha & -\alpha \end{matrix} & \\
\hline
& \begin{matrix} -\beta \\ \vdots \\ -\beta \end{matrix} & \begin{matrix} \beta & & \\ & \ddots & \\ & & \beta \end{matrix}
\end{array}
\right], \tag{67}
$$

where $\alpha, \beta, \gamma > 0$ are defined as follows:

$$
\alpha = \frac{L_{xy}}{2}, \quad \beta = \frac{L_{xy}}{n}, \quad \gamma = \frac{2\mu_{xy}}{\sqrt{n}}. \tag{68}
$$

One can verify that the problem described above satisfies Assumptions 2.3 to 2.5. Indeed, each function $g_i(z)$ is obviously $L_y$-smooth and $\mu_y$-strongly convex, and each function $f_i(z)$ is $L_x$-smooth and $\mu_x$-strongly convex due to the fact that $\sigma_{\max}^2(\mathbf{F}_1) = \sigma_{\max}^2(\mathbf{F}_2) = 1$, and $\mu_x \leq \tilde{\delta}_x \leq L_x$, where the latter inequality is implied by Assumption 2.7 as follows:

$$
\tilde{\delta}_x = \mu_x + 4\mu_{xy}^2 / L_y < \tfrac{1}{4}L_x + \tfrac{1}{4}L_x = \tfrac{1}{2}L_x. \tag{69}
$$

Moreover, we establish the following Lemma I.1, which describes the spectral properties of matrix $\mathbf{B}$. The proof is available in Appendix I.1.2.

**Lemma I.1.** *Let $n \in \{2, 3, \ldots\}$ and $\alpha, \beta > 0$. Then for $\gamma > 0$, the singular values of matrix $\mathbf{E}$ defined in equation (66) can be bounded as follows:*

$$
\min\left\{ \frac{n\gamma^2}{4}, \frac{\beta^2}{36}, \frac{\alpha^2}{9n^2} \right\} \leq \sigma_{\min}^2(\mathbf{E}) \leq \sigma_{\max}^2(\mathbf{E}) \leq \max\left\{ 2n\gamma^2, 2(n+1)\beta^2, 4\alpha^2 \right\}, \tag{70}
$$

*and for $\gamma = 0$, the singular values of matrix $\mathbf{E}'$ defined in equation (67) can be bounded as follows:*

$$
\min\left\{ \frac{\beta^2}{36}, \frac{\alpha^2}{9n^2} \right\} \leq (\sigma_{\min}^+(\mathbf{E}'))^2 \leq \sigma_{\max}^2(\mathbf{E}') \leq \max\left\{ 2(n+1)\beta^2, 4\alpha^2 \right\}. \tag{71}
$$

Using Lemma I.1 and the definition of $\alpha, \beta, \gamma$ in equation (68), we can show that matrix $\mathbf{B}$ satisfies Assumption 2.5 as long as $n$ is defined as follows:

$$
n = \left\lfloor \frac{L_{xy}}{6\mu_{yx}} \right\rfloor. \tag{72}
$$

Indeed, the definition of $n$ in equation (72), the definitions of $\alpha, \beta, \gamma$ in equation (68), and Lemma I.1 imply

$$\mu_{xy}^2 = \mu_{yx}^2 \leq \sigma_{\min}^2(\mathbf{E}) \leq \sigma_{\max}^2(\mathbf{E}) \leq L_{xy}^2 \tag{73}$$

in the case $\mu_{xy} > 0$, and

$$\mu_{yx}^2 \leq (\sigma_{\min}^+(\mathbf{E}'))^2 \leq \sigma_{\max}^2(\mathbf{E}') \leq L_{xy}^2 \tag{74}$$

in the case $\mu_{xy} = 0$. Moreover, it is not hard to verify that $\operatorname{range} \mathbf{B}^\top = \mathcal{X}$ in the case $\mu_{xy} > 0$ and $\operatorname{range} \mathbf{B} = \mathcal{Y}$ in both cases. Also note that $n \geq 3$ due to Assumption 2.7.

Next, we establish the following Lemma I.2, which describes the solution to the problem defined above, the proof is available in Appendix I.1.1.

**Lemma I.2.** *For all $d \in \{1, 2, \ldots\}$, the instance of problem (1) defined above has a unique solution $(x^*, y^*) \in \mathcal{X} \times \mathcal{Y}$. Moreover, there exists a vector $(x^\circ, y^\circ) \in (\mathcal{L}_{d,\rho})^{n_x} \times (\mathcal{L}_{d,\rho})^{n_y}$ such that the following inequality holds:*

$$\mathcal{R}_{\delta_x \delta_y}^2(x^\circ, y^\circ) \leq C_\pi A^2 \rho^{2d}, \tag{75}$$

*where $C_\pi > 0$ is some constant that possibly depends on the parameters $\pi \in \Pi$, but does not depend on $d$, $\mathcal{L}_{d,\rho} \subset \mathbb{R}^d$ is a linear space which is defined for $\rho \in (0, 1)$ as follows:*

$$\mathcal{L}_{d,\rho} = \operatorname{range} \begin{bmatrix} 1 & 0 & 0 & 0 \cdots \cdots \cdots \cdots \cdots \cdots \cdots \\ 1 & 0 & \rho^2 & 0 & \rho^4 \cdots \cdots \cdots \cdots \\ 0 & 1 & 0 & \rho^2 & 0 & \rho^4 \cdots \cdots \end{bmatrix}^\top, \tag{76}$$

*and $\rho \in (0, 1)$ satisfies the following inequality:*

$$\rho \geq \max\left\{ 1 - 89 \cdot \frac{n}{\sqrt{\kappa_{xy}}}, \frac{1}{346} \right\}. \tag{77}$$

Let $(x^0, y^0) = (x_1^0, \ldots, x_{n_x}^0, y_1^0, \ldots, y_{n_y}^0) \in \mathcal{X} \times \mathcal{Y}$ be defined as follows:

$$(x^0, y^0) = \operatorname{proj}_{\operatorname{span}(\{\mathbf{e}_1^d\})^{n_x + n_y}}((x^*, y^*)) = (\mathbf{I}_{n_x + n_y} \otimes \mathbf{P})(x^*, y^*), \tag{78}$$

where $\mathbf{P} \in \mathbb{R}^{d \times d}$ is the orthogonal projection matrix onto the linear space $\operatorname{span}(\{\mathbf{e}_1^d\}) \subset \mathbb{R}^d$, which is given as follows:

$$\mathbf{P} = \mathbf{e}_1^d (\mathbf{e}_1^d)^\top. \tag{79}$$

Then vector $(x^\circ, y^\circ)$ from Lemma I.2 satisfies the following relation:

$$\begin{aligned}
(\mathbf{I}_{n_x + n_y} \otimes (\mathbf{I}_d - \mathbf{P}))(x^\circ, y^\circ) &= (u_x, u_y) \otimes (0, 1, 0, \rho^2, \ldots) + (v_x, v_y) \otimes (0, 0, 1, 0, \rho^2, \ldots), \\
\text{where} \quad u_x &= (u_{x,1}, \ldots, u_{x,n_x}) \in \mathbb{R}^{n_x}, \quad v_x = (v_{x,1}, \ldots, v_{x,n_x}) \in \mathbb{R}^{n_x}, \\
u_y &= (u_{y,1}, \ldots, u_{y,n_y}) \in \mathbb{R}^{n_y}, \quad v_y = (v_{y,1}, \ldots, v_{y,n_y}) \in \mathbb{R}^{n_y}.
\end{aligned} \tag{80}$$

Hence, we can obtain the following relation:

$$\begin{aligned}
\mathcal{R}_{\delta_x \delta_y}^2(x^0, y^0) &= \delta_x \|x^* - x^0\|^2 + \delta_y \|y^* - y^0\|^2 \\
&\overset{(a)}{=} \delta_x \|(\mathbf{I}_{n_x} \otimes (\mathbf{I}_d - \mathbf{P}))x^*\|^2 + \delta_y \|(\mathbf{I}_{n_y} \otimes (\mathbf{I}_d - \mathbf{P}))y^*\|^2 \\
&= \delta_x \|(\mathbf{I}_{n_x} \otimes (\mathbf{I}_d - \mathbf{P}))(x^* - x^\circ + x^\circ)\|^2 + \delta_y \|(\mathbf{I}_{n_y} \otimes (\mathbf{I}_d - \mathbf{P}))(y^* - y^\circ + y^\circ)\|^2 \\
&\overset{(b)}{\leq} 2\delta_x \|(\mathbf{I}_{n_x} \otimes (\mathbf{I}_d - \mathbf{P}))(x^* - x^\circ)\|^2 + 2\delta_y \|(\mathbf{I}_{n_y} \otimes (\mathbf{I}_d - \mathbf{P}))(y^* - y^\circ)\|^2 \\
&\quad + 2\delta_x \|(\mathbf{I}_{n_x} \otimes (\mathbf{I}_d - \mathbf{P}))x^\circ\|^2 + 2\delta_y \|(\mathbf{I}_{n_y} \otimes (\mathbf{I}_d - \mathbf{P}))y^\circ\|^2 \\
&\leq 2\delta_x \|(\mathbf{I}_{n_x} \otimes (\mathbf{I}_d - \mathbf{P}))x^\circ\|^2 + 2\delta_y \|(\mathbf{I}_{n_y} \otimes (\mathbf{I}_d - \mathbf{P}))y^\circ\|^2 + 2\mathcal{R}_{\delta_x \delta_y}^2(x^\circ, y^\circ) \\
&\overset{(c)}{=} 2(\delta_x \|u_x\|^2 + \delta_y \|u_y\|^2)\|(0, 1, 0, \rho^2, \ldots)\|^2 \\
&\quad + 2(\delta_x \|v_x\|^2 + \delta_y \|v_y\|^2)\|(0, 0, 1, 0, \rho^2, \ldots)\|^2 + 2\mathcal{R}_{\delta_x \delta_y}^2(x^\circ, y^\circ)
\end{aligned}$$

$$= 2(\delta_x\|u_x\|^2 + \delta_y\|u_y\|^2)\sum_{j=0}^{\lfloor d/2\rfloor-1}\rho^{4j}$$

$$+ 2(\delta_x\|v_x\|^2 + \delta_y\|v_y\|^2)\sum_{j=0}^{\lfloor(d-1)/2\rfloor-1}\rho^{4j} + 2\mathcal{R}^2_{\delta_x\delta_y}(x^\circ, y^\circ)$$

$$= \frac{2(\delta_x\|u_x\|^2 + \delta_y\|u_y\|^2)(1 - \rho^{4\lfloor d/2\rfloor})}{1 - \rho^4}$$

$$+ \frac{2(\delta_x\|v_x\|^2 + \delta_y\|v_y\|^2)(1 - \rho^{4\lfloor(d-1)/2\rfloor})}{1 - \rho^4} + 2\mathcal{R}^2_{\delta_x\delta_y}(x^\circ, y^\circ)$$

$$\leq \frac{2(\delta_x\|(u_x, v_x)\|^2 + \delta_y\|(u_y, v_y)\|^2)}{1 - \rho^4} + 2\mathcal{R}^2_{\delta_x\delta_y}(x^\circ, y^\circ),$$

where (a) uses the definition of $(x^0, y^0)$ in equation (78); (b) uses Young's inequality; (c) uses equation (80) and the properties of the Kronecker product.

Further, we fix $k \in \{1, \ldots, d\}$. using the sparse structure of the matrices $\mathbf{F}_1, \mathbf{F}_2$ and $\mathbf{E}$ defined in equations (62) and (66), respectively, and using the standard arguments (Nesterov, 2013; Ibrahim et al., 2020; Zhang et al., 2022a; Scaman et al., 2017; 2018; Kovalev et al., 2024), we can show that the output vectors $x_o(\tau) = (x_{o,1}(\tau), \ldots, x_{o,3n}(\tau)) \in \mathcal{X}$ and $y_o(\tau) = (y_{o,1}(\tau), \ldots, y_{o,3n}(\tau)) \in \mathcal{Y}$ satisfy the following implication:

$$\tau \leq D \cdot \tau_{\mathbf{B}} n(k-1) \quad \Rightarrow \quad x_{o,i}(\tau), y_{o,j}(\tau) \in \mathrm{span}(\{\mathbf{e}_1^d, \ldots, \mathbf{e}_k^d\}) \tag{81}$$

for all $i \in \{1, \ldots, n_x\}$ and $j \in \{1, \ldots, n_y\}$, where $D > 0$ is a universal constant. The right-hand side of this implication implies the following:

$$\mathcal{R}^2_{\delta_x\delta_y}(x_o(\tau), y_o(\tau)) \overset{(a)}{\geq} \tfrac{1}{2}\delta_x\|x_o(\tau) - x^\circ\|^2 + \tfrac{1}{2}\delta_y\|y_o(\tau) - y^\circ\|^2 - \mathcal{R}^2_{\delta_x\delta_y}(x^\circ, y^\circ)$$

$$\overset{(b)}{\geq} \frac{(\delta_x\|u_x\|^2 + \delta_y\|u_y\|^2)}{2}\sum_{j=\lfloor k/2\rfloor}^{\lfloor d/2\rfloor-1}\rho^{4j}$$

$$+ \frac{(\delta_x\|v_x\|^2 + \delta_y\|v_y\|^2)}{2}\sum_{j=\lfloor(k-1)/2\rfloor}^{\lfloor(d-1)/2\rfloor-1}\rho^{4j} - \mathcal{R}^2_{\delta_x\delta_y}(x^\circ, y^\circ)$$

$$= \frac{(\delta_x\|u_x\|^2 + \delta_y\|u_y\|^2)(\rho^{4\lfloor k/2\rfloor} - \rho^{4\lfloor d/2\rfloor})}{2(1 - \rho^4)}$$

$$+ \frac{(\delta_x\|v_x\|^2 + \delta_y\|v_y\|^2)(\rho^{4\lfloor(k-1)/2\rfloor} - \rho^{4\lfloor(d-1)/2\rfloor})}{2(1 - \rho^4)} - \mathcal{R}^2_{\delta_x\delta_y}(x^\circ, y^\circ)$$

$$\geq \frac{(\delta_x\|(u_x, v_x)\|^2 + \delta_y\|(u_y, v_y)\|^2)(\rho^{2k} - \rho^{2d-4})}{2(1 - \rho^4)} - \mathcal{R}^2_{\delta_x\delta_y}(x^\circ, y^\circ)$$

$$\overset{(c)}{\geq} \frac{(\rho^{2k} - \rho^{2d-4})}{4}\left(\mathcal{R}^2_{\delta_x\delta_y}(x^0, y^0) - 2\mathcal{R}^2_{\delta_x\delta_y}(x^\circ, y^\circ)\right) - \mathcal{R}^2_{\delta_x\delta_y}(x^\circ, y^\circ)$$

$$= \frac{(\rho^{2k} - \rho^{2d-4})}{4}\mathcal{R}^2_{\delta_x\delta_y}(x^0, y^0) - \left(1 + \frac{(\rho^{2k} - \rho^{2d-4})}{2}\right)\mathcal{R}^2_{\delta_x\delta_y}(x^\circ, y^\circ)$$

$$\overset{(d)}{\geq} \frac{(\rho^{2k} - \rho^{2d-4})}{4}\mathcal{R}^2_{\delta_x\delta_y}(x^0, y^0) - \left(1 + \frac{(\rho^{2k} - \rho^{2d-4})}{2}\right)C_\pi A^2\rho^{2d},$$

where (a) uses Young's inequality; (b) uses equation (81) and the expression for $(x^\circ, y^\circ)$ in equation (80); (c) uses the previously obtained upper bound on $\mathcal{R}^2_{\delta_x\delta_y}(x^0, y^0)$; (d) uses Lemma I.2.

Next, we establish the following Lemma I.3, the proof is available in Appendix I.1.9.

**Lemma I.3.** *For all $d \in \{2, 3, \ldots\}$, the unique solution to the instance of problem (1) defined above satisfies the following relation:*

$$\mathcal{R}^2_{\delta_x\delta_y}(x^0, y^0) = B_{\pi,d}A^2, \tag{82}$$

where $B_{\pi,d} > 0$ is a constant that possibly depends on $d \in \{2, 3, \ldots\}$ and the parameters $\pi \in \Pi$, Moreover there exists $\hat{d} \in \{2, 3, \ldots\}$ such that the following inequality holds:

$$\min_{d \in \{\hat{d}, \hat{d}+1, \ldots\}} B_{\pi,d} > 0 \tag{83}$$

Using Lemma I.3, for $d \geq \hat{d}$, we can further lower-bound $\mathcal{R}^2_{\delta_x \delta_y}(x_o(\tau), y_o(\tau))$ as follows:

$$\mathcal{R}^2_{\delta_x \delta_y}(x_o(\tau), y_o(\tau)) \geq \frac{(\rho^{2k} - \rho^{2d-4})}{4} B_{\pi,d} A^2 - \left(1 + \frac{(\rho^{2k} - \rho^{2d-4})}{2}\right) C_\pi A^2 \rho^{2d}. \tag{84}$$

Next, we can choose $A = R/\sqrt{B_{\pi,d}}$ to ensure $\mathcal{R}^2_{\delta_x \delta_y}(x^0, y^0) = R^2$ and obtain the following:

$$\begin{aligned}
\mathcal{R}^2_{\delta_x \delta_y}(x_o(\tau), y_o(\tau)) &\geq \frac{(\rho^{2k} - \rho^{2d-4})}{4} R^2 - \left(1 + \frac{(\rho^{2k} - \rho^{2d-4})}{2}\right) \frac{C_\pi R^2 \rho^{2d}}{B_{\pi,d}} \\
&\overset{(a)}{\geq} \frac{(\rho^{2k} - \rho^{2d-4})}{4} R^2 - \left(1 + \frac{(\rho^{2k} - \rho^{2d-4})}{2}\right) \frac{C_\pi R^2 \rho^{2d}}{\min_{d \in \{\hat{d}, \hat{d}+1, \ldots\}} B_{\pi,d}} \\
&\overset{(b)}{\geq} \frac{1}{5} \rho^{2k} R^2,
\end{aligned}$$

where (a) uses Lemma I.3; (b) is implied by choosing a large enough value of $d$. The rest of the proof uses the lower bound on $\rho$ in Lemma I.2 and is almost identical to the final steps of the proof of Lemma G.3 in Appendix J. $\qquad\square$

## I.1. Proofs of Auxiliary Lemmas

### I.1.1. PROOF OF LEMMA I.2

In this proof, we consider the case $\mu_{xy} > 0$, since the case $\mu_{xy} = 0$ is almost identical. Using the first-order optimality conditions (7) in problem (1), we obtain the following expression for the optimal dual variable $y^* \in \mathcal{Y}$:

$$y^* = \nabla g^*(\mathbf{B}x^*) \Rightarrow y^* = \left(\left(\begin{bmatrix} 1/L_y & & & \\ & 1/\mu_y & & \\ & & \ddots & \\ & & & 1/\mu_y \end{bmatrix} \mathbf{E}\right) \otimes \mathbf{I}_d\right) x^*, \tag{85}$$

where the optimal primal variable is the solution to the primal minimization problem in equation (5):

$$x^* = \arg\min_{x \in \mathcal{X}} f(x) + g^*(\mathbf{B}x). \tag{86}$$

Moreover, using the definition of functions $f(x)$ and $g(y)$ in equations (59) and (63) and the definition of matrix $\mathbf{B}$ in equation (65), we can rewrite this problem as follows:

$$\begin{aligned}
\min_{x \in \mathcal{X}} \quad & \sum_{i=1}^{n} \left(f_1(x_i) + \frac{\beta^2}{2\mu_y} \|x_i - x_{n+1}\|^2\right) + \sum_{i=2n+1}^{3n} \left(f_{3n}(x_i) + \frac{\beta^2}{2\mu_y} \|x_i - x_{2n}\|^2\right) \\
& + \frac{\gamma^2}{2L_y} \|\tfrac{1}{n} \sum_{i=1}^{n} x_i\|^2 + \sum_{i=n+1}^{2n} \frac{\tilde{\delta}_x}{2} \|x_i\|^2 + \sum_{i=n+1}^{2n-1} \frac{\alpha^2}{2\mu_y} \|x_{i+1} - x_i\|^2.
\end{aligned} \tag{87}$$

It is also not hard to verify that the following inequality holds:

$$\sum_{i=1}^{n} \left(f_1(x_i) + \frac{\beta^2}{2\mu_y} \|x_i - x_{n+1}\|^2\right) \geq n f_1(\tfrac{1}{n} \sum_{i=1}^{n} x_i) + \frac{n\beta^2}{2\mu_y} \|\tfrac{1}{n} \sum_{i=1}^{n} x_i - x_{n+1}\|^2, \tag{88}$$

where equality is attained if and only if $x_1 = \cdots = x_n$. Consequently, the problem can be further reformulated as follows:

$$\min_{x \in \mathcal{X}} \quad n M_{\frac{\mu_y}{\beta^2} h_1}(x_{n+1}) + n M_{\frac{\mu_y}{\beta^2} h_2}(x_{2n}) + \sum_{i=n+1}^{2n} \frac{\tilde{\delta}_x}{2} \|x_i\|^2 + \sum_{i=n+1}^{2n-1} \frac{\alpha^2}{2\mu_y} \|x_{i+1} - x_i\|^2, \tag{89}$$

where functions $h_1(z), h_2(z) \colon \mathbb{R}^d \to \mathbb{R}$ are defined as follows:

$$
\begin{aligned}
h_1(z) &= \frac{\tilde{\delta}_x}{2} \|z\|^2 + \frac{L_x - \tilde{\delta}_x}{2} \|\mathbf{F}_1 z\|^2, \\
h_2(z) &= \frac{\tilde{\delta}_x}{2} \|z\|^2 + \frac{L_x - \tilde{\delta}_x}{2} \|\mathbf{F}_2 z\|^2 - A\langle \mathbf{e}_1^d, z \rangle,
\end{aligned}
\tag{90}
$$

and $M_{\frac{\mu_y}{\beta^2} h_1}(z)$ and $M_{\frac{\mu_y}{\beta^2} h_2}(z)$ are the corresponding Moreau envelopes. Moreover, the solution $x^*$ satisfies the following relations:

$$
\begin{aligned}
x_1^* &= \cdots = x_n^* = \operatorname{prox}_{\frac{\mu_y}{\beta^2} h_1}(x_{n+1}^*) \\
x_{2n+1}^* &= \cdots = x_{3n}^* = \operatorname{prox}_{\frac{\mu_y}{\beta^2} h_2}(x_{2n}^*)
\end{aligned}
\tag{91}
$$

Further, we perform the minimization in the variables $x_{n+2}, \ldots, x_{2n-1}$. Using the first-order optimality conditions, we obtain the following relations:

$$
\left(2 + \frac{\tilde{\delta}_x \mu_y}{\alpha^2}\right) x_i^* = x_{i-1}^* + x_{i+1}^* \quad \text{for} \quad i \in \{n+2, \ldots, 2n-1\}.
\tag{92}
$$

This is nothing else but a linear recurrence, which is not hard to solve. Let $q > 0$ be the smallest root of the following characteristic polynomial:

$$
\left(2 + \frac{\tilde{\delta}_x \mu_y}{\alpha^2}\right) q = 1 + q^2,
\tag{93}
$$

which is given as follows:

$$
q = \frac{\sqrt{4\alpha^2 + \tilde{\delta}_x \mu_y} - \sqrt{\tilde{\delta}_x \mu_y}}{\sqrt{4\alpha^2 + \tilde{\delta}_x \mu_y} + \sqrt{\tilde{\delta}_x \mu_y}}.
\tag{94}
$$

Then $x_{n+2}^*, \ldots, x_{2n-1}^*$ can be expressed as follows:

$$
x_{n+i}^* = \frac{x_{n+1}^*(q^{n-i} - q^{i-n}) + x_{2n}^*(q^{i-1} - q^{1-i})}{(q^{n-1} - q^{1-n})}.
\tag{95}
$$

Moreover, one can observe the following:

$$
\begin{aligned}
&\sum_{i=n+1}^{2n} \frac{\tilde{\delta}_x}{2} \|x_i^*\|^2 + \sum_{i=n+1}^{2n-1} \frac{\alpha^2}{2\mu_y} \|x_{i+1}^* - x_i^*\|^2 \\
&= \sum_{i=n+1}^{2n} \frac{\tilde{\delta}_x}{2} \|x_i^*\|^2 + \sum_{i=n+1}^{2n-1} \frac{\alpha^2}{2\mu_y} \left( \|x_i^*\|^2 + \|x_{i+1}^*\|^2 - 2\langle x_{i+1}^*, x_i^* \rangle \right) \\
&= \left( \frac{\tilde{\delta}_x}{2} + \frac{\alpha^2}{2\mu_y} \right) \left( \|x_{n+1}^*\|^2 + \|x_{2n}^*\|^2 \right) + \sum_{i=n+2}^{2n-1} \left( \frac{\tilde{\delta}_x}{2} + \frac{\alpha^2}{\mu_y} \right) \|x_i^*\|^2 - \sum_{i=n+1}^{2n-1} \frac{\alpha^2}{\mu_y} \langle x_{i+1}^*, x_i^* \rangle \\
&= \left( \frac{\tilde{\delta}_x}{2} + \frac{\alpha^2}{2\mu_y} \right) \left( \|x_{n+1}^*\|^2 + \|x_{2n}^*\|^2 \right) - \frac{\alpha^2}{2\mu_y} \left( \langle x_{n+1}^*, x_{n+2}^* \rangle + \langle x_{2n}^*, x_{2n-1}^* \rangle \right) \\
&\quad + \sum_{i=n+2}^{2n-1} \frac{\alpha^2}{2\mu_y} \left( \left( \frac{\tilde{\delta}_x \mu_y}{\alpha^2} + 2 \right) \|x_i^*\|^2 - \langle x_i^*, x_{i+1}^* + x_{i-1}^* \rangle \right) \\
&\overset{(a)}{=} \left( \frac{\tilde{\delta}_x}{2} + \frac{\alpha^2}{2\mu_y} \right) \left( \|x_{n+1}^*\|^2 + \|x_{2n}^*\|^2 \right) - \frac{\alpha^2}{2\mu_y} \left( \langle x_{n+1}^*, x_{n+2}^* \rangle + \langle x_{2n}^*, x_{2n-1}^* \rangle \right) \\
&\overset{(b)}{=} \left( \frac{\tilde{\delta}_x}{2} + \frac{\alpha^2}{2\mu_y} \right) \left( \|x_{n+1}^*\|^2 + \|x_{2n}^*\|^2 \right)
\end{aligned}
$$

$$- \frac{\alpha^2}{2\mu_y} \cdot \frac{(\|x_{n+1}^*\|^2 + \|x_{2n}^*\|^2)(q^{n-2} - q^{2-n}) + 2\langle x_{n+1}^*, x_{2n}^* \rangle(q - q^{-1})}{(q^{n-1} - q^{1-n})}$$

$$= \left( \frac{\tilde{\delta}_x}{2} + \frac{\alpha^2}{2\mu_y} \left( 1 - \frac{(q^{n-2} - q^{2-n} + q - q^{-1})}{(q^{n-1} - q^{1-n})} \right) \right) (\|x_{n+1}^*\|^2 + \|x_{2n}^*\|^2)$$

$$+ \frac{\alpha^2}{2\mu_y} \cdot \frac{(q - q^{-1})}{(q^{n-1} - q^{1-n})} \|x_{n+1}^* - x_{2n}^*\|^2$$

$$= \left( \frac{\tilde{\delta}_x}{2} + \frac{\alpha^2}{2\mu_y} \cdot \frac{(1-q)(1-q^{n-2})}{(1+q^{n-1})} \right) (\|x_{n+1}^*\|^2 + \|x_{2n}^*\|^2)$$

$$+ \frac{\alpha^2}{2\mu_y} \cdot \frac{(1-q)(1+q)}{q(q^{1-n} - q^{n-1})} \|x_{n+1}^* - x_{2n}^*\|^2$$

$$= \left( \frac{\tilde{\delta}_x}{2} + \frac{\omega_n \alpha^2}{2\mu_y} \right) (\|x_{n+1}^*\|^2 + \|x_{2n}^*\|^2) + \frac{\nu_n \alpha^2}{2\mu_y} \|x_{n+1}^* - x_{2n}^*\|^2$$

where (a) uses equations (93) and (95); (b) uses equation (95), and $\omega_n, \nu_n > 0$ are defined as follows:

$$\omega_n = \frac{(1-q)(1-q^{n-2})}{(1+q^{n-1})}, \quad \nu_n = \frac{(1-q)(1+q)}{q(q^{1-n} - q^{n-1})}. \tag{96}$$

In addition, we can observe that the following relation holds:

$$\frac{1}{2} \|x_{n+1}^* - x_{2n}^*\|^2 = \min_{z \in \mathbb{R}^d} \left( \|z - x_{n+1}^*\|^2 + \|z - x_{2n}^*\|^2 \right). \tag{97}$$

Therefore, the problem can be further reformulated as follows:

$$\min_{x \in \mathcal{X}} \min_{z \in \mathbb{R}^d} M_{\frac{\mu_y}{\beta^2} h_1}(x_{n+1}) + \left( \frac{\tilde{\delta}_x}{2n} + \frac{\omega_n \alpha^2}{2n\mu_y} \right) \|x_{n+1}\|^2 + \frac{\nu_n \alpha^2}{n\mu_y} \|x_{n+1} - z\|^2$$

$$+ M_{\frac{\mu_y}{\beta^2} h_2}(x_{2n}) + \left( \frac{\tilde{\delta}_x}{2n} + \frac{\omega_n \alpha^2}{2n\mu_y} \right) \|x_{2n}\|^2 + \frac{\nu_n \alpha^2}{n\mu_y} \|x_{2n} - z\|^2. \tag{98}$$

Let functions $h_1^+(z), h_2^+(z) \colon \mathbb{R}^d \to \mathbb{R}$ be defined as follows:

$$h_j^+(z) = M_{\frac{\mu_y}{\beta^2} h_j}(z) + \left( \frac{\tilde{\delta}_x}{2n} + \frac{\omega_n \alpha^2}{2n\mu_y} \right) \|z\|^2, \quad j = 1, 2, \tag{99}$$

and let functions $h_1^{++}(z), h_2^{++}(z) \colon \mathbb{R}^d \to \mathbb{R}$ be defined as follows:

$$h_j^{++}(z) = M_{\frac{n\mu_y}{2\nu_n \alpha^2} h_j^+}(z), \quad j = 1, 2. \tag{100}$$

Then the latter problem reformulation can be rewritten as follows:

$$z^* = \arg\min_{z \in \mathbb{R}^d} h_1^{++}(z) + h_2^{++}(z), \tag{101}$$

and the solution $x^* \in \mathcal{X}$ satisfies the following relation:

$$x_{n+1}^* = \text{prox}_{\frac{n\mu_y}{2\nu_n \alpha^2} h_1^+}(z^*), \quad x_{2n}^* = \text{prox}_{\frac{n\mu_y}{2\nu_n \alpha^2} h_2^+}(z^*). \tag{102}$$

Next, we establish the following Lemma I.4, which is used to obtain the explicit expressions for the Moreau envelopes. The proof is available in Appendix I.1.3.

**Lemma I.4.** *Let function $h(z)\colon \mathbb{R}^d \to \mathbb{R}$ be defined as follows:*

$$h(z) = \frac{\mu}{2}\|z\|^2 + \frac{L-\mu}{2}\|\mathbf{F}z\|^2 - \langle b, z \rangle, \tag{103}$$

*where $L > \mu > 0$, and $\mathbf{F} \in \mathbb{R}^{p \times d}$ and $b \in \mathbb{R}^d$ satisfy the following assumptions:*

$$\mathbf{F}\mathbf{F}^\top = \mathbf{I}_p, \quad \mathbf{F}^\top \mathbf{F} b = 0. \tag{104}$$

*Then for $\lambda > 0$, the Moreau envelope $M_{\lambda h}(z)$ is given as follows:*

$$M_{\lambda h}(z) = \frac{\mu_\lambda}{2}\|z\|^2 + \frac{L_\lambda - \mu_\lambda}{2}\|\mathbf{F}z\|^2 - B_\lambda \langle b, z \rangle - C_\lambda, \tag{105}$$

*where constants $L_\lambda > \mu_\lambda > 0$ and $B_\lambda, C_\lambda \in \mathbb{R}$ are defined as follows:*

$$L_\lambda = (\lambda + 1/L)^{-1}, \quad \mu_\lambda = (\lambda + 1/\mu)^{-1}, \quad B_\lambda = (1 + \lambda\mu)^{-1}, \quad C_\lambda = \frac{\lambda\|b\|^2}{2(1 + \lambda\mu)}. \tag{106}$$

Using Lemma I.4 and the definition of functions $h_1(z), h_2(z)$ in equation (90), we can express functions $h_1^+(z), h_2^+(z)$ as follows:

$$\begin{aligned}
h_1^+(z) &= \frac{\mu^+}{2}\|z\|^2 + \frac{L^+ - \mu^+}{2}\|\mathbf{F}_1 z\|^2 + \text{const}, \\
h_2^+(z) &= \frac{\mu^+}{2}\|z\|^2 + \frac{L^+ - \mu^+}{2}\|\mathbf{F}_2 z\|^2 - A^+ \langle \mathbf{e}_1^d, z \rangle + \text{const},
\end{aligned} \tag{107}$$

and functions $h_1^{++}(z), h_2^{++}(z)$ can be expressed as follows:

$$\begin{aligned}
h_1^{++}(z) &= \frac{\mu^{++}}{2}\|z\|^2 + \frac{L^{++} - \mu^{++}}{2}\|\mathbf{F}_1 z\|^2 + \text{const}, \\
h_2^{++}(z) &= \frac{\mu^{++}}{2}\|z\|^2 + \frac{L^{++} - \mu^{++}}{2}\|\mathbf{F}_2 z\|^2 - A^{++} \langle \mathbf{e}_1^d, z \rangle + \text{const},
\end{aligned} \tag{108}$$

where constants $L^+ > \mu^+ > 0$ are defined as follows:

$$L^+ = \left(\frac{1}{L_x} + \frac{\mu_y}{\beta^2}\right)^{-1} + \frac{\tilde{\delta}_x}{n} + \frac{\omega_n \alpha^2}{n\mu_y}, \quad \mu^+ = \left(\frac{1}{\tilde{\delta}_x} + \frac{\mu_y}{\beta^2}\right)^{-1} + \frac{\tilde{\delta}_x}{n} + \frac{\omega_n \alpha^2}{n\mu_y}, \tag{109}$$

constants $L^{++} > \mu^{++} > 0$ are defined as follows:

$$L^{++} = \left(\frac{1}{L^+} + \frac{n\mu_y}{2\nu_n \alpha^2}\right)^{-1}, \quad \mu^{++} = \left(\frac{1}{\mu^+} + \frac{n\mu_y}{2\nu_n \alpha^2}\right)^{-1}, \tag{110}$$

and constants $A^+, A^{++} \in R$ are defined as follows:

$$A^+ = \frac{\beta^2 A}{\beta^2 + \mu_y \tilde{\delta}_x}, \quad A^{++} = \frac{2\nu_n \alpha^2 A^+}{2\nu_n \alpha^2 + n\mu_y \mu^+}. \tag{111}$$

Next, we establish the following Lemma I.5, the proof is available in Appendix I.1.4.

**Lemma I.5.** *Let $\hat{z} = (\hat{z}_1, \ldots, \hat{z}_d) \in \mathbb{R}^d$ be defined as follows:*

$$\hat{z} = \arg\min_{z \in \mathbb{R}^d} \mu\|z\|^2 + \frac{L-\mu}{2}\|\mathbf{F}z\|^2 - B\langle \mathbf{e}_1^d, z \rangle, \tag{112}$$

*where $L > \mu > 0$, $B \in \mathbb{R}$ and matrix $\mathbf{F} \in \mathbb{R}^{(d-1) \times d}$ is defined as follows:*

$$\mathbf{F} = \frac{1}{\sqrt{2}}\begin{bmatrix} 1 & -1 & & \\ & \ddots & \ddots & \\ & & \ddots & \ddots \\ & & & 1 & -1 \end{bmatrix}. \tag{113}$$

*Then there exists $z^\circ \in \operatorname{span}(\{(1, \rho, \ldots, \rho^{d-1})\})$ such that the following inequality holds:*

$$\|z^\circ - \hat{z}\| \leq \frac{B\rho^d}{2\mu}, \tag{114}$$

*where $\rho \in (0, 1)$ is defined as follows:*

$$\rho = \frac{\sqrt{L} - \sqrt{\mu}}{\sqrt{L} + \sqrt{\mu}}. \tag{115}$$

From Lemma I.5, the definition of $z^*$ in equation (101), and the definition of functions $h_j(z)$, $h_j^+(z)$, and $h_j^{++}(z)$ in equations (90), (107) and (108), it follows that there exists $z^\circ \in \operatorname{span}(\{(1, \rho, \ldots, \rho^{d-1})\})$ such that the following inequality holds:

$$\|z^\circ - z^*\| \leq \frac{A^{++}\rho^d}{2\mu^{++}}, \tag{116}$$

where $\rho \in (0, 1)$ is defined as follows:

$$\rho = \frac{\sqrt{L^{++}} - \sqrt{\mu^{++}}}{\sqrt{L^{++}} + \sqrt{\mu^{++}}}. \tag{117}$$

Finally, we obtain the desired statement of Lemma I.2 with the help of the following Lemma I.6. The proof is available in Appendix I.1.8.

**Lemma I.6.** *Let vector $(x^\circ, y^\circ) = (x_1^\circ, \ldots, x_{n_x}^\circ, y_1^\circ, \ldots, y_{n_y}^\circ) \in \mathcal{X} \times \mathcal{Y}$ be defined as follows:*

$$x_i^\circ = \operatorname{proj}_{\mathcal{L}_{d,\rho}}(x_i'), \quad y_i^\circ = \operatorname{proj}_{\mathcal{L}_{d,\rho}}(y_i'), \tag{118}$$

*where vector $(x', y') = (x_1', \ldots, x_{n_x}', y_1', \ldots, y_{n_y}') \in \mathcal{X} \times \mathcal{Y}$ is defined as follows:*

$$x_i' = \begin{cases} \operatorname{prox}_{\frac{\mu_y}{\beta^2}h_1}(x_{n+1}') & i \in \{1, \ldots, n\} \\ \operatorname{prox}_{\frac{n\mu_y}{2\nu_n\alpha^2}h_1^+}(z^\circ) & i = n+1 \\ \frac{x_{n+1}'(q^{2n-i}-q^{i-2n})+x_{2n}'(q^{i-(n+1)}-q^{(n+1)-i})}{(q^{n-1}-q^{1-n})} & i \in \{n+2, \ldots, 2n-1\} \\ \operatorname{prox}_{\frac{n\mu_y}{2\nu_n\alpha^2}h_2^+}(z^\circ) & i = 2n \\ \operatorname{prox}_{\frac{\mu_y}{\beta^2}h_2}(x_{2n}') & i \in \{2n+1, \ldots, 3n\} \end{cases} \tag{119}$$
$$y' = \nabla g^*(\mathbf{B}x').$$

*Then the following inequality holds:*

$$\mathcal{R}_{\delta_x\delta_y}^2(x^\circ, y^\circ) \leq C_\pi A^2 \rho^{2d}. \tag{120}$$

*where $C_\pi > 0$ is some constant that possibly depends on the parameters $\pi \in \Pi$*

It remains to lower-bound $\rho$. It is done with the help of the following Lemmas I.7, I.8 and I.9, the proofs are available in Appendices I.1.5, I.1.6 and I.1.7, respectively.

**Lemma I.7.** *Under assumption $\mu_x\mu_y \leq \mu_{xy}^2$, the following inequality holds:*

$$q^{-n} \leq 2. \tag{121}$$

**Lemma I.8.** *The following inequalities hold:*

$$\omega_n \leq \frac{(n-2)\tilde{\delta}_x\mu_y}{\alpha^2},$$
$$\nu_n \geq \frac{1}{4(n-1)}. \tag{122}$$

**Lemma I.9.** *Constants $L^{++}$ and $\mu^{++}$ defined in equation (110) satisfy the following inequality:*

$$\frac{L^{++}}{\mu^{++}} \geq 1 + \max\left\{\frac{1}{86}, \frac{1}{55} \cdot \frac{\mu_{yx}^2}{\delta_x\delta_y}\right\}. \tag{123}$$

Using Lemma I.9, we can lower-bound $\rho$ as follows:

$$
\begin{aligned}
\rho &\stackrel{\text{(a)}}{=} 1 - \frac{2}{\sqrt{\frac{L^{++}}{\mu^{++}}} + 1} \\
&\stackrel{\text{(b)}}{\geq} \max\left\{ 1 - \sqrt{220} \cdot \sqrt{\frac{\delta_x \delta_y}{\mu_{yx}^2}},\ \frac{\sqrt{87} - \sqrt{86}}{\sqrt{87} + \sqrt{86}} \right\} \\
&\geq \max\left\{ 1 - \sqrt{220} \cdot \sqrt{\frac{\delta_x \delta_y}{\mu_{yx}^2}},\ \frac{1}{346} \right\} \\
&\stackrel{\text{(c)}}{=} \max\left\{ 1 - \frac{\sqrt{220}}{\sqrt{\kappa_{xy}}} \cdot \frac{L_{xy}}{\mu_{yx}},\ \frac{1}{346} \right\} \\
&\stackrel{\text{(d)}}{\geq} \max\left\{ 1 - 89 \cdot \frac{n}{\sqrt{\kappa_{xy}}},\ \frac{1}{346} \right\},
\end{aligned}
$$

where (a) uses the definition of $\rho$ in equation (117); (b) uses Lemma I.9; (c) uses the definition of $\kappa_{xy}$ in equation (6); (d) uses the definition of $n$ in equation (72), which concludes the proof. $\qquad\square$

### I.1.2. PROOF OF LEMMA I.1

Let matrices $\mathbf{W}_i, \mathbf{W}_i' \in \mathbb{R}^{i \times i}$ be defined for $i \in \{1, \ldots, n-1\}$ as follows:

$$
\mathbf{W}_i = \begin{bmatrix} 2 & -1 & & & \\ -1 & 2 & \ddots & & \\ & \ddots & \ddots & \ddots & \\ & & \ddots & 2 & -1 \\ & & & -1 & 2 \end{bmatrix}, \quad
\mathbf{W}_i' = \begin{bmatrix} 1 & -1 & & & \\ -1 & 2 & \ddots & & \\ & \ddots & \ddots & \ddots & \\ & & \ddots & 2 & -1 \\ & & & -1 & 2 \end{bmatrix}. \tag{124}
$$

Then using the definition of matrix $\mathbf{E}$ in equation (66), we can write the matrix $\mathbf{E}\mathbf{E}^\top$ as the following block matrix:

$$
\mathbf{E}\mathbf{E}^\top = \left[ \begin{array}{c|c|c}
n\gamma^2 & \beta\gamma \overset{n \text{ times}}{\cdots\cdots} \beta\gamma & \\ \hline
\begin{matrix} \beta\gamma \\ \vdots \\ \beta\gamma \end{matrix} & \beta^2(\mathbf{I}_n + \mathbf{J}_n) & \begin{matrix} -\alpha\beta \\ \vdots \\ -\alpha\beta \end{matrix} \\ \hline
& \begin{matrix} -\alpha\beta\cdots\cdots-\alpha\beta \end{matrix} & \begin{matrix} \alpha^2 \mathbf{W}_{n-1} \\[1em] \end{matrix} & \begin{matrix} \alpha\beta \overset{n \text{ times}}{\cdots\cdots} \alpha\beta \end{matrix} \\ \hline
& & \begin{matrix} \alpha\beta \\ \vdots \\ \alpha\beta \end{matrix} & \beta^2(\mathbf{I}_n + \mathbf{J}_n)
\end{array} \right]. \tag{125}
$$

Furthermore, let matrices $\mathbf{Q}_i, \mathbf{Q}_i' \in \mathbb{R}^{(n+i) \times (n+i)}$ be defined for $i \in \{1, \ldots, n\}$ as follows:

$$
\mathbf{Q}_i = \left[ \begin{array}{c|c} \alpha^2 \mathbf{W}_i & \alpha\beta \overset{n \text{ times}}{\cdots\cdots} \alpha\beta \\ \hline \begin{matrix} \alpha\beta \\ \vdots \\ \alpha\beta \end{matrix} & \beta^2(\mathbf{I}_n + \mathbf{J}_n) \end{array} \right], \quad
\mathbf{Q}_i' = \left[ \begin{array}{c|c} \alpha^2 \mathbf{W}_i' & \alpha\beta \overset{n \text{ times}}{\cdots\cdots} \alpha\beta \\ \hline \begin{matrix} \alpha\beta \\ \vdots \\ \alpha\beta \end{matrix} & \beta^2(\mathbf{I}_n + \mathbf{J}_n) \end{array} \right]. \tag{126}
$$

It is not hard to verify that the following matrix inequality holds:

$$
\mathbf{Q}_{n-1}' \succeq \beta^2 \left[ \begin{array}{c|c} \mathbf{O}_{n-1} & \\ \hline & \mathbf{I}_n \end{array} \right]. \tag{127}
$$

Moreover, we can show that for $i \in \{1, \ldots, n\}$, matrices $\mathbf{Q}_i, \mathbf{Q}_i'$ satisfy the following inequalities:

$$
\begin{aligned}
\mathbf{Q}_i &\frac{\alpha^2(n+i+1)}{n+i}(\mathbf{e}_1^{n+i})(\mathbf{e}_1^{n+i})^\top, \\
\mathbf{Q}_i' &\succeq \frac{\alpha^2}{n+i}(\mathbf{e}_1^{n+i})(\mathbf{e}_1^{n+i})^\top.
\end{aligned}
\tag{128}
$$

Indeed, let us prove equation (128) by induction. The base case $i = 1$ is trivial:

$$
\mathbf{Q}_1' = \begin{bmatrix} \alpha^2 & \alpha\beta \cdots \cdots \alpha\beta \\ \hline \alpha\beta & \\ \vdots & \beta^2(\mathbf{I}_n + \mathbf{J}_n) \\ \alpha\beta & \end{bmatrix} \succeq \begin{bmatrix} \frac{\alpha^2}{n+1} & \\ \hline & 0 \\ & \ddots \\ & & 0 \end{bmatrix} = \frac{\alpha^2}{n+1}(\mathbf{e}_1^{n+1})(\mathbf{e}_1^{n+1})^\top.
\tag{129}
$$

Furthermore, for an arbitrary index $i \in \{2, \ldots, n-1\}$, we can show that

$$
\mathbf{Q}_i' = \begin{bmatrix} \alpha^2 & \alpha^2 & 0 \cdots \cdots 0 \\ \hline \alpha^2 & \\ 0 & \\ \vdots & \mathbf{Q}_{i-1} \\ 0 & \end{bmatrix} \overset{(a)}{\succeq} \begin{bmatrix} \alpha^2 & \alpha^2 & 0 \cdots \cdots 0 \\ \hline \alpha^2 & \frac{\alpha^2(n+i)}{n+i-1} \\ 0 & & 0 \\ \vdots & & \ddots \\ 0 & & & 0 \end{bmatrix} \overset{(b)}{\succeq} \begin{bmatrix} \frac{\alpha^2}{n+i} & \\ \hline & 0 \\ & \ddots \\ & & 0 \end{bmatrix}.
$$

which is nothing else bu the induction hypothesis (128) for the index $i$, and where (a) uses the induction hypothesis (128) for the index $i-1$; (b) uses Young's inequality. Next, we can obtain the following inequality for the matrix $\mathbf{E}\mathbf{E}^\top$:

$$
\mathbf{E}\mathbf{E}^\top = \begin{bmatrix} n\gamma^2 & \beta\gamma \cdots \cdots \beta\gamma & \\ \hline \beta\gamma & & -\alpha\beta \\ \vdots & \beta^2(\mathbf{I}_n + \mathbf{J}_n) & \vdots \\ \beta\gamma & & -\alpha\beta \\ \hline -\alpha\beta \cdots \cdots -\alpha\beta & & \mathbf{Q}_{n-1} \end{bmatrix} \succeq \begin{bmatrix} n\gamma^2 & \beta\gamma \cdots \cdots \beta\gamma & \\ \hline \beta\gamma & & -\alpha\beta \\ \vdots & \frac{\beta^2(n+1)}{n}\mathbf{J}_n & \vdots \\ \beta\gamma & & -\alpha\beta \\ \hline -\alpha\beta \cdots \cdots -\alpha\beta & & \mathbf{Q}_{n-1} \end{bmatrix}
$$

$$
\succeq \begin{bmatrix} 0 & & \\ \hline & & -\alpha\beta \\ & \beta^2\mathbf{J}_n & \vdots \\ & & -\alpha\beta \\ \hline & -\alpha\beta \cdots \cdots -\alpha\beta & \mathbf{Q}_{n-1} \end{bmatrix} \succeq \begin{bmatrix} 0 & & \\ \hline & \mathbf{O}_n & \\ \hline & & \mathbf{Q}_{n-1}' \end{bmatrix}
$$

$$
\succeq \frac{\alpha^2}{n+i}(\mathbf{e}_{2n+1-i}^{3n})(\mathbf{e}_{2n+1-i}^{3n})^\top.
$$

This implies the following inequality for the matrix $\mathbf{E}\mathbf{E}^\top$:

$$
\begin{aligned}
\mathbf{E}\mathbf{E}^\top &\overset{(a)}{=} \sum_{i=1}^{n-1} \frac{2(n+i)}{3n(n-1)}\mathbf{E}\mathbf{E}^\top \succeq \sum_{i=1}^{n-1} \frac{2(n+i)}{3n(n-1)} \cdot \frac{\alpha^2}{n+i}(\mathbf{e}_{2n+1-i}^{3n})(\mathbf{e}_{2n+1-i}^{3n})^\top \\
&= \frac{2\alpha^2}{3n(n-1)} \begin{bmatrix} \mathbf{O}_{n+1} & & \\ \hline & \mathbf{I}_{n-1} & \\ \hline & & \mathbf{O}_n \end{bmatrix}.
\end{aligned}
$$

where (a) uses the facdt that $\sum_{i=1}^{n-1} \frac{2(n+i)}{3n(n-1)} = 1$. Furthermore, using equation (127), we can obtain the following inequality for the matrix $\mathbf{EE}^\top$:

$$
\mathbf{EE}^\top \succeq
\left[
\begin{array}{c|cc}
0 & & \\
\hline
 & \mathbf{O}_n & \\
 & & \mathbf{Q}'_{n-1}
\end{array}
\right]
\succeq \beta^2
\left[
\begin{array}{c|c}
\mathbf{O}_{2n} & \\
\hline
 & \mathbf{I}_n
\end{array}
\right].
$$

Next, we can obtain the following inequality for the matrix $\mathbf{EE}^\top$:

$$
\mathbf{EE}^\top =
\left[
\begin{array}{c|cc|c}
n\gamma^2 & \beta\gamma \cdots\cdots \beta\gamma & & \\
\hline
\beta\gamma & & & -\alpha\beta \\
\vdots & \beta^2(\mathbf{I}_n + \mathbf{J}_n) & & \vdots \\
\beta\gamma & & & -\alpha\beta \\
\hline
 & -\alpha\beta\cdots\cdots-\alpha\beta & & \\
 & & & \mathbf{Q}_{n-1}
\end{array}
\right]
$$

$$
\overset{(a)}{\succeq}
\left[
\begin{array}{c|c|c}
n\gamma^2 & \beta\gamma \cdots\cdots \beta\gamma & \\
\hline
\beta\gamma & & -\alpha\beta \\
\vdots & \beta^2(\mathbf{I}_n + \mathbf{J}_n) & \vdots \\
\beta\gamma & & -\alpha\beta \\
\hline
 & -\alpha\beta\cdots\cdots-\alpha\beta & \frac{2n\alpha^2}{2n-1} \\
 & & \mathbf{O}_{2n-2}
\end{array}
\right]
$$

$$
\overset{(b)}{\succeq}
\left[
\begin{array}{c|c|c}
0 & & \\
\hline
 & \beta^2\mathbf{I}_n + \beta^2\left(1 - \frac{1}{n} - \frac{2n-1}{2n}\right)\mathbf{J}_n & \\
\hline
 & & 0 \\
 & & \mathbf{O}_{2n-2}
\end{array}
\right]
$$

$$
=
\left[
\begin{array}{c|c|c}
0 & & \\
\hline
 & \beta^2\mathbf{I}_n - \frac{\beta^2}{2n}\mathbf{J}_n & \\
\hline
 & & 0 \\
 & & \mathbf{O}_{2n-2}
\end{array}
\right]
$$

$$
\overset{(c)}{\succeq}
\left[
\begin{array}{c|c|c}
0 & & \\
\hline
 & \frac{\beta^2}{2}\mathbf{I}_n & \\
\hline
 & & 0 \\
 & & \mathbf{O}_{2n-2}
\end{array}
\right].
$$

where (a) uses equation (128); (b) uses Young's inequality; (c) uses the fact that $\mathbf{J}_n \preceq n\mathbf{I}_n$. Furthermore, we can obtain the following inequality for the matrix $\mathbf{EE}^\top$:

$$
\mathbf{EE}^\top \overset{(a)}{\succeq}
\left[
\begin{array}{c|c|c}
n\gamma^2 & \beta\gamma \cdots\cdots \beta\gamma & \\
\hline
\beta\gamma & & -\alpha\beta \\
\vdots & \beta(\mathbf{I}_n + \mathbf{J}_n) & \vdots \\
\beta\gamma & & -\alpha\beta \\
\hline
 & -\alpha\beta\cdots\cdots-\alpha\beta & \frac{2n\alpha^2}{2n-1} \\
 & & \mathbf{O}_{2n-2}
\end{array}
\right]
$$

$$\overset{(b)}{\succeq} \begin{bmatrix} n\gamma^2 & \beta\gamma \cdots\cdots\cdots \beta\gamma & & \\ \begin{matrix}\beta\gamma \\ \vdots \\ \beta\gamma\end{matrix} & \frac{\beta^2(n+1)}{n}\mathbf{J}_n & \begin{matrix}-\alpha\beta \\ \vdots \\ -\alpha\beta\end{matrix} & \\ & -\alpha\beta\cdots\cdots\cdots-\alpha\beta & \frac{2n\alpha^2}{2n-1} & \\ & & & \mathbf{O}_{2n-2} \end{bmatrix}$$

$$\overset{(c)}{\succeq} \begin{bmatrix} n\gamma^2 & \beta\gamma\cdots\cdots\cdot\beta\gamma & & \\ \begin{matrix}\beta\gamma \\ \vdots \\ \beta\gamma\end{matrix} & \frac{3\beta^2}{2n}\mathbf{J}_n & & \\ & & 0 & \\ & & & \mathbf{O}_{2n-2} \end{bmatrix}$$

$$\overset{(d)}{\succeq} \frac{n\gamma^2}{3}\begin{bmatrix} 1 & & & \\ & \mathbf{O}_n & & \\ & & 0 & \\ & & & \mathbf{O}_{2n-2} \end{bmatrix}.$$

where (a) uses the ineqality obtained above; (b) uses the fact that $\mathbf{J}_n \preceq n\mathbf{I}_n$; (c) and (d) use Young's ineqality. Now, we sum all the inequalities for the matrix $\mathbf{EE}^\top$ obtained above with positive coefficients $\theta_1, \theta_2, \theta_3, \theta_4 > 0$ and obtain the following:

$$\mathbf{EE}^\top \succeq \begin{bmatrix} \theta_1 \cdot \frac{n\gamma^2}{3} & & & \\ & \theta_2 \cdot \frac{\beta^2}{2}\mathbf{I}_n & & \\ & & \theta_3 \cdot \frac{2\alpha^2}{3n^2}\mathbf{I}_{n-1} & \\ & & & \theta_4 \cdot \beta^2\mathbf{I}_n \end{bmatrix}. \tag{130}$$

Choosing $\theta_1 = \frac{3}{4}$, $\theta_2 = \frac{1}{18}$, $\theta_3 = \frac{1}{6}$, and $\theta_4 = \frac{1}{36}$ implies the following:

$$\mathbf{EE}^\top \succeq \min\left\{\frac{n\gamma^2}{4}, \frac{\beta^2}{36}, \frac{\alpha^2}{9n^2}\right\}\mathbf{I}_{3n}. \tag{131}$$

Finally, we obtain the following inequality for the matrix $\mathbf{EE}^\top$:

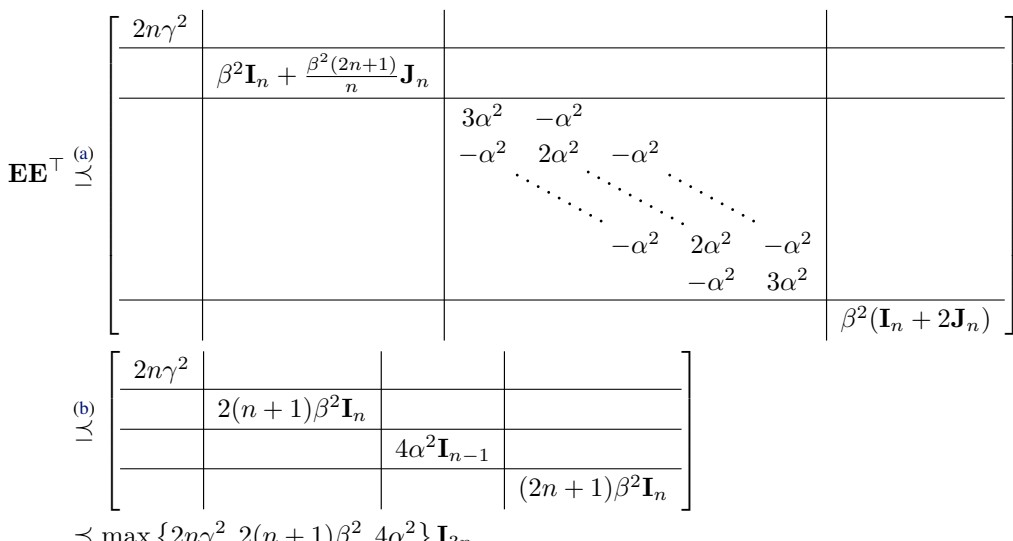

$$\mathbf{EE}^\top \overset{(a)}{\preceq} \begin{bmatrix} 2n\gamma^2 & & & \\ & \beta^2\mathbf{I}_n + \frac{\beta^2(2n+1)}{n}\mathbf{J}_n & & \\ & & \begin{matrix}3\alpha^2 & -\alpha^2 & & & \\ -\alpha^2 & 2\alpha^2 & -\alpha^2 & & \\ & \ddots & \ddots & \ddots & \\ & & -\alpha^2 & 2\alpha^2 & -\alpha^2 \\ & & & -\alpha^2 & 3\alpha^2\end{matrix} & \\ & & & \beta^2(\mathbf{I}_n + 2\mathbf{J}_n) \end{bmatrix}.$$

$$\overset{(b)}{\preceq} \begin{bmatrix} 2n\gamma^2 & & & \\ & 2(n+1)\beta^2\mathbf{I}_n & & \\ & & 4\alpha^2\mathbf{I}_{n-1} & \\ & & & (2n+1)\beta^2\mathbf{I}_n \end{bmatrix}$$

$$\preceq \max\left\{2n\gamma^2, 2(n+1)\beta^2, 4\alpha^2\right\}\mathbf{I}_{3n}.$$

where (a) uses the expression for the matrix $\mathbf{EE}^\top$ and Young's inequality; (b) uses Young's inequality, which concludes the proof in the case $\gamma > 0$. The remaining case $\gamma = 0$ is a trivial extension of the case $\gamma > 0$. $\qquad\square$

### I.1.3. PROOF OF LEMMA I.4

Using the definition of the Moreau envelope, we get

$$M_{\lambda h} = \underset{z' \in \mathbb{R}^d}{\arg\min} \frac{1}{2\lambda}\|z' - z\|^2 + \frac{\mu}{2}\|z'\|^2 + \frac{L - \mu}{2}\|\mathbf{F}z'\|^2 - \langle b, z' \rangle. \tag{132}$$

Using the first-order optimality conditions, we get

$$\frac{1}{\lambda}z + b = \left(\left(\frac{1}{\lambda} + \mu\right)\mathbf{I}_d + (L - \mu)\mathbf{F}\mathbf{F}^\top\right)z'. \tag{133}$$

Hence, we obtain the following:

$$
\begin{aligned}
M_{\lambda h} &\overset{(a)}{=} \frac{1}{2\lambda}\|z' - z\|^2 + \frac{\mu}{2}\|z'\|^2 + \frac{L - \mu}{2}\|\mathbf{F}z'\|^2 - \langle b, z' \rangle \\
&= \frac{1}{2\lambda}\|z\|^2 + \frac{1}{2}\|z'\|^2_{(\frac{1}{\lambda}+\mu)\mathbf{I}_d+(L-\mu)\mathbf{F}^\top\mathbf{F}} - \langle z', \tfrac{1}{\lambda}z + b \rangle \\
&\overset{(b)}{=} \frac{1}{2\lambda}\|z\|^2 - \frac{1}{2}\|\tfrac{1}{\lambda}z + b\|^2_{((\frac{1}{\lambda}+\mu)\mathbf{I}_d+(L-\mu)\mathbf{F}^\top\mathbf{F})^{-1}} \\
&\overset{(c)}{=} \frac{1}{2\lambda}\|z\|^2 - \frac{1}{2}\|\tfrac{1}{\lambda}z + b\|^2_{(\frac{1}{\lambda}+\mu)^{-1}\mathbf{I}_d-(\frac{1}{\lambda}+\mu)^{-2}\mathbf{F}^\top\left(\frac{1}{L-\mu}\mathbf{I}_p+(\frac{1}{\lambda}+\mu)^{-1}\mathbf{F}\mathbf{F}^\top\right)^{-1}\mathbf{F}} \\
&\overset{(d)}{=} \frac{1}{2\lambda}\|z\|^2 - \frac{1}{2}\|\tfrac{1}{\lambda}z + b\|^2_{(\frac{1}{\lambda}+\mu)^{-1}\mathbf{I}_d-(\frac{1}{\lambda}+\mu)^{-2}\mathbf{F}^\top\left(\frac{1}{L-\mu}\mathbf{I}_p+(\frac{1}{\lambda}+\mu)^{-1}\mathbf{I}_p\right)^{-1}\mathbf{F}} \\
&= \frac{1}{2\lambda}\|z\|^2 - \frac{1}{2}\|\tfrac{1}{\lambda}z + b\|^2_{\frac{\lambda}{1+\lambda\mu}\mathbf{I}_d-\frac{\lambda^2(L-\mu)}{(1+\lambda L)(1+\lambda\mu)}\mathbf{F}^\top\mathbf{F}} \\
&\overset{(e)}{=} \frac{1}{2\lambda}\|z\|^2 - \frac{1}{2}\|\tfrac{1}{\lambda}z + b\|^2_{(\lambda-\lambda^2\mu_\lambda)\mathbf{I}_d-\lambda^2(L_\lambda-\mu_\lambda)\mathbf{F}^\top\mathbf{F}} \\
&= \frac{\mu_\lambda}{2}\|z\|^2 + \frac{L_\lambda - \mu_\lambda}{2}\|\mathbf{F}z\|^2 - \langle z, ((1 - \lambda\mu_\lambda)\mathbf{I}_d - \lambda(L_\lambda - \mu_\lambda)\mathbf{F}^\top\mathbf{F})b \rangle \\
&\quad - \frac{1}{2}\|b\|^2_{(\lambda-\lambda^2\mu_\lambda)\mathbf{I}_d-\lambda^2(L_\lambda-\mu_\lambda)\mathbf{F}^\top\mathbf{F}} \\
&\overset{(f)}{=} \frac{\mu_\lambda}{2}\|z\|^2 + \frac{L_\lambda - \mu_\lambda}{2}\|\mathbf{F}z\|^2 - (1 - \lambda\mu_\lambda)\langle z, b \rangle - \frac{\lambda(1 - \lambda\mu_\lambda)}{2}\|b\|^2 \\
&\overset{(g)}{=} \frac{\mu_\lambda}{2}\|z\|^2 + \frac{L_\lambda - \mu_\lambda}{2}\|\mathbf{F}z\|^2 - B_\lambda\langle b, z \rangle - C_\lambda,
\end{aligned}
$$

where (a) uses the definition of the Moreau envelope; (b) uses the expression for $z'$; (c) uses the Woodbury matrix identity; (d) uses the assumption $\mathbf{F}\mathbf{F}^\top = \mathbf{I}_p$; (e) uses the definitions of $L_\lambda$ and $\mu_\lambda$; (f) uses the assumption $\mathbf{F}^\top\mathbf{F}b = 0$; (g) uses the definitions of $B_\lambda$ and $C_\lambda$, which concludes the proof. □

### I.1.4. PROOF OF LEMMA I.5

One can verify that vector $\hat{z}$ satisfies the following linear system:

$$\left(2\mu\mathbf{I}_d + (L - \mu)\mathbf{F}^\top\mathbf{F}\right)\hat{z} = B\mathbf{e}_1^d, \tag{134}$$

which can be rewritten as follows:

$$
\begin{bmatrix}
(L+3\mu) & -(L-\mu) & & & \\
-(L-\mu) & 2(L+\mu) & -(L-\mu) & & \\
& \ddots & \ddots & \ddots & \\
& & -(L-\mu) & 2(L+\mu) & -(L-\mu) \\
& & & -(L-\mu) & (L+3\mu)
\end{bmatrix}
\begin{bmatrix}
\hat{z}_1 \\
\hat{z}_2 \\
\vdots \\
\hat{z}_d
\end{bmatrix}
=
\begin{bmatrix}
2B \\
0 \\
\vdots \\
0
\end{bmatrix}. \tag{135}
$$

Let $z^\circ \in \mathbb{R}^d$ be defined as follows:

$$z^\circ = \frac{2B}{(1 - \rho)(L - \mu)} \cdot (\rho, \rho^2, \ldots, \rho^d). \tag{136}$$

Then one can observe that

$$\left(2\mu\mathbf{I}_d + (L - \mu)\mathbf{F}^\top\mathbf{F}\right) z^\circ = B\mathbf{e}_1^d - B\rho^d\mathbf{e}_d^d, \tag{137}$$

which implies

$$\left(2\mu\mathbf{I}_d + (L - \mu)\mathbf{F}^\top\mathbf{F}\right)(z^\circ - \hat{z}) = -B\rho^d\mathbf{e}_d^d. \tag{138}$$

Hence, we obtain the following inequality:

$$\|z^\circ - \hat{z}\| \leq \frac{B\rho^d}{2\mu}, \tag{139}$$

which concludes the proof. $\qquad\square$

### I.1.5. PROOF OF LEMMA I.7

We can upper-bound $q^{-n}$ as follows:

$$
\begin{aligned}
q^{-n} &= \exp\left(n\log\left(1 + \frac{1-q}{q}\right)\right) \\
&\overset{(a)}{\leq} \exp\left(\frac{n(1-q)}{q}\right) \\
&\overset{(b)}{\leq} \exp\left(\frac{L_{xy}}{6\mu_{yx}} \cdot \frac{1-q}{q}\right) \\
&\overset{(c)}{\leq} \exp\left(\frac{L_{xy}}{6\mu_{yx}} \cdot \frac{2\sqrt{\tilde{\delta}_x\mu_y}}{\sqrt{4\alpha^2 + \tilde{\delta}_x\mu_y} - \sqrt{\tilde{\delta}_x\mu_y}}\right) \\
&= \exp\left(\frac{L_{xy}}{6\mu_{yx}} \cdot \frac{2\sqrt{\tilde{\delta}_x\mu_y}\left(\sqrt{4\alpha^2 + \tilde{\delta}_x\mu_y} + \sqrt{\tilde{\delta}_x\mu_y}\right)}{4\alpha^2}\right) \\
&\overset{(d)}{\leq} \exp\left(\frac{L_{xy}}{6\mu_{yx}} \cdot \frac{4\sqrt{\tilde{\delta}_x\mu_y}\left(\alpha + \sqrt{\tilde{\delta}_x\mu_y}\right)}{4\alpha^2}\right) \\
&\overset{(e)}{\leq} \exp\left(\frac{L_{xy}}{6\mu_{yx}} \cdot \frac{\sqrt{\mu_x\mu_y + 4\mu_{xy}^2\frac{\mu_y}{L_y}}\left(\alpha + \sqrt{\mu_x\mu_y + 4\mu_{xy}^2\frac{\mu_y}{L_y}}\right)}{\alpha^2}\right) \\
&\overset{(f)}{\leq} \exp\left(\frac{L_{xy}}{6\mu_{yx}} \cdot \frac{\mu_{yx}\sqrt{1 + \frac{4\mu_y}{L_y}}\left(\alpha + \mu_{yx}\sqrt{1 + \frac{4\mu_y}{L_y}}\right)}{\alpha^2}\right) \\
&\overset{(g)}{\leq} \exp\left(\frac{L_{xy}}{6\mu_{yx}} \cdot \frac{\mu_{yx}\sqrt{2}\left(\alpha + \mu_{yx}\sqrt{2}\right)}{\alpha^2}\right) \\
&\overset{(h)}{\leq} \exp\left(\frac{\sqrt{2}}{3}\left(1 + \frac{2\sqrt{2}\mu_{yx}}{L_{xy}}\right)\right) \\
&\overset{(i)}{\leq} \exp\left(\frac{\sqrt{2}}{3} + \frac{2}{27}\right) \\
&\leq 2,
\end{aligned}
$$

where (a) uses the concavity of the logarithm; (b) uses the definition of $n$ in equation (72); (c) uses the definition of $q$ in equation (94); (d) uses the ineqality $\sqrt{a+b} \leq \sqrt{a} + \sqrt{b}$ for $a, b > 0$; (e) uses the definition of $\tilde{\delta}_x$ in equation (61); (f) uses the assumption $\mu_x\mu_y \leq \mu_{yx}^2$ and the assumption $\mu_{yx} > 0$, which, together with Assumption 2.5, implies $\mu_{xy} \leq \mu_{yx}$; (g) and (i) use Assumption 2.7; (h) uses the definition of $\alpha$ in equation (68), which concludes the proof. $\qquad\square$

### I.1.6. PROOF OF LEMMA I.8

We can upper-bound $\omega_n$ as follows:

$$
\begin{aligned}
\omega_n &\stackrel{(a)}{=} \frac{(1-q)(1-q^{n-2})}{(1+q^{n-1})} \\
&\stackrel{(b)}{\leq} (1-q)(1-q^{n-2}) = (1-q)^2 \sum_{j=0}^{n-3} q^j \\
&\stackrel{(c)}{\leq} (n-2)(1-q)^2 \\
&\stackrel{(d)}{=} \frac{4(n-2)\tilde{\delta}_x \mu_y}{\left(\sqrt{4\alpha^2 + \tilde{\delta}_x \mu_y} + \sqrt{\tilde{\delta}_x \mu_y}\right)^2} \\
&\leq \frac{(n-2)\tilde{\delta}_x \mu_y}{\alpha^2},
\end{aligned}
$$

where (a) uses the definition of $\omega_n$ in equation (96); (b) uses the fact that $q \geq 0$; (c) uses the fact that $q \leq 1$; (d) uses the definition of $q$ in equation (94). Next, we can upper-bound $\frac{1}{\nu_n}$ as follows:

$$
\frac{1}{\nu_n} \stackrel{(a)}{=} \frac{q(q^{1-n} - q^{n-1})}{(1-q)(1+q)} = \frac{q^{2-n}}{(1+q)} \sum_{j=0}^{2n-3} q^j \stackrel{(b)}{\leq} 2(n-1)q^{-n} \stackrel{(c)}{\leq} 4(n-1).
$$

where (a) uses the definition of $\nu_n$ in equation (96); (b) uses the fact that $0 \leq q \leq 1$; (c) uses Lemma I.7, which concludes the proof. $\qquad\square$

### I.1.7. PROOF OF LEMMA I.9

First, we can lower-bound $\frac{L^+}{\mu^+}$ as follows:

$$
\begin{aligned}
\frac{L^+}{\mu^+} &\stackrel{(a)}{=} \frac{\frac{\beta^2 L_x}{\beta^2 + L_x \mu_y} + \frac{\tilde{\delta}_x}{n} + \frac{\omega_n \alpha^2}{n\mu_y}}{\frac{\beta^2 \tilde{\delta}_x}{\beta^2 + \tilde{\delta}_x \mu_y} + \frac{\tilde{\delta}_x}{n} + \frac{\omega_n \alpha^2}{n\mu_y}} = 1 + \frac{\beta^4 (L_x - \tilde{\delta}_x)}{\left(\beta^2 \tilde{\delta}_x + (\beta^2 + \tilde{\delta}_x \mu_y)\left(\frac{\tilde{\delta}_x}{n} + \frac{\omega_n \alpha^2}{n\mu_y}\right)\right)(\beta^2 + L_x \mu_y)} \\
&\stackrel{(b)}{\geq} 1 + \frac{\beta^4 L_x}{2\left(\beta^2 \tilde{\delta}_x + (\beta^2 + \tilde{\delta}_x \mu_y)\left(\frac{\tilde{\delta}_x}{n} + \frac{\omega_n \alpha^2}{n\mu_y}\right)\right)(\beta^2 + L_x \mu_y)} \\
&\stackrel{(c)}{\geq} 1 + \frac{\beta^4 L_x}{2\tilde{\delta}_x \left(\beta^2 + \frac{(n-1)}{n}(\beta^2 + \tilde{\delta}_x \mu_y)\right)(\beta^2 + L_x \mu_y)} \\
&\stackrel{(d)}{=} 1 + \frac{\beta^4 L_x}{2\tilde{\delta}_x \left(\beta^2 + \frac{(n-1)}{n}\left(\beta^2 + \mu_x \mu_y + \mu_{xy}^2 \frac{4\mu_y}{L_y}\right)\right)(\beta^2 + L_x \mu_y)} \\
&\stackrel{(e)}{\geq} 1 + \frac{\beta^4 L_x}{2\tilde{\delta}_x \left(\beta^2 + \frac{(n-1)}{n}\left(\beta^2 + \mu_{yx}^2(1 + \frac{4\mu_y}{L_y})\right)\right)(\beta^2 + L_x \mu_y)} \\
&\stackrel{(f)}{\geq} 1 + \frac{\beta^4 L_x}{2\tilde{\delta}_x \left(\beta^2 + \frac{(n-1)}{n}\left(\beta^2 + 2\mu_{yx}^2\right)\right)(\beta^2 + L_x \mu_y)} \\
&\stackrel{(g)}{=} 1 + \frac{\beta^4 L_x}{2\tilde{\delta}_x \beta^4 \left(1 + \frac{(n-1)}{n}\left(1 + \frac{2n^2 \mu_{yx}^2}{L_{xy}^2}\right)\right)\left(1 + \frac{n^2 L_x \mu_y}{L_{xy}^2}\right)} \\
&\stackrel{(h)}{\geq} 1 + \frac{L_x}{2\tilde{\delta}_x \left(1 + \frac{(n-1)}{n}\left(1 + \frac{1}{18}\right)\right)\left(1 + \frac{L_x \mu_y}{36\mu_{yx}^2}\right)} \\
&\geq 1 + \frac{9 L_x \mu_{yx}^2}{37\tilde{\delta}_x \left(\mu_{yx}^2 + \frac{1}{36}L_x \mu_y\right)} = \frac{\frac{9}{37}L_x \mu_{yx}^2 + \tilde{\delta}_x \left(\mu_{yx}^2 + \frac{1}{36}L_x \mu_y\right)}{\tilde{\delta}_x \left(\mu_{yx}^2 + \frac{1}{36}L_x \mu_y\right)},
\end{aligned}
$$

where (a) uses the definition of $L^+$ and $\mu^+$ in equation (109); (b) uses equation (69); (c) uses Lemma I.8; (d) uses the definition of $\tilde{\delta}_x$ in equation (61); (e) uses the assumption $\mu_x\mu_y \leq \mu_{yx}^2$ and the assumption $\mu_{yx} > 0$, which, together with Assumption 2.5, implies $\mu_{xy} \leq \mu_{yx}$; (f) uses Assumption 2.7; (g) uses the definition of $\beta$ in equation (68); (h) uses the definition of $n$ in equation (72). Furthermore, we can lower-bound $\frac{L^+}{\mu^+}$ as follows:

$$\frac{L^+}{\mu^+} \geq 1 + \frac{9L_x\mu_{yx}^2}{37\tilde{\delta}_x\left(\mu_{yx}^2 + \frac{1}{36}L_x\mu_y\right)} = 1 + \frac{9\mu_{yx}^2}{37\left(\frac{\tilde{\delta}_x\mu_{yx}^2}{L_x} + \frac{1}{36}\tilde{\delta}_x\mu_y\right)}$$

$$\overset{(a)}{\geq} 1 + \frac{9\mu_{yx}^2}{37\left(\frac{1}{2}\mu_{yx}^2 + \frac{1}{36}\left(\mu_x\mu_y + \frac{4\mu_y\mu_{xy}^2}{L_y}\right)\right)}$$

$$\overset{(b)}{\geq} 1 + \frac{9\mu_{xy}^2}{37\left(\frac{1}{2}\mu_{yx}^2 + \frac{1}{36}\mu_{yx}^2\left(1 + \frac{4\mu_y}{L_y}\right)\right)}$$

$$\overset{(c)}{\geq} 1 + \frac{9\mu_{yx}^2}{37\left(\frac{1}{2}\mu_{yx}^2 + \frac{1}{18}\mu_{yx}^2\right)} = 1 + \frac{81}{185}$$

$$\geq \frac{10}{7},$$

where (a) uses equation (69) and the definition of $\tilde{\delta}_x$ in equation (61); (b) uses the assumption $\mu_x\mu_y \leq \mu_{yx}^2$ and the assumption $\mu_{yx} > 0$, which, together with Assumption 2.5, implies $\mu_{xy} \leq \mu_{yx}$; (c) uses Assumption 2.7. Next, we can lower-bound $\frac{L^{++}}{\mu^{++}}$ as follows:

$$\frac{L^{++}}{\mu^{++}} \overset{(a)}{=} \frac{\frac{2\nu_n\alpha^2 L^+}{2\nu_n\alpha^2 + n\mu_y L^+}}{\frac{2\nu_n\alpha^2\mu^+}{2\nu_n\alpha^2 + n\mu_y\mu^+}} = 1 + \frac{(L^+ - \mu^+)}{\mu^+\left(1 + \frac{n\mu_y L^+}{2\nu_n\alpha^2}\right)}$$

$$\overset{(b)}{\geq} 1 + \frac{3L^+}{10\mu^+\left(1 + \frac{n\mu_y L^+}{2\nu_n\alpha^2}\right)}$$

$$\overset{(c)}{\geq} 1 + \frac{3L^+}{10\mu^+\left(1 + \frac{2n(n-1)\mu_y L^+}{\alpha^2}\right)}$$

$$\overset{(d)}{\geq} 1 + \frac{3L^+}{10\mu^+\left(1 + \frac{8n(n-1)\mu_y L^+}{L_{xy}^2}\right)}$$

$$\overset{(e)}{\geq} 1 + \frac{3L^+}{10\mu^+\left(1 + \frac{2\mu_y L^+}{9\mu_{yx}^2}\right)} = 1 + \frac{3\mu_{yx}^2}{10\left(\frac{\mu^+}{L^+}\mu_{yx}^2 + \frac{2}{9}\mu_y\mu^+\right)}$$

$$\overset{(f)}{\geq} 1 + \frac{3\mu_{yx}^2}{10\left(\frac{\mu^+}{L^+}\mu_{yx}^2 + \frac{2}{9}\mu_y\left(\frac{\beta^2\tilde{\delta}_x}{\tilde{\delta}_x\mu_y+\beta^2} + \frac{\tilde{\delta}_x}{n} + \frac{\omega_n\alpha^2}{n\mu_y}\right)\right)}$$

$$\leq 1 + \frac{3\mu_{yx}^2}{10\left(\frac{\mu^+}{L^+}\mu_{yx}^2 + \frac{2}{9}\mu_y\left(\frac{(n+1)\tilde{\delta}_x}{n} + \frac{\omega_n\alpha^2}{n\mu_y}\right)\right)}$$

$$\overset{(g)}{\geq} 1 + \frac{3\mu_{yx}^2}{10\left(\frac{\mu^+}{L^+}\mu_{yx}^2 + \frac{2}{9}\mu_y\left(\frac{(n+1)\tilde{\delta}_x}{n} + \frac{(n-2)\tilde{\delta}_x}{n}\right)\right)} \geq 1 + \frac{3\mu_{yx}^2}{10\left(\frac{\mu^+}{L^+}\mu_{yx}^2 + \frac{4}{9}\mu_y\tilde{\delta}_x\right)}$$

$$\overset{(h)}{\geq} 1 + \frac{3\mu_{yx}^2}{10\left(\frac{\tilde{\delta}_x\mu_{yx}^2\left(\mu_{yx}^2 + \frac{1}{36}L_x\mu_y\right)}{\frac{9}{37}L_x\mu_{yx}^2 + \tilde{\delta}_x\left(\mu_{yx}^2 + \frac{1}{36}L_x\mu_y\right)} + \frac{4}{9}\mu_y\tilde{\delta}_x\right)}$$

$$\geq 1 + \frac{3\mu_{yx}^2}{10\left(\frac{37\tilde{\delta}_x\left(\mu_{yx}^2 + \frac{1}{36}L_x\mu_y\right)}{9L_x} + \frac{4}{9}\mu_y\tilde{\delta}_x\right)} \geq 1 + \frac{27\mu_{yx}^2}{370\tilde{\delta}_x\left(\mu_y + \frac{\mu_{yx}^2}{L_x}\right)}$$

$$\overset{(i)}{=} 1 + \frac{27\mu_{yx}^2}{370\tilde{\delta}_x\delta_y} \overset{(j)}{\geq} 1 + \frac{27\mu_{yx}^2}{1480\delta_x\delta_y}$$

$$\geq 1 + \frac{\mu_{yx}^2}{55\delta_x\delta_y},$$

where (a) uses the definition of $L^{++}$ and $\mu^{++}$ in equation (110); (b) uses the previously obtained inequality $\mu^+ \leq \frac{7}{10}L^+$; (c) and (g) use Lemma I.8; (d) uses the definition of $\alpha$ in equation (68); (e) uses the definition of $n$ in equation (72); (f) uses the definition of $\mu^+$ in equation (109); (h) uses the previously obtained lower bound on $\frac{L^+}{\mu^+}$; (i) uses the definition of $\delta_y$ in equation (4); (j) uses the definition of $\tilde{\delta}_x$ in equation (61) and the definition of $\delta_x$ in equation (4). Furthermore, we can lower-bound $\frac{L^{++}}{\mu^{++}}$ as follows:

$$\frac{L^{++}}{\mu^{++}} \geq 1 + \frac{\mu_{yx}^2}{55\delta_x\delta_y} \overset{(a)}{\geq} 1 + \frac{\mu_{yx}^2}{55\left(\mu_x\mu_y + \frac{\mu_y}{L_y}\mu_{xy}^2 + \frac{\mu_x}{L_x}\mu_{yx}^2 + \frac{\mu_{xy}^2\mu_{yx}^2}{L_xL_y}\right)}$$

$$\overset{(b)}{\geq} 1 + \frac{1}{55\left(1 + \frac{\mu_y}{L_y} + \frac{\mu_x}{L_x} + \frac{\mu_{xy}^2}{L_xL_y}\right)} \overset{(c)}{\geq} 1 + \frac{16}{25\cdot 55} \geq 1 + \frac{1}{86},$$

where (a) uses the definitions of $\delta_x$ and $\delta_y$ in equation (4); (b) uses the assumption $\mu_x\mu_y \leq \mu_{yx}^2$ and the assumption $\mu_{yx} > 0$, which, together with Assumption 2.5, implies $\mu_{xy} \leq \mu_{yx}$; (c) uses Assumption 2.7, which concludes the proof. $\qquad\square$

### I.1.8. PROOF OF LEMMA I.6

First, we can upper-bound $\|x' - x^*\|^2$ as follows:

$$\|x' - x^*\|^2 \overset{(a)}{=} \sum_{i=1}^{n}\|\text{prox}_{\frac{\mu_y}{\beta^2}h_1}(x'_{n+1}) - \text{prox}_{\frac{\mu_y}{\beta^2}h_1}(x^*_{n+1})\|^2$$

$$+ \|\text{prox}_{\frac{n\mu_y}{2\nu_n\alpha^2}h_1^+}(z^\circ) - \text{prox}_{\frac{n\mu_y}{2\nu_n\alpha^2}h_1^+}(z^*)\|^2$$

$$+ \sum_{i=n+2}^{2n-1}\left\|\frac{(x'_{n+1} - x^*_{n+1})(q^{2n-i} - q^{i-2n}) + (x'_{2n} - x^*_{2n})(q^{i-(n+1)} - q^{(n+1)-i})}{(q^{n-1} - q^{1-n})}\right\|^2$$

$$+ \|\text{prox}_{\frac{n\mu_y}{2\nu_n\alpha^2}h_2^+}(z^\circ) - \text{prox}_{\frac{n\mu_y}{2\nu_n\alpha^2}h_2^+}(z^*)\|^2$$

$$+ \sum_{i=2n+1}^{3n}\|\text{prox}_{\frac{\mu_y}{\beta^2}h_2}(x'_{2n}) - \text{prox}_{\frac{\mu_y}{\beta^2}h_2}(x^*_{2n})\|^2$$

$$\overset{(b)}{\leq} n\|x'_{n+1} - x^*_{n+1}\|^2 + \|z^\circ - z^*\|^2$$

$$+ \sum_{i=n+2}^{2n-1}\left\|\frac{(x'_{n+1} - x^*_{n+1})(q^{2n-i} - q^{i-2n}) + (x'_{2n} - x^*_{2n})(q^{i-(n+1)} - q^{(n+1)-i})}{(q^{n-1} - q^{1-n})}\right\|^2$$

$$+ \|z^\circ - z^*\|^2 + n\|x'_{2n} - x^*_{2n}\|^2$$

$$\overset{(c)}{\leq} 2\|z^\circ - z^*\|^2 + n(\|x'_{n+1} - x^*_{n+1}\|^2 + \|x'_{2n} - x^*_{2n}\|^2)$$

$$+ 2\sum_{i=n+2}^{2n-1}\frac{(q^{2n-i} - q^{i-2n})^2}{(q^{n-1} - q^{1-n})^2}\|x'_{n+1} - x^*_{n+1}\|^2$$

$$+ 2\sum_{i=n+2}^{2n-1}\frac{(q^{i-(n+1)} - q^{(n+1)-i})^2}{(q^{n-1} - q^{1-n})^2}\|x'_{2n} - x^*_{2n}\|^2$$

$$= 2\|z^\circ - z^*\|^2 + \left(n + 2\sum_{i=1}^{n-2}\frac{(q^i - q^{-i})^2}{(q^{n-1} - q^{1-n})^2}\right)(\|x'_{n+1} - x^*_{n+1}\|^2 + \|x'_{2n} - x^*_{2n}\|^2)$$

$$\overset{(d)}{=} 2\|z^\circ - z^*\|^2$$

$$+ \left( n + 2 \sum_{i=1}^{n-2} \frac{(q^i - q^{-i})^2}{(q^{n-1} - q^{1-n})^2} \right) \left\| \mathrm{prox}_{\frac{n\mu_y}{2\nu_n\alpha^2} h_1^+}(z^\circ) - \mathrm{prox}_{\frac{n\mu_y}{2\nu_n\alpha^2} h_1^+}(z^*) \right\|^2$$

$$+ \left( n + 2 \sum_{i=1}^{n-2} \frac{(q^i - q^{-i})^2}{(q^{n-1} - q^{1-n})^2} \right) \left\| \mathrm{prox}_{\frac{n\mu_y}{2\nu_n\alpha^2} h_2^+}(z^\circ) - \mathrm{prox}_{\frac{n\mu_y}{2\nu_n\alpha^2} h_2^+}(z^*) \right\|^2$$

$$\overset{(e)}{\le} \left( 2 + 2n + 4 \sum_{i=1}^{n-2} \frac{(q^i - q^{-i})^2}{(q^{n-1} - q^{1-n})^2} \right) \| z^\circ - z^* \|^2$$

$$\overset{(f)}{\le} \left( 2 + 2n + 4 \sum_{i=1}^{n-2} \frac{(q^i - q^{-i})^2}{(q^{n-1} - q^{1-n})^2} \right) \left( \frac{A^{++} \rho^d}{2\mu^{++}} \right)^2$$

$$\overset{(g)}{=} C'_\pi A^2 \rho^{2d},$$

where (a) uses the expressions for $x^*$ in equations (91), (95) and (102) and the definition of $x'$ in equation (119); (b) and (e) use the nonexpansiveness of the proximal operator; (c) uses Young's inequality; (d) uses the definitions of $x^*_{n+1}$ and $x^*_{2n}$ in equation (102) and the definitions of $x'_{n+1}$ and $x'_{2n}$ in equation (119); (f) uses equation (116); (g) uses the definition of $A^{++}$ in equation (111), and $C'_\pi \ge 0$ is a constant that depends on the parameters $\pi \in \Pi$.

Next, we can upper-bound $\mathcal{R}^2_{\delta_x \delta_y}(x', y')$ as follows:

$$\mathcal{R}^2_{\delta_x \delta_y}(x', y') \overset{(a)}{=} \delta_x \| x' - x^* \|^2 + \delta_y \| y' - y^* \|^2$$

$$\overset{(b)}{=} \delta_x \| x' - x^* \|^2 + \delta_y \| \nabla g^*(\mathbf{B}x') - \nabla g^*(\mathbf{B}x^*) \|^2$$

$$\overset{(c)}{\le} \delta_x \| x' - x^* \|^2 + \frac{\delta_y}{\mu_y^2} \| \mathbf{B}(x' - x^*) \|^2$$

$$\overset{(d)}{\le} \delta_x \| x' - x^* \|^2 + \frac{\delta_y L_{xy}^2}{\mu_y^2} \| x' - x^* \|^2$$

$$\overset{(e)}{\le} \left( \delta_x + \frac{\delta_y L_{xy}^2}{\mu_y^2} \right) C'_\pi A^2 \rho^{2d}$$

$$= C''_\pi A^2 \rho^{2d},$$

where (a) uses the definition of $\mathcal{R}^2_{\delta_x \delta_y}$ in equation (9); (b) uses the definition of $y'$ in equation (119) and the expression for $y^*$ in equation (85); (c) uses the $(1/\mu_y)$-smoothness of function $g^*(y)$; (d) uses Assumption 2.5; (e) uses the previously obtained upper bound on $\| x' - x^* \|^2$, and $C''_\pi \ge 0$ is a constant that depends on the parameters $\pi \in \Pi$.

Furthermore, we can express $x'_{2n}$ as follows:

$$x'_{2n} \overset{(a)}{=} \mathrm{prox}_{\frac{n\mu_y}{2\nu_n\alpha^2} h_2^+}(z^\circ)$$

$$\overset{(b)}{=} z^\circ - \frac{n\mu_y}{2\nu_n\alpha^2} \nabla h_2^{++}(z^\circ)$$

$$\overset{(c)}{=} \left( \mathbf{I}_d - \frac{n\mu_y}{2\nu_n\alpha^2} \left( \mu^{++} \mathbf{I}_d + (L^{++} - \mu^{++}) \mathbf{F}_2^\top \mathbf{F}_2 \right) \right) z^\circ + A^{++} \mathbf{e}_1^d$$

$$= \left( 1 - \frac{n\mu_y \mu^{++}}{2\nu_n\alpha^2} \right) z^\circ - \frac{n\mu_y (L^{++} - \mu^{++})}{2\nu_n\alpha^2} \mathbf{F}_2^\top \mathbf{F}_2 z^\circ + A^{++} \mathbf{e}_1^d$$

$$\overset{(d)}{=} \frac{2A^{++}}{(1-\rho)(L^{++} - \mu^{++})} \left( \left( 1 - \frac{n\mu_y \mu^{++}}{2\nu_n\alpha^2} \right) \mathbf{I}_d - \frac{n\mu_y (L^{++} - \mu^{++})}{2\nu_n\alpha^2} \mathbf{F}_2^\top \mathbf{F}_2 \right) (\rho, \dots, \rho^d)$$
$$+ A^{++} \mathbf{e}_1^d$$

$$\overset{(e)}{=} A( C'_{\pi,2n} \mathbf{I}_d + 2 C''_{\pi,2n} \mathbf{F}_2^\top \mathbf{F}_2 )(\rho, \dots, \rho^d) + A C'''_{\pi,2n} \mathbf{e}_1^d$$

$$\overset{(f)}{=} A C'''_{\pi,2n} \mathbf{e}_1^d + A C'_{\pi,2n}(\rho, \dots, \rho^d)$$
$$+ A C''_{\pi,2n}(0, \rho^2(1-\rho), \rho^2(\rho-1), \rho^4(1-\rho), \rho^4(\rho-1), \dots)$$

$$= A(C'''_{\pi,2n} + \rho C'_{\pi,2n})(1, 0, 0, \ldots)$$
$$+ A(\rho^2 C'_{\pi,2n} + \rho^2(1-\rho)C''_{\pi,2n})(0, 1, 0, \rho^2, 0, \rho^4, \ldots)$$
$$+ A(\rho^3 C'_{\pi,2n} + \rho^2(\rho-1)C''_{\pi,2n})(0, 0, 1, 0, \rho^2, 0, \rho^4, \ldots),$$

where (a) uses the definition of $x'$ on equation (119); (b) uses the properties of the proximal operator and the definition of $h_2^{++}(z)$ in equation (100); (c) uses the definition of $h_2^{++}(z)$ in equation (108); (d) uses the adapted version of the definition of $z^\circ$ in equation (136); (e) uses the definition of $A^{++}$ in equation (111); (f) uses the definition of matrix $\mathbf{F}_2$ in equation (62), and $C'_{\pi,2n}, C''_{\pi,2n} \in \mathbb{R}$ are constants that depend on the parameters $\pi \in \Pi$. From this expression, we can conclude that the following inequality holds:

$$\|x'_{2n} - x^\circ_{2n}\|^2 \le A^2 C_{\pi,2n} \rho^{2d}, \tag{140}$$

where $C_{\pi,2n} \ge 0$ is some constant that depends on the parameters $\pi \in \Pi$. Similarly, we can obtain the following inequality:

$$\delta_x \|x' - x^\circ\|^2 + \delta_y \|y' - y^\circ\|^2 \le C'''_\pi A^2 \rho^{2d}, \tag{141}$$

where $C'''_\pi \ge 0$ is some constant that depends on the parameters $\pi \in \Pi$. Therefore, we obtain the following inequality:

$$\begin{aligned}
\mathcal{R}^2_{\delta_x \delta_y}(x^\circ, y^\circ) &\le 2\mathcal{R}^2_{\delta_x \delta_y}(x', y') + 2\delta_x \|x' - x^\circ\|^2 + 2\delta_y \|y' - y^\circ\|^2 \\
&\le 2(C'''_\pi + C''_\pi) A^2 \rho^{2d} \\
&= C_\pi A^2 \rho^{2d},
\end{aligned}$$

which concludes the proof. $\qquad\square$

### I.1.9. PROOF OF LEMMA I.3

Functions $f(x)$ and $g(y)$ defined in equations (59) and (63) are quadratic. Hence, the optimality conditions (7) can be written in the following form:

$$\begin{aligned}
\mathbf{G}_x x^* + \mathbf{B}^\top y^* &= Ab \\
\mathbf{G}_y y^* - \mathbf{B} x^* &= 0,
\end{aligned} \tag{142}$$

where $b = (\mathbf{0}_{2n}, \mathbf{1}_n) \otimes \mathbf{e}_1^d \in \mathbb{R}^{n_x d}$, and $\mathbf{G}_x \in \mathbb{R}^{n_x d \times n_x d}$ and $\mathbf{G}_y \in \mathbb{R}^{n_y d \times n_y d}$ are symmetric matrices that can be expressed from $d \in \{1, 2, \ldots\}$ and the parameters $\pi \in \Pi$ using the definitions of functions $f(x)$ and $g(y)$ in equations (59) and (63). Hence, the solution $(x^*, y^*)$ can be expressed as follows:

$$\begin{aligned}
x^* &= A \cdot \mathbf{Q}_x^{-1} b, \\
y^* &= A \cdot G_y^{-1} \mathbf{B} \mathbf{Q}_x^{-1} b,
\end{aligned} \tag{143}$$

where $\mathbf{Q}_x \in \mathbb{R}^{n_x d \times n_x d}$ is defined as follows:

$$\mathbf{Q}_x = \mathbf{G}_x + \mathbf{B}^\top \mathbf{G}_y^{-1} \mathbf{B}. \tag{144}$$

Note that matrix $\mathbf{G}_x + \mathbf{B}^\top \mathbf{G}_y^{-1} \mathbf{B}$ is invertible due to Assumptions 2.3, 2.4, 2.5 and 2.6:

$$\mathbf{G}_x + \mathbf{B}^\top \mathbf{G}_y^{-1} \mathbf{B} \succeq \delta_x \mathbf{I}_{n_x d} \succ \mathbf{O}_{n_x d}. \tag{145}$$

Furthermore, using the definition of $(x^0, y^0)$ in equation (78), we obtain the following:

$$\begin{aligned}
(x^* - x^0, y^* - y^0) &= (\mathbf{I}_{n_x + n_y} \otimes (\mathbf{I}_d - \mathbf{P}))(x^*, y^*) \\
&= (\mathbf{I}_{n_x + n_y} \otimes (\mathbf{I}_d - \mathbf{P}))(\mathbf{Q}_x^{-1} b, G_y^{-1} \mathbf{B} \mathbf{Q}_x^{-1} b) \cdot A.
\end{aligned}$$

Hence, it is easy to observe that $\mathcal{R}^2_{\delta_x \delta_y}(x^0, y^0) = B_{\pi,d} A^2$ for some constant $B_{\pi,d} \ge 0$ that depends on $d \in \{2, 3, \ldots\}$ and the parameters $\pi \in \Pi$. Finally, we need to show the existence of $\bar{d} \in \{2, 3, \ldots\}$ such that for all $A > 0$ the following inequality holds:

$$\min_{d \in \{\hat{d}, \hat{d}+1, \ldots\}} B_{\pi,d} = \min_{d \in \{\hat{d}, \hat{d}+1, \ldots\}} \frac{\mathcal{R}^2_{\delta_x \delta_y}(x^0, y^0)}{A^2} > 0. \tag{146}$$

To do this, we can lower-bound $\mathcal{R}^2_{\delta_x \delta_y}(x^0, y^0)$ as follows:

$$
\begin{aligned}
\mathcal{R}^2_{\delta_x \delta_y}(x^0, y^0) &\geq \delta_x \left( \|x^0_{n+1} - x^*_{n+1}\|^2 + \|x^0_{2n} - x^*_{2n}\|^2 \right) \\
&\overset{(a)}{=} \delta_x \left( \|(\mathbf{I}_d - \mathbf{P})x^*_{n+1}\|^2 + \|(\mathbf{I}_d - \mathbf{P})x^*_{2n}\|^2 \right) \\
&\overset{(b)}{\geq} \tfrac{1}{2}\delta_x \|(\mathbf{I}_d - \mathbf{P})(x^*_{n+1} + x^*_{2n})\|^2 \\
&\overset{(c)}{=} \tfrac{1}{2}\delta_x \|(\mathbf{I}_d - \mathbf{P})(\mathrm{prox}_{\frac{n\mu_y}{2\nu_n \alpha^2} h_1^+}(z^*) + \mathrm{prox}_{\frac{n\mu_y}{2\nu_n \alpha^2} h_2^+}(z^*))\|^2 \\
&\overset{(d)}{=} \tfrac{1}{2}\delta_x \|(\mathbf{I}_d - \mathbf{P})(2z^* - \tfrac{n\mu_y}{2\nu_n \alpha^2}(\nabla h_1^{++}(z^*) + \nabla h_2^{++}(z^*)))\|^2 \\
&\overset{(e)}{=} 2\delta_x \|(\mathbf{I}_d - \mathbf{P})z^*\|^2 \\
&\overset{(f)}{\geq} \delta_x \|(\mathbf{I}_d - \mathbf{P})z^\circ\|^2 - 2\delta_x \|(\mathbf{I}_d - \mathbf{P})(z^* - z^\circ)\|^2 \\
&\overset{(g)}{\geq} \delta_x \|(\mathbf{I}_d - \mathbf{P})z^\circ\|^2 - 2\delta_x \left( \frac{A^{++} \rho^d}{2\mu^{++}} \right)^2 \\
&\overset{(h)}{\geq} \delta_x \left( \frac{2A^{++}\rho^2}{(1-\rho)(L^{++} - \mu^{++})} \right)^2 - 2\delta_x \left( \frac{A^{++}\rho^d}{2\mu^{++}} \right)^2 \\
&\geq (A^{++})^2 \left( \frac{2\delta_x \rho^4}{(L^{++})^2} - \frac{\delta_x \rho^{2d}}{2(\mu^{++})^2} \right) \\
&\overset{(i)}{\geq} A^2 \cdot \left( \frac{2\nu_n \alpha^2 \beta^2}{(2\nu_n \alpha^2 + n\mu_y \mu^+)(\beta^2 + \mu_y \tilde{\delta}_x)} \right)^2 \cdot \left( \frac{2\delta_x \rho^4}{(L^{++})^2} - \frac{\delta_x \rho^{2d}}{2(\mu^{++})^2} \right),
\end{aligned}
$$

where (a) uses the definition of $x^0$ in equation (78); (b) uses the convexity of $\|\cdot\|^2$; (c) uses the expressions for $x^*_{n+1}$ and $x^*_{2n}$ in equation (102); (d) uses the properties of the proximal operator and the definition of $h_j^{++}(z)$ in equation (100); (e) uses the definition of $z^*$ in equation (101); (f) uses Young's inequality; (g) uses equation (116); (h) uses the adapted version of the definition of $z^\circ$ in equation (136) in the proof of Lemma I.5 in Appendix I.1.4; (i) uses the definition of $A^{++}$ in equation (111). It is not hard to verify that $\left( \frac{2\nu_n \alpha^2 \beta^2}{(2\nu_n \alpha^2 + n\mu_y \mu^+)(\beta^2 + \mu_y \tilde{\delta}_x)} \right)^2$ and $\frac{2\delta_x \rho^4}{(L^{++})^2}$ are positive constants that do not depend on $d$. Hence, we can easily obtain the following relation:

$$
\liminf_{d \to +\infty} \frac{\mathcal{R}^2_{\delta_x \delta_y}(x^0, y^0)}{A^2} > 0, \tag{147}
$$

which concludes the proof. $\qquad\square$

# J. Proof of Lemma G.3

Let $\bar{\mu}_{xy} = \max\{\mu_{xy}, \mu_{yx}\}$. We consider a special instance of problem (1), where $\mathcal{X} = \mathcal{Y} = \mathbb{R}^d$, functions $f(x)$ and $g(y)$ are defined as follows:

$$f(x) = \frac{\mu_x}{2}\|x\|^2 - A\langle \mathbf{e}_1^d, x\rangle, \quad g(y) = \frac{\mu_y}{2}\|y\|^2, \tag{148}$$

and matrix $\mathbf{B} \in \mathbb{R}^{d\times d}$ is defined as follows:

$$\mathbf{B} = \frac{1}{2}\begin{bmatrix} \alpha & -\beta & & & \\ & \ddots & \ddots & & \\ & & \ddots & \ddots & \\ & & & \alpha & -\beta \\ & & & & \alpha \end{bmatrix}, \quad \text{where} \quad \alpha = L_{xy} + \bar{\mu}_{xy}, \quad \beta = L_{xy} - \bar{\mu}_{xy}. \tag{149}$$

Functions $f(x)$ and $g(y)$ obviously satisfy Assumptions 2.3 and 2.4. Moreover, matrix $\mathbf{B}$ satisfies Assumption 2.5 due to the following Lemma J.1, the proof is available in Appendix J.1.1.

**Lemma J.1.** *The singular values of matrix $\mathbf{B}$ defined in equation (149) satisfy the following inequalities:*

$$\bar{\mu}_{xy} \le \sigma_{\min}(\mathbf{B}) \le \sigma_{\max}(\mathbf{B}) \le L_{xy}. \tag{150}$$

Next, we establish the following Lemma J.2, which describes the solution to the problem defined above, the proof is available in Appendix J.1.2.

**Lemma J.2.** *For all $d \in \{2, 3, \ldots\}$, the instance of problem (1) defined above has a unique solution $(x^*, y^*) \in \mathcal{X} \times \mathcal{Y}$. Moreover, there exist vectors $x^\circ, y^\circ \in \mathrm{span}(\{(1, \rho, \ldots, \rho^{d-1})\})$ such that the following inequality holds:*

$$\mathcal{R}^2_{\delta_x\delta_y}(x^\circ, y^\circ) \le C_\pi A^2 \rho^{2d}, \tag{151}$$

*where $C_\pi > 0$ is some constant that possibly depends on the parameters $\pi \in \Pi$, but does not depend on $d$, and $\rho \in (0,1)$ satisfies the following inequality:*

$$\rho \ge \max\left\{1 - \frac{\sqrt{8}}{\sqrt{\kappa_{xy}}}, \frac{4}{5}\right\}. \tag{152}$$

*In addition, the initial distance to the solution $\mathcal{R}^2_{\delta_x\delta_y}(0,0)$ is a quadratic function of $A$, that is,*

$$\mathcal{R}^2_{\delta_x\delta_y}(0,0) = B_{\pi,d}A^2, \tag{153}$$

*where $B_{\pi,d} > 0$ is a constant that possibly depends on $d \in \{1, 2, \ldots\}$ and the parameters $\pi \in \Pi$, and satisfies the inequality $\min_{d\in\{1,2,\ldots\}} B_{\pi,d} > 0$.*

We can express $(x^\circ, y^\circ)$ from Lemma J.2 as follows:

$$x^\circ = u_x(1, \rho, \ldots, \rho^{d-1}), \quad y^\circ = u_y(1, \rho, \ldots, \rho^{d-1}), \quad \text{where} \quad u_x, u_y \in \mathbb{R}, \tag{154}$$

which implies

$$\|x^\circ\|^2 = \frac{u_x^2(1-\rho^{2d})}{1-\rho^2}, \quad \|y^\circ\|^2 = \frac{u_y^2(1-\rho^{2d})}{1-\rho^2}. \tag{155}$$

Further, we fix $k \in \{0, \ldots, d\}$. Using the sparse structure of matrix $\mathbf{B}$ defined in equation (149), and using the standard arguments (Nesterov, 2013; Ibrahim et al., 2020; Zhang et al., 2022a; Scaman et al., 2017; 2018; Kovalev et al., 2024), we can show that the output vector $(x_o(\tau), y_o(\tau)) \in \mathcal{X} \times \mathcal{Y}$ satisfies the following implication:

$$\tau \le D \cdot \tau_{\mathbf{B}}k \quad \Rightarrow \quad x_o(\tau), y_o(\tau) \in \begin{cases} \mathrm{span}(\{\mathbf{e}_1^d, \ldots, \mathbf{e}_k^d\}) & k > 0 \\ \{0\} & k = 0 \end{cases}, \tag{156}$$

where $D > 0$ is a universal constant. The right-hand side of this implication implies the following:

$$\mathcal{R}^2_{\delta_x\delta_y}(x_o(\tau), y_o(\tau)) = \delta_x\|x_o(\tau) - x^*\|^2 + \delta_y\|y_o(\tau) - y^*\|^2$$

$$\overset{(a)}{\geq} \tfrac{1}{2}\delta_x \|x_o(\tau) - x^\circ\|^2 + \tfrac{1}{2}\delta_y \|y_o(\tau) - y^\circ\|^2 - \mathcal{R}^2_{\delta_x \delta_y}(x^\circ, y^\circ)$$

$$\overset{(b)}{\geq} \tfrac{1}{2}(\delta_x u_x^2 + \delta_y u_y^2)\sum_{j=k}^{d-1}\rho^{2j} - \mathcal{R}^2_{\delta_x \delta_y}(x^\circ, y^\circ)$$

$$= \tfrac{1}{2}(\delta_x u_x^2 + \delta_y u_y^2)\frac{\rho^{2k} - \rho^{2d}}{1 - \rho^2} - \mathcal{R}^2_{\delta_x \delta_y}(x^\circ, y^\circ)$$

$$= \tfrac{1}{2}(\delta_x \|x^\circ\|^2 + \delta_y \|y^\circ\|^2)\frac{\rho^{2k} - \rho^{2d}}{1 - \rho^{2d}} - \mathcal{R}^2_{\delta_x \delta_y}(x^\circ, y^\circ)$$

$$\overset{(c)}{\geq} \left(\tfrac{1}{4}\mathcal{R}^2_{\delta_x \delta_y}(0, 0) - \tfrac{1}{2}\mathcal{R}^2_{\delta_x \delta_y}(x^\circ, y^\circ)\right)\frac{\rho^{2k} - \rho^{2d}}{1 - \rho^{2d}} - \mathcal{R}^2_{\delta_x \delta_y}(x^\circ, y^\circ)$$

$$= \frac{\rho^{2k} - \rho^{2d}}{4(1 - \rho^{2d})}\mathcal{R}^2_{\delta_x \delta_y}(0, 0) - \left(1 + \frac{\rho^{2k} - \rho^{2d}}{2(1 - \rho^{2d})}\right)\mathcal{R}^2_{\delta_x \delta_y}(x^\circ, y^\circ)$$

$$\overset{(d)}{\geq} \frac{\rho^{2k} - \rho^{2d}}{4(1 - \rho^{2d})}B_{\pi, d}A^2 - \left(1 + \frac{\rho^{2k} - \rho^{2d}}{2(1 - \rho^{2d})}\right)C_\pi A^2 \rho^{2d},$$

where (a) and (c) use Young's inequality; (b) uses equation (156) and the expression for $(x^\circ, y^\circ)$ in equation (154); (d) uses Lemma J.2. Furthermore, we can choose $A = R/\sqrt{B_{\pi, d}}$ to ensure the initial distance $\mathcal{R}^2_{\delta_x \delta_y}(0, 0) = R^2$ and obtain the following:

$$\mathcal{R}^2_{\delta_x \delta_y}(x_o(\tau), y_o(\tau)) = \frac{\rho^{2k} - \rho^{2d}}{4(1 - \rho^{2d})}B_{\pi, d}A^2 - \left(1 + \frac{\rho^{2k} - \rho^{2d}}{2(1 - \rho^{2d})}\right)C_\pi A^2 \rho^{2d}$$

$$\overset{(a)}{=} \frac{\rho^{2k} - \rho^{2d}}{4(1 - \rho^{2d})}R^2 - \left(1 + \frac{\rho^{2k} - \rho^{2d}}{2(1 - \rho^{2d})}\right)\frac{C_\pi R^2 \rho^{2d}}{B_{\pi, d}}$$

$$\overset{(b)}{\geq} \frac{\rho^{2k} - \rho^{2d}}{4(1 - \rho^{2d})}R^2 - \left(1 + \frac{\rho^{2k} - \rho^{2d}}{2(1 - \rho^{2d})}\right)\frac{C_\pi R^2 \rho^{2d}}{\min_{d' \in \{1,2,\dots\}} B_{\pi, d'}}$$

$$= \rho^{2k}R^2 \cdot \left(\frac{1 - \rho^{2(d-k)}}{4(1 - \rho^{2d})} - \left(1 + \frac{\rho^{2k} - \rho^{2d}}{2(1 - \rho^{2d})}\right)\frac{C_\pi \rho^{2(d-k)}}{\min_{d' \in \{1,2,\dots\}} B_{\pi, d'}}\right)$$

$$\overset{(c)}{\geq} \tfrac{1}{5}\rho^{2k}R^2,$$

where (a) uses the choice of $A$ above; (b) uses the inequalities $B_{\pi, d} \geq \min_{d' \in \{1,2,\dots\}} B_{\pi, d'} > 0$ implied by Lemma J.2; (c) is implied by choosing a large enough value of $d \in \{1, 2, \dots\}$.

Next, we consider the case $\epsilon \leq \tfrac{1}{5}R^2$. Choosing $k$ as follows:

$$k = \left\lfloor \frac{\log\left(\tfrac{1}{5}R^2/\epsilon\right)}{2\log\left(1/\rho\right)} \right\rfloor \tag{157}$$

implies $\mathcal{R}^2_{\delta_x \delta_y}(x_o(\tau), y_o(\tau)) \geq \epsilon$. Therefore, by contraposition, from implication (156), we obtain the following:

$$\mathcal{R}^2_{\delta_x \delta_y}(0, 0) < \epsilon \quad \Rightarrow \quad \tau > D \cdot \tau_{\mathbf{B}} k. \tag{158}$$

The right-hand side of this implication implies the following lower bound on the execution time $\tau$:

$$\tau > D \cdot \tau_{\mathbf{B}} k$$

$$\overset{(a)}{=} D \cdot \tau_{\mathbf{B}} \left\lfloor \frac{\log\left(\tfrac{1}{5}R^2/\epsilon\right)}{2\log\left(1/\rho\right)} \right\rfloor$$

$$\geq D \cdot \tau_{\mathbf{B}} \cdot \frac{\log\left(\tfrac{1}{5}R^2/\epsilon\right) - 2\log\left(1/\rho\right)}{2\log\left(1/\rho\right)}$$

$$\overset{(b)}{\geq} D \cdot \tau_{\mathbf{B}} \cdot \frac{\log\left(\tfrac{1}{5}R^2/\epsilon\right) - 2\log\left(5/4\right)}{2\log\left(1/\rho\right)}$$

$$= \frac{D}{2} \cdot \tau_{\mathbf{B}} \cdot \frac{\log\left(\frac{16}{125}R^2/\epsilon\right)}{\log\left(1/\rho\right)}$$

$$\overset{\text{(c)}}{\geq} \frac{D}{2} \cdot \tau_{\mathbf{B}} \cdot \frac{\log\left(\frac{16}{125}R^2/\epsilon\right)}{(1/\rho - 1)}$$

$$= \frac{D}{2} \cdot \frac{\tau_{\mathbf{B}}\rho}{1 - \rho} \cdot \log\left(\frac{16R^2}{125\epsilon}\right)$$

$$\overset{\text{(d)}}{\geq} \frac{D}{5\sqrt{2}} \cdot \tau_{\mathbf{B}}\sqrt{\kappa_{xy}} \cdot \log\left(\frac{16R^2}{125\epsilon}\right)$$

$$= \Omega\left(\tau_{\mathbf{B}} \cdot \sqrt{\kappa_{xy}}\log\left(\frac{16R^2}{125\epsilon}\right)\right),$$

where (a) uses the definition of $k$ in equation (157); (b) and (d) use Lemma J.2; (c) uses the concavity of the logarithm, which concludes the proof in the case $\epsilon \leq \frac{1}{5}R^2$. In the remaining case $\epsilon > \frac{1}{5}R^2$, we have $\log\left(\frac{16R^2}{125\epsilon}\right) \leq 0$. Hence, the latter lower bound holds due to the fact that $\tau \geq 0$, which concludes the proof. $\square$

## J.1. Proofs of Auxiliary Lemmas

### J.1.1. PROOF OF LEMMA J.1

We can write matrix $\mathbf{B}^\top\mathbf{B}$ as follows:

$$\mathbf{B}^\top\mathbf{B} = \frac{1}{4}\begin{bmatrix} \alpha^2 & -\alpha\beta & & & \\ -\alpha\beta & \alpha^2 + \beta^2 & -\alpha\beta & & \\ & \ddots & \ddots & \ddots & \\ & & -\alpha\beta & \alpha^2 + \beta^2 & -\alpha\beta \\ & & & -\alpha\beta & \alpha^2 + \beta^2 \end{bmatrix}$$

$$= \frac{1}{4}\begin{bmatrix} \alpha\beta & -\alpha\beta & & & \\ -\alpha\beta & 2\alpha\beta & -\alpha\beta & & \\ & \ddots & \ddots & \ddots & \\ & & -\alpha\beta & 2\alpha\beta & -\alpha\beta \\ & & & -\alpha\beta & 2\alpha\beta \end{bmatrix} + \frac{1}{4}\begin{bmatrix} \alpha(\alpha - \beta) & & & \\ & (\alpha - \beta)^2 & & \\ & & \ddots & \\ & & & (\alpha - \beta)^2 \end{bmatrix}.$$

Therefore, we can obtain the following matrix inequalities:

$$\begin{aligned} \mathbf{B}^\top\mathbf{B} &\succeq \tfrac{1}{4}\min\{(\alpha - \beta)^2, \alpha(\alpha - \beta)\}\mathbf{I}_d = \bar{\mu}_{yx}^2\mathbf{I}_d, \\ \mathbf{B}^\top\mathbf{B} &\preceq \tfrac{1}{4}\max\left\{(\alpha - \beta)^2 + 4\alpha\beta, \alpha(\alpha - \beta) + 2\alpha\beta\right\}\mathbf{I}_d = L_{xy}^2\mathbf{I}_d, \end{aligned} \tag{159}$$

which conclude the proof. $\square$

### J.1.2. PROOF OF LEMMA J.2

The first-order optimality conditions (7) imply that the unique solution to the problem $(x^*, y^*) \in \mathcal{X} \times \mathcal{Y}$ is defined by the following linear system:

$$\begin{aligned} (\mu_x\mu_y\mathbf{I}_d + \mathbf{B}^\top\mathbf{B})x^* &= \mu_y A\mathbf{e}_1^d, \\ \mu_y y^* - \mathbf{B}x^* &= 0. \end{aligned} \tag{160}$$

Hence, we can express the initial distance $\mathcal{R}_{\delta_x\delta_y}^2(0, 0)$ as follows:

$$\begin{aligned} \mathcal{R}_{\delta_x\delta_y}^2(0, 0) &= \delta_x\|x^*\|^2 + \delta_y\|y^*\|^2 \\ &\overset{\text{(a)}}{=} \delta_x\|\mu_y A(\mu_x\mu_y\mathbf{I}_d + \mathbf{B}^\top\mathbf{B})^{-1}\mathbf{e}_1^d\|^2 + \delta_y\|A\mathbf{B}(\mu_x\mu_y\mathbf{I}_d + \mathbf{B}^\top\mathbf{B})^{-1}\mathbf{e}_1^d\|^2 \\ &= A^2 B_{\pi,d}, \end{aligned}$$

where (a) uses the linear system (160) and the fact that the matrix $(\mu_x\mu_y\mathbf{I}_d + \mathbf{B}^\top\mathbf{B})$ is invertible, which is implied by the matrix inequality $(\mu_x\mu_y\mathbf{I}_d + \mathbf{B}^\top\mathbf{B}) \succeq (\mu_x\mu_y + \bar{\mu}_{xy}^2)\mathbf{I}_d$ and Assumption 2.6, and where $B_{\pi,d}$ is defined as follows:

$$B_{\pi,d} = \delta_x\mu_y^2\|(\mu_x\mu_y\mathbf{I}_d + \mathbf{B}^\top\mathbf{B})^{-1}\mathbf{e}_1^d\|^2 + \delta_y\|\mathbf{B}(\mu_x\mu_y\mathbf{I}_d + \mathbf{B}^\top\mathbf{B})^{-1}\mathbf{e}_1^d\|^2. \tag{161}$$

We can also lower-bound $B_{\pi,d}$ as follows:

$$B_{\pi,d} = \delta_x \mu_y^2 \|(\mu_x \mu_y \mathbf{I}_d + \mathbf{B}^\top \mathbf{B})^{-1} \mathbf{e}_1^d\|^2 + \delta_y \|\mathbf{B}(\mu_x \mu_y \mathbf{I}_d + \mathbf{B}^\top \mathbf{B})^{-1} \mathbf{e}_1^d\|^2$$
$$\overset{(a)}{\geq} \frac{\delta_x \mu_y^2 + \delta_y \bar{\mu}_{xy}^2}{(\mu_x \mu_y + L_{xy}^2)^2} \overset{(b)}{>} 0,$$

where (a) uses Lemma J.1 and the fact that $\|\mathbf{e}_1^d\|^2 = 1$; (b) uses Assumption 2.6 and the assumption $\mu_y > 0$. Therefore, we obtain the desired inequality $\min_{d \in \{1,2,\dots\}} B_{\pi,d} > 0$.

Furthermore, the first equation in equation (160) can be written as follows:

$$\begin{bmatrix} (\gamma^2 - \frac{\beta}{\alpha}) & -1 & & & & \\ -1 & \gamma^2 & -1 & & & \\ & \ddots & \ddots & \ddots & & \\ & & \ddots & \ddots & \ddots & \\ & & & -1 & \gamma^2 & -1 \\ & & & & -1 & \gamma^2 \end{bmatrix} x^* = \frac{4\mu_y A}{\alpha\beta} \mathbf{e}_1^d, \tag{162}$$

where $\gamma^2$ is defined as follows:

$$\gamma^2 = \frac{\alpha^2 + \beta^2 + 4\mu_x \mu_y}{\alpha\beta} = \frac{2(q+1)}{(q-1)}, \tag{163}$$

where $q > 1$ is defined as follows:

$$q = \frac{L_{xy}^2 + \mu_x \mu_y}{\bar{\mu}_{xy}^2 + \mu_x \mu_y}, \tag{164}$$

where the denominator is always positive due to Assumption 2.6. Let $(x^\circ, y^\circ) \in \mathcal{X} \times \mathcal{Y}$ be defined as follows:

$$x^\circ = \frac{4\mu_y A}{\beta(\alpha - \beta\rho)} \cdot (\rho, \rho^2, \dots, \rho^d), \quad y^\circ = \frac{2A}{\beta}(\rho, \rho^2, \dots, \rho^d), \tag{165}$$

where $\rho \in (0,1)$ is defined as follows:

$$\rho = \frac{\sqrt{q} - 1}{\sqrt{q} + 1}. \tag{166}$$

One can verify that $(x^\circ, y^\circ)$ satisfies the following linear system:

$$(\mu_x \mu_y \mathbf{I}_d + \mathbf{B}^\top \mathbf{B})x^\circ = \mu_y A \mathbf{e}_1^d + \frac{\alpha \rho^{d+1}}{\alpha - \beta\rho} \cdot \mu_y A \mathbf{e}_d^d,$$
$$\mu_y y^\circ - \mathbf{B}x^\circ = -\frac{2\rho^{d+1}}{\alpha - \beta\rho} \cdot \mu_y A \mathbf{e}_d^d, \tag{167}$$

which, together with equation (160), implies

$$(\mu_x \mu_y \mathbf{I}_d + \mathbf{B}^\top \mathbf{B})(x^\circ - x^*) = \frac{\alpha \rho^{d+1}}{\alpha - \beta\rho} \cdot \mu_y A \mathbf{e}_d^d,$$
$$\mu_y(y^\circ - y^*) - \mathbf{B}(x^\circ - x^*) = -\frac{2\rho^{d+1}}{\alpha - \beta\rho} \cdot \mu_y A \mathbf{e}_d^d, \tag{168}$$

Hence, we can upper-bound $\|x^\circ - x^*\|$ as follows:

$$\|x^\circ - x^*\| \overset{(a)}{=} \frac{\alpha \mu_y A \rho^{d+1}}{\alpha - \beta\rho} \cdot \|(\mu_x \mu_y \mathbf{I}_d + \mathbf{B}^\top \mathbf{B})^{-1} \mathbf{e}_d^d\|$$
$$\overset{(b)}{\leq} \frac{\alpha \mu_y A \rho^{d+1}}{(\mu_x \mu_y + \bar{\mu}_{xy}^2)(\alpha - \beta\rho)},$$

where (a) uses the linear system (160) and the fact that the matrix $(\mu_x\mu_y\mathbf{I}_d + \mathbf{B}^\top\mathbf{B})$ is invertible, which is implied by the matrix inequality $(\mu_x\mu_y\mathbf{I}_d + \mathbf{B}^\top\mathbf{B}) \succeq (\mu_x\mu_y + \bar{\mu}_{xy}^2)\mathbf{I}_d$ and Assumption 2.6; (b) uses the fact that $\mu_x\mu_y\mathbf{I}_d + \mathbf{B}^\top\mathbf{B} \succeq (\mu_x\mu_y + \bar{\mu}_{xy}^2)\mathbf{I}_d$ and $\|\mathbf{e}_d^d\| = 1$. Furthermore, we can upper-bound $\|y^\circ - y^*\|$ as follows:

$$
\begin{aligned}
\|y^\circ - y^*\| &\overset{\text{(a)}}{=} \frac{1}{\mu_y}\left\|\mathbf{B}(x^\circ - x^*) - \frac{2\rho^{d+1}}{\alpha - \beta\rho}\cdot\mu_y A\mathbf{e}_d^d\right\| \\
&\overset{\text{(b)}}{\leq} \frac{1}{\mu_y}\|\mathbf{B}(x^\circ - x^*)\| + \frac{2A\rho^{d+1}}{\alpha - \beta\rho}\cdot\|\mathbf{e}_d^d\| \\
&\overset{\text{(c)}}{\leq} \frac{L_{xy}}{\mu_y}\|x^\circ - x^*\| + \frac{2A\rho^{d+1}}{\alpha - \beta\rho} \\
&\overset{\text{(d)}}{\leq} \frac{\alpha L_{xy}A\rho^{d+1}}{(\mu_x\mu_y + \bar{\mu}_{xy}^2)(\alpha - \beta\rho)} + \frac{2A\rho^{d+1}}{\alpha - \beta\rho} \\
&= \left(2 + \frac{\alpha L_{xy}}{\mu_x\mu_y + \bar{\mu}_{xy}^2}\right)\frac{A\rho^{d+1}}{\alpha - \beta\rho},
\end{aligned}
$$

where (a) uses the linear system above; (b) uses the triangle inequality; (c) uses the fact that $\|\mathbf{B}\| \leq L_{xy}$ and $\|\mathbf{e}_d^d\| = 1$; (d) uses the previously obtained upper bound on $\|x^\circ - x^*\|$. Combining the upper bounds on $\|x^\circ - x^*\|$ and $\|y^\circ - y^*\|$ gives the desired inequality:

$$
\mathcal{R}^2_{\delta_x\delta_y}(x^\circ, y^\circ) = \delta_x\|x^\circ - x^*\|^2 + \delta_y\|y^\circ - y^*\|^2 \leq C_\pi A^2\rho^{2d}. \tag{169}
$$

It remains to lower-bound $\rho$ as follows:

$$
\begin{aligned}
\rho &\overset{\text{(a)}}{=} 1 - \frac{2}{\sqrt{q}+1} \overset{\text{(b)}}{=} 1 - \frac{2}{\sqrt{\frac{L_{xy}^2 + \mu_x\mu_y}{\bar{\mu}_{xy}^2 + \mu_x\mu_y}} + 1} \overset{\text{(c)}}{\geq} 1 - \frac{2}{\sqrt{\frac{18^2\max\{\bar{\mu}_{xy}^2,\mu_x\mu_y\} + \mu_x\mu_y}{\bar{\mu}_{xy}^2 + \mu_x\mu_y}} + 1} \\
&\geq 1 - \frac{2}{\sqrt{\frac{(18^2-1)\max\{\bar{\mu}_{xy}^2,\mu_x\mu_y\}}{\bar{\mu}_{xy}^2 + \mu_x\mu_y} + 1} + 1} \geq 1 - \frac{2}{\sqrt{\frac{(18^2-1)}{2} + 1} + 1} \geq \frac{4}{5},
\end{aligned}
$$

where (a) uses the definition of $\rho$ in equation (166); (b) uses the definition of $q$ in equation (164); (c) uses Assumption 2.7. In addition, we can lower-bound $\rho$ as follows:

$$
\rho \overset{\text{(a)}}{=} 1 - \frac{2}{\sqrt{q}+1} \overset{\text{(b)}}{=} 1 - \frac{2}{\sqrt{\frac{L_{xy}^2 + \mu_x\mu_y}{\bar{\mu}_{xy}^2 + \mu_x\mu_y}} + 1} \overset{\text{(c)}}{\geq} 1 - \frac{2}{\sqrt{\frac{L_{xy}^2}{2\mu_x\mu_y}}} \overset{\text{(d)}}{\geq} 1 - \frac{\sqrt{8}}{\sqrt{\kappa_{xy}}},
$$

where (a) uses the definition of $\rho$ in equation (166); (b) uses the definition of $q$ in equation (164); (c) uses the assumption $\mu_x\mu_y \geq \max\{\mu_{xy}^2, \mu_{yx}^2\}$; (d) uses the definition of $\kappa_{xy}$ in equation (6) and the definitions of $\delta_x$ and $\delta_y$ in equation (4), which concludes the proof. $\qquad\square$

## K. Proof of Theorem 4.5

Without loss of generality, we can assume $\mathbf{P} = \mathbf{I}_{d_z}$. Otherwise, we can simply make a variable change $z \to \mathbf{P}^{1/2} z$. That is, we can replace functions $p_i(z)$ and operators $Q_i(z)$ with $p_i(\mathbf{P}^{-1/2} z)$ and $\mathbf{P}^{-1/2} Q_i(-\mathbf{P}^{1/2} z)$.

We start with the following Lemmas K.1, K.2 and K.3 that describe the basic properties of functions $p_i^{k;t_1,\dots,t_k}(z)$ and $\hat{p}_i^{k;t_1,\dots,t_k}(z)$, and operators $Q_i(z)$. The proofs of these lemmas is a trivial utilization of the definitions of functions $p_i^{k;t_1,\dots,t_k}(z)$ and $\hat{p}_i^{k;t_1,\dots,t_k}(z)$ on lines 17 and 22 of Algorithm 1, and the definitions of constants $L_i^{k;t_1,\dots,t_k}$, $M_i^{k;t_1,\dots,t_k}$, and $H_i^{k;t_1,\dots,t_k}$ on lines 18, 19 and 20 of Algorithm 1. We omit these proofs due to their simplicity.

**Lemma K.1.** *For all $k \le i$, function $\hat{p}_i^{k;t_1,\dots,t_k}(z)$ is $L_i^{k;t_1,\dots,t_k}$-smooth.*

**Lemma K.2.** *For all $k \ge i$, function $p_i^{k;t_1,\dots,t_k}(z)$ is $H_i^{k;t_1,\dots,t_k}$-strongly convex.*

**Lemma K.3.** *For all $1 \le i \le n$, operator $Q_i(z)$ is $M_i^{k;t_1,\dots,t_k}$-Lipschitz.*

Next, we establish the key Lemma K.4 that describes the convergence properties of Algorithm 1. The proof is available in Appendix K.1.1.

**Lemma K.4.** *For all $1 \le k \le n$ and $\hat{z} \in \mathcal{C}_z$, the following inequality holds:*

$$
\begin{aligned}
0 \ge \sum_{i=1}^{n} & \left( r_i^{k;t_1,\dots,t_k}(z_{t_1,\dots,t_{k-1},t_k+1/2}^k) - r_i^{k;t_1,\dots,t_k}(\hat{z}) \right) + \frac{H_\Sigma^k}{2} \| z_{t_1,\dots,t_{k-1},t_k+1/2}^k - \hat{z} \|^2 \\
& + \sum_{i=k+1}^{n} \left( D_{i;t_k+1}^{k;t_1,\dots,t_k} - D_{i;t_k}^{k;t_1,\dots,t_k} \right),
\end{aligned}
\tag{170}
$$

*where function $r_i^{k;t_1,\dots,t_k}(z) \colon \mathcal{Z} \to \mathbb{R}$ is defined as follows:*

$$
r_i^{k;t_1,\dots,t_k}(z) = p_i^{k;t_1,\dots,t_k}(z) + \begin{cases} \langle z, Q_i(\hat{z}) \rangle & i > k \\ 0 & i \le k \end{cases},
\tag{171}
$$

*constant $H_\Sigma^k \ge 0$ is defined as follows:*

$$
H_\Sigma^k = \sum_{i=1}^{k} H_i^{i;t_1,\dots,t_i},
\tag{172}
$$

*and $D_{i;t}^{k;t_1,\dots,t_k} \ge 0$ is defined as follows:*

$$
D_{i;t}^{k;t_1,\dots,t_k} = \frac{L_i^{k;t_1\dots,t_k} \prod_{j=k+1}^{i} \alpha_{T_j-1}^2 + M_i^{k;t_1\dots,t_k} \prod_{j=k+1}^{i} \alpha_{T_j-1}}{2} \| z_{t_1,\dots,t_{k-1},t,0\dots0}^i - \hat{z} \|^2.
\tag{173}
$$

Now, we are ready to prove Theorem 4.5. Using the initialization steps on lines 4 and 5 of Algorithm 1, the definition of the output $z_{\text{out}} \in \mathcal{Z}$ on line 7 of Algorithm 1, and the arguments that are identical to the proof of Lemma K.4, we obtain the following simplified version of equation (170) for the case $k = 0$:

$$
\sum_{i=1}^{n} \left( r_i^0(z_{\text{out}}) - r_i^0(\hat{z}) \right) \le \sum_{i=1}^{n} \frac{L_i \prod_{j=1}^{i} \alpha_{T_j-1}^2 + M_i \prod_{j=1}^{i} \alpha_{T_j-1}}{2} \cdot \| z_{\text{in}} - \hat{z} \|^2.
\tag{174}
$$

Furthermore, we prove the following Lemma K.5 in Appendix K.1.2.

**Lemma K.5.** *The output $z_{out}$ of Algorithm 1 satisfies the inclusion $z_{out} \in \mathcal{C}_z$.*

It remains to upper-bound $\alpha_t$. First, we lower-bound $\alpha_t^{-1}$ as follows:

$$
\alpha_t^{-1} \overset{(a)}{=} \tfrac{1}{2} + \tfrac{1}{2}\sqrt{1 + 4\alpha_{t-1}^{-2}} \ge \tfrac{1}{2} + \alpha_{t-1}^{-1} \ge \cdots \ge \tfrac{1}{2}t + \alpha_0^{-1} \overset{(b)}{=} \tfrac{1}{2}(t+2),
$$

where (a) and (b) use the definition of $\alpha_t$ in equation (20). This lower bound on $\alpha_t^{-1}$ implies the following upper bound on $\alpha_{T-1}$:

$$
\alpha_{T-1} \le \frac{2}{T+1},
\tag{175}
$$

which concludes the proof. $\qquad\square$

## K.1. Proofs of Auxiliary Lemmas

### K.1.1. PROOF OF LEMMA K.4

We prove this lemma by induction for $k = n, \ldots, 1$.

**Base case ($k = n$).** Using lines 12 and 24 of Algorithm 1, we get the following relation:

$$z^n_{t_1,\ldots,t_{n-1},t_n+1/2} = \underset{z \in \mathcal{C}_z}{\arg\min} \sum_{i=1}^{n} p^{n;t_1,\ldots,t_n}_i(z). \tag{176}$$

Moreover, function $\sum_{i=1}^{n} p^{n;t_1,\ldots,t_n}_i(z)$ is $H^n_\Sigma$-strongly convex due to Lemma K.2 and the definition of $H^n_\Sigma$ in equation (172). Hence, using the fact that $\hat{z} \in \mathcal{C}_z$ and the fact that

$$p^{n;t_1,\ldots,t_n}_i(\hat{z}) = r^{n;t_1,\ldots,t_n}_i(\hat{z}), \tag{177}$$

which is implied by equation (171), we obtain the following inequality:

$$\sum_{i=1}^{n} r^{n;t_1,\ldots,t_n}_i(\hat{z}) \geq \sum_{i=1}^{n} r^{n;t_1,\ldots,t_n}_i(z^n_{t_1,\ldots,t_{n-1},t_n+1/2}) + \frac{H^n_\Sigma}{2}\|z^n_{t_1,\ldots,t_{n-1},t_n+1/2} - \hat{z}\|^2, \tag{178}$$

which is nothing else but the desired equation (170) in the case $k = n$.

**Induction step ($k \to k-1$).** We assume that equation (170) holds for $2 \leq k \leq n$, which implies the following:

$$0 \geq \sum_{i=k+1}^{n} \left( D^{k;t_1,\ldots,t_k}_{i;t_k+1} - D^{k;t_1,\ldots,t_k}_{i;t_k} \right) + \frac{H^k_\Sigma}{2}\|z^k_{t_1,\ldots,t_{k-1},t_k+1/2} - \hat{z}\|^2$$

$$+ \sum_{i=1}^{n} \left( r^{k;t_1,\ldots,t_k}_i(z^k_{t_1,\ldots,t_{k-1},t_k+1/2}) - r^{k;t_1,\ldots,t_k}_i(\hat{z}) \right)$$

$$\overset{(a)}{=} \sum_{i=k+1}^{n} \left( D^{k;t_1,\ldots,t_k}_{i;t_k+1} - D^{k;t_1,\ldots,t_k}_{i;t_k} \right) + \frac{H^k_\Sigma}{2}\|z^k_{t_1,\ldots,t_{k-1},t_k+1/2} - \hat{z}\|^2$$

$$+ \sum_{i=1,i\neq k}^{n} \left( \hat{p}^{k;t_1,\ldots,t_k}_i(z^k_{t_1,\ldots,t_{k-1},t_k+1/2}) - \hat{p}^{k;t_1,\ldots,t_k}_i(\hat{z}) \right) + \sum_{i=k+1}^{n} \langle Q_i(\hat{z}), z^k_{t_1,\ldots,t_{k-1},t_k+1/2} - \hat{z} \rangle$$

$$+ \frac{H^{k;t_1,\ldots,t_k}_k}{2}\|z^k_{t_1,\ldots,t_{k-1},t_k+1/2} - z^k_{t_1,\ldots,t_k}\|^2 - \frac{H^{k;t_1,\ldots,t_k}_k}{2}\|\hat{z} - z^k_{t_1,\ldots,t_k}\|^2$$

$$+ \langle z^k_{t_1,\ldots,t_{k-1},t_k+1/2} - \hat{z}, \Delta^{k;t_1,\ldots,t_k}_k \rangle$$

$$\overset{(b)}{=} \sum_{i=k+1}^{n} \left( D^{k;t_1,\ldots,t_k}_{i;t_k+1} - D^{k;t_1,\ldots,t_k}_{i;t_k} \right) + \frac{H^k_\Sigma}{2}\|z^k_{t_1,\ldots,t_{k-1},t_k+1/2} - \hat{z}\|^2$$

$$+ \sum_{i=1,i\neq k}^{n} \left( \hat{p}^{k;t_1,\ldots,t_k}_i(z^k_{t_1,\ldots,t_{k-1},t_k+1/2}) - \hat{p}^{k;t_1,\ldots,t_k}_i(\hat{z}) \right) + \sum_{i=k+1}^{n} \langle Q_i(\hat{z}), z^k_{t_1,\ldots,t_{k-1},t_k+1/2} - \hat{z} \rangle$$

$$+ \langle z^k_{t_1,\ldots,t_{k-1},t_k+1/2} - \hat{z}, \nabla\hat{p}^{k;t_1,\ldots,t_k}_k(z^k_{t_1,\ldots,t_k}) \rangle + \frac{H^{k;t_1,\ldots,t_k}_k}{2}\|z^k_{t_1,\ldots,t_{k-1},t_k+1/2} - z^k_{t_1,\ldots,t_k}\|^2$$

$$+ \langle z^k_{t_1,\ldots,t_{k-1},t_k+1/2} - \hat{z}, Q_k(z^k_{t_1,\ldots,t_k}) \rangle - \frac{H^{k;t_1,\ldots,t_k}_k}{2}\|\hat{z} - z^k_{t_1,\ldots,t_k}\|^2$$

$$\overset{(c)}{\geq} \sum_{i=k+1}^{n} \left( D^{k;t_1,\ldots,t_k}_{i;t_k+1} - D^{k;t_1,\ldots,t_k}_{i;t_k} \right) + \frac{H^k_\Sigma}{2}\|z^k_{t_1,\ldots,t_{k-1},t_k+1/2} - \hat{z}\|^2$$

$$+ \sum_{i=1}^{n} \left( \hat{p}^{k;t_1,\ldots,t_k}_i(z^k_{t_1,\ldots,t_{k-1},t_k+1/2}) - \hat{p}^{k;t_1,\ldots,t_k}_i(\hat{z}) \right) + \sum_{i=k+1}^{n} \langle Q_i(\hat{z}), z^k_{t_1,\ldots,t_{k-1},t_k+1/2} - \hat{z} \rangle$$

$$+ \frac{H^{k;t_1,\ldots,t_k}_k - L^{k;t_1,\ldots,t_k}_k}{2}\|z^k_{t_1,\ldots,t_{k-1},t_k+1/2} - z^k_{t_1,\ldots,t_k}\|^2 - \frac{H^{k;t_1,\ldots,t_k}_k}{2}\|\hat{z} - z^k_{t_1,\ldots,t_k}\|^2$$

$$+ \langle z_{t_1,\ldots,t_{k-1},t_k+1/2}^k - \hat{z}, Q_k(z_{t_1,\ldots,t_k}^k)\rangle$$

$$\overset{\text{(d)}}{=} \sum_{i=k+1}^n \left( D_{i;t_k+1}^{k;t_1,\ldots,t_k} - D_{i;t_k}^{k;t_1,\ldots,t_k} \right) + \frac{H_\Sigma^k}{2}\|z_{t_1,\ldots,t_{k-1},t_k+1/2}^k - \hat{z}\|^2$$

$$+ \sum_{i=1}^n \left( \hat{p}_i^{k;t_1,\ldots,t_k}(z_{t_1,\ldots,t_{k-1},t_k+1/2}^k) - \hat{p}_i^{k;t_1,\ldots,t_k}(\hat{z}) \right) + \sum_{i=k+1}^n \langle Q_i(\hat{z}), z_{t_1,\ldots,t_{k-1},t_k+1/2}^k - \hat{z}\rangle$$

$$+ \frac{H_k^{k;t_1,\ldots,t_k} - L_k^{k;t_1,\ldots,t_k}}{2}\|z_{t_1,\ldots,t_{k-1},t_k+1/2}^k - z_{t_1,\ldots,t_k}^k\|^2 - \frac{H_k^{k;t_1,\ldots,t_k}}{2}\|\hat{z} - z_{t_1,\ldots,t_k}^k\|^2$$

$$+ \langle z_{t_1,\ldots,t_{k-1},t_k+1/2}^k - \hat{z}, Q_k(z_{t_1,\ldots,t_{k-1},t_k+1/2}^k) + \Delta_{Q_k}^{k;t_1,\ldots,t_k}\rangle$$

$$\overset{\text{(e)}}{\geq} \sum_{i=k+1}^n \left( D_{i;t_k+1}^{k;t_1,\ldots,t_k} - D_{i;t_k}^{k;t_1,\ldots,t_k} \right) + \frac{H_\Sigma^k}{2}\|z_{t_1,\ldots,t_{k-1},t_k+1/2}^k - \hat{z}\|^2$$

$$+ \sum_{i=1}^n \left( \hat{p}_i^{k;t_1,\ldots,t_k}(z_{t_1,\ldots,t_{k-1},t_k+1/2}^k) - \hat{p}_i^{k;t_1,\ldots,t_k}(\hat{z}) \right) + \sum_{i=k}^n \langle Q_i(\hat{z}), z_{t_1,\ldots,t_{k-1},t_k+1/2}^k - \hat{z}\rangle$$

$$+ \frac{H_k^{k;t_1,\ldots,t_k} - L_k^{k;t_1,\ldots,t_k}}{2}\|z_{t_1,\ldots,t_{k-1},t_k+1/2}^k - z_{t_1,\ldots,t_k}^k\|^2 - \frac{H_k^{k;t_1,\ldots,t_k}}{2}\|\hat{z} - z_{t_1,\ldots,t_k}^k\|^2$$

$$+ \langle z_{t_1,\ldots,t_{k-1},t_k+1/2}^k - \hat{z}, \Delta_{Q_k}^{k;t_1,\ldots,t_k}\rangle$$

$$\overset{\text{(f)}}{=} \sum_{i=k+1}^n \left( D_{i;t_k+1}^{k;t_1,\ldots,t_k} - D_{i;t_k}^{k;t_1,\ldots,t_k} \right) + \frac{H_\Sigma^k}{2}\|z_{t_1,\ldots,t_{k-1},t_k+1/2}^k - \hat{z}\|^2$$

$$+ \sum_{i=1}^n \left( \hat{p}_i^{k;t_1,\ldots,t_k}(z_{t_1,\ldots,t_{k-1},t_k+1/2}^k) - \hat{p}_i^{k;t_1,\ldots,t_k}(\hat{z}) \right) + \sum_{i=k}^n \langle Q_i(\hat{z}), z_{t_1,\ldots,t_{k-1},t_k+1/2}^k - \hat{z}\rangle$$

$$+ \frac{H_k^{k;t_1,\ldots,t_k} - L_k^{k;t_1,\ldots,t_k}}{2}\|z_{t_1,\ldots,t_{k-1},t_k+1/2}^k - z_{t_1,\ldots,t_k}^k\|^2 - \frac{H_k^{k;t_1,\ldots,t_k}}{2}\|\hat{z} - z_{t_1,\ldots,t_k}^k\|^2$$

$$+ \frac{H_k^{k;t_1,\ldots,t_k}}{2}\|z_{t_1,\ldots,t_{k-1},t_k+1/2}^k + (H_k^{k;t_1,\ldots,t_k})^{-1}\Delta_{Q_k}^{k;t_1,\ldots,t_k} - \hat{z}\|^2$$

$$- \frac{H_k^{k;t_1,\ldots,t_k}}{2}\|z_{t_1,\ldots,t_{k-1},t_k+1/2}^k - \hat{z}\|^2 - \frac{1}{2H_k^{k;t_1,\ldots,t_k}}\|\Delta_{Q_k}^{k;t_1,\ldots,t_k}\|^2$$

$$\overset{\text{(g)}}{=} \sum_{i=k+1}^n \left( D_{i;t_k+1}^{k;t_1,\ldots,t_k} - D_{i;t_k}^{k;t_1,\ldots,t_k} \right) + \frac{H_\Sigma^{k-1}}{2}\|z_{t_1,\ldots,t_{k-1},t_k+1/2}^k - \hat{z}\|^2$$

$$+ \sum_{i=1}^n \left( \hat{p}_i^{k;t_1,\ldots,t_k}(z_{t_1,\ldots,t_{k-1},t_k+1/2}^k) - \hat{p}_i^{k;t_1,\ldots,t_k}(\hat{z}) \right) + \sum_{i=k}^n \langle Q_i(\hat{z}), z_{t_1,\ldots,t_{k-1},t_k+1/2}^k - \hat{z}\rangle$$

$$+ \frac{H_k^{k;t_1,\ldots,t_k} - L_k^{k;t_1,\ldots,t_k}}{2}\|z_{t_1,\ldots,t_{k-1},t_k+1/2}^k - z_{t_1,\ldots,t_k}^k\|^2 - \frac{H_k^{k;t_1,\ldots,t_k}}{2}\|\hat{z} - z_{t_1,\ldots,t_k}^k\|^2$$

$$+ \frac{H_k^{k;t_1,\ldots,t_k}}{2}\|z_{t_1,\ldots,t_{k-1},t_k+1}^k - \hat{z}\|^2 - \frac{1}{2H_k^{k;t_1,\ldots,t_k}}\|Q_k(z_{t_1,\ldots,t_k}^k) - Q_k(z_{t_1,\ldots,t_{k-1},t_k+1/2}^k)\|^2$$

$$\overset{\text{(h)}}{\geq} \sum_{i=k+1}^n \left( D_{i;t_k+1}^{k;t_1,\ldots,t_k} - D_{i;t_k}^{k;t_1,\ldots,t_k} \right) + \frac{H_\Sigma^{k-1}}{2}\|z_{t_1,\ldots,t_{k-1},t_k+1/2}^k - \hat{z}\|^2$$

$$+ \sum_{i=1}^n \left( \hat{p}_i^{k;t_1,\ldots,t_k}(z_{t_1,\ldots,t_{k-1},t_k+1/2}^k) - \hat{p}_i^{k;t_1,\ldots,t_k}(\hat{z}) \right) + \sum_{i=k}^n \langle Q_i(\hat{z}), z_{t_1,\ldots,t_{k-1},t_k+1/2}^k - \hat{z}\rangle$$

$$+ \frac{(H_k^{k;t_1,\ldots,t_k})^2 - L_k^{k;t_1,\ldots,t_k}H_k^{k;t_1,\ldots,t_k} - (M_k^{k;t_1,\ldots,t_k})^2}{2H_k^{k;t_1,\ldots,t_k}}\|z_{t_1,\ldots,t_{k-1},t_k+1/2}^k - z_{t_1,\ldots,t_k}^k\|^2$$

$$- \frac{H_k^{k;t_1,\ldots,t_k}}{2}\|\hat{z} - z_{t_1,\ldots,t_k}^k\|^2 + \frac{H_k^{k;t_1,\ldots,t_k}}{2}\|z_{t_1,\ldots,t_{k-1},t_k+1}^k - \hat{z}\|^2$$

$$\overset{(i)}{\geq} \sum_{i=k+1}^{n} \left( D_{i;t_k+1}^{k;t_1,\ldots,t_k} - D_{i;t_k}^{k;t_1,\ldots,t_k} \right) + \frac{H_{\Sigma}^{k-1}}{2} \| z_{t_1,\ldots,t_{k-1},t_k+1/2}^{k} - \hat{z} \|^2$$

$$+ \sum_{i=1}^{n} \left( \hat{p}_i^{k;t_1,\ldots,t_k}(z_{t_1,\ldots,t_{k-1},t_k+1/2}^{k}) - \hat{p}_i^{k;t_1,\ldots,t_k}(\hat{z}) \right) + \sum_{i=k}^{n} \langle Q_i(\hat{z}), z_{t_1,\ldots,t_{k-1},t_k+1/2}^{k} - \hat{z} \rangle$$

$$- \frac{H_k^{k;t_1,\ldots,t_k}}{2} \| \hat{z} - z_{t_1,\ldots,t_k}^{k} \|^2 + \frac{H_k^{k;t_1,\ldots,t_k}}{2} \| z_{t_1,\ldots,t_{k-1},t_k+1}^{k} - \hat{z} \|^2,$$

where (a) uses the definition of functions $r_i^{k;t_1,\ldots,t_k}(z)$ in equation (171) and the definition of functions $p_i^{k;t_1,\ldots,t_k}(z)$ on line 22 of Algorithm 1; (b) uses the definition of $\Delta_k^{k;t_1,\ldots,t_k}$ on line 21 of Algorithm 1; (c) uses the convexity and $L_k^{k;t_1,\ldots,t_k}$-smoothness of function $\hat{p}_k^{k;t_1,\ldots,t_k}(z_{t_1,\ldots,t_k}^{k})$, where the smoothness property is implied by Lemma K.1; (d) uses the definition of $\Delta_{Q_k}^{k;t_1,\ldots,t_k}$ on line 26 of Algorithm 1; (e) uses the monotonicity of operator $Q_k(z)$; (f) uses the parallelogram rule of the form $\langle a, b \rangle = \frac{c}{2}\|a + b/c\|^2 - \frac{c}{2}\|a\|^2 - \frac{1}{2c}\|b\|^2$; (g) uses line 27 of Algorithm 1, the definition of $H_{\Sigma}^{k}$ in equation (172), and the definition of $\Delta_{Q_k}^{k;t_1,\ldots,t_k}$ on line 26 of Algorithm 1; (h) uses the $M_k^{k;t_1,\ldots,t_k}$-Lipszhitzness of operator $Q_k(z)$, which is implied by Lemma K.3; (i) uses the definition of $H_k^{k;t_1,\ldots,t_k}$ on line 20 of Algorithm 1. Furthermore, we obtain the following:

$$0 \overset{(a)}{\geq} \sum_{i=k+1}^{n} \left( D_{i;t_k+1}^{k;t_1,\ldots,t_k} - D_{i;t_k}^{k;t_1,\ldots,t_k} \right) + \frac{H_{\Sigma}^{k-1}}{2} \| z_{t_1,\ldots,t_{k-1},t_k+1/2}^{k} - \hat{z} \|^2$$

$$+ \sum_{i=1}^{k-1} \left( p_i^{k-1;t_1,\ldots,t_{k-1}}(z_{t_1,\ldots,t_{k-1},t_k+1/2}^{k}) - p_i^{k-1;t_1,\ldots,t_{k-1}}(\hat{z}) \right)$$

$$+ \frac{1}{\alpha_{t_k}} \sum_{i=k}^{n} p_i^{k-1;t_1,\ldots,t_{k-1}}(\alpha_{t_k} z_{t_1,\ldots,t_{k-1},t_k+1/2}^{k} + (1 - \alpha_{t_k})\overline{z}_{t_k}^{k})$$

$$- \frac{1}{\alpha_{t_k}} \sum_{i=k}^{n} p_i^{k-1;t_1,\ldots,t_{k-1}}(\alpha_{t_k}\hat{z} + (1 - \alpha_{t_k})\overline{z}_{t_k}^{k}) + \sum_{i=k}^{n} \langle Q_i(\hat{z}), z_{t_1,\ldots,t_{k-1},t_k+1/2}^{k} - \hat{z} \rangle$$

$$- \frac{H_k^{k;t_1,\ldots,t_k}}{2} \| \hat{z} - z_{t_1,\ldots,t_k}^{k} \|^2 + \frac{H_k^{k;t_1,\ldots,t_k}}{2} \| z_{t_1,\ldots,t_{k-1},t_k+1}^{k} - \hat{z} \|^2$$

$$\overset{(b)}{\geq} \sum_{i=k+1}^{n} \left( D_{i;t_k+1}^{k;t_1,\ldots,t_k} - D_{i;t_k}^{k;t_1,\ldots,t_k} \right) + \frac{H_{\Sigma}^{k-1}}{2} \| z_{t_1,\ldots,t_{k-1},t_k+1/2}^{k} - \hat{z} \|^2$$

$$+ \sum_{i=1}^{k-1} \left( p_i^{k-1;t_1,\ldots,t_{k-1}}(z_{t_1,\ldots,t_{k-1},t_k+1/2}^{k}) - p_i^{k-1;t_1,\ldots,t_{k-1}}(\hat{z}) \right)$$

$$+ \frac{1}{\alpha_{t_k}} \sum_{i=k}^{n} p_i^{k-1;t_1,\ldots,t_{k-1}}(\overline{z}_{t_k+1}^{k}) - \frac{1 - \alpha_{t_k}}{\alpha_{t_k}} \sum_{i=k}^{n} p_i^{k-1;t_1,\ldots,t_{k-1}}(\overline{z}_{t_k}^{k})$$

$$- \sum_{i=k}^{n} p_i^{k-1;t_1,\ldots,t_{k-1}}(\hat{z}) + \sum_{i=k}^{n} \langle Q_i(\hat{z}), z_{t_1,\ldots,t_{k-1},t_k+1/2}^{k} - \hat{z} \rangle$$

$$- \frac{H_k^{k;t_1,\ldots,t_k}}{2} \| \hat{z} - z_{t_1,\ldots,t_k}^{k} \|^2 + \frac{H_k^{k;t_1,\ldots,t_k}}{2} \| z_{t_1,\ldots,t_{k-1},t_k+1}^{k} - \hat{z} \|^2$$

$$\overset{(c)}{=} \sum_{i=k+1}^{n} \left( D_{i;t_k+1}^{k;t_1,\ldots,t_k} - D_{i;t_k}^{k;t_1,\ldots,t_k} \right) + \frac{H_{\Sigma}^{k-1}}{2} \| \alpha_{t_k}^{-1}\overline{z}_{t_k+1}^{k} - (1 - \alpha_{t_k})\alpha_{t_k}^{-1}\overline{z}_{t_k}^{k} - \hat{z} \|^2$$

$$+ \sum_{i=1}^{k-1} p_i^{k-1;t_1,\ldots,t_{k-1}}(\alpha_{t_k}^{-1}\overline{z}_{t_k+1}^{k} - (1 - \alpha_{t_k})\alpha_{t_k}^{-1}\overline{z}_{t_k}^{k})$$

$$+ \sum_{i=k}^{n} \langle Q_i(\hat{z}), \alpha_{t_k}^{-1}\overline{z}_{t_k+1}^{k} - (1 - \alpha_{t_k})\alpha_{t_k}^{-1}\overline{z}_{t_k}^{k} - \hat{z} \rangle$$

$$+ \frac{1}{\alpha_{t_k}} \sum_{i=k}^{n} p_i^{k-1;t_1,\ldots,t_{k-1}}(\overline{z}_{t_k+1}^k) - \frac{1-\alpha_{t_k}}{\alpha_{t_k}} \sum_{i=k}^{n} p_i^{k-1;t_1,\ldots,t_{k-1}}(\overline{z}_{t_k}^k) - \sum_{i=1}^{n} p_i^{k-1;t_1,\ldots,t_{k-1}}(\hat{z})$$

$$- \frac{H_k^{k;t_1,\ldots,t_k}}{2} \|\hat{z} - z_{t_1,\ldots,t_k}^k\|^2 + \frac{H_k^{k;t_1,\ldots,t_k}}{2} \|z_{t_1,\ldots,t_{k-1},t_k+1}^k - \hat{z}\|^2$$

$$\overset{(d)}{\geq} \sum_{i=k+1}^{n} \left( D_{i;t_k+1}^{k;t_1,\ldots,t_k} - D_{i;t_k}^{k;t_1,\ldots,t_k} \right)$$

$$+ \frac{1}{\alpha_{t_k}} \left( \frac{H_\Sigma^{k-1}}{2} \|\overline{z}_{t_k+1}^k - \hat{z}\|^2 + \sum_{i=k}^{n} \langle Q_i(\hat{z}), \overline{z}_{t_k+1}^k - \hat{z} \rangle \right)$$

$$- \frac{1-\alpha_{t_k}}{\alpha_{t_k}} \left( \frac{H_\Sigma^{k-1}}{2} \|\overline{z}_{t_k}^k - \hat{z}\|^2 + \sum_{i=k}^{n} \langle Q_i(\hat{z}), \overline{z}_{t_k}^k - \hat{z} \rangle \right)$$

$$+ \frac{1}{\alpha_{t_k}} \sum_{i=1}^{n} p_i^{k-1;t_1,\ldots,t_{k-1}}(\overline{z}_{t_k+1}^k) - \frac{1-\alpha_{t_k}}{\alpha_{t_k}} \sum_{i=1}^{n} p_i^{k-1;t_1,\ldots,t_{k-1}}(\overline{z}_{t_k}^k) - \sum_{i=1}^{n} p_i^{k-1;t_1,\ldots,t_{k-1}}(\hat{z})$$

$$- \frac{H_k^{k;t_1,\ldots,t_k}}{2} \|\hat{z} - z_{t_1,\ldots,t_k}^k\|^2 + \frac{H_k^{k;t_1,\ldots,t_k}}{2} \|z_{t_1,\ldots,t_{k-1},t_k+1}^k - \hat{z}\|^2$$

$$\overset{(e)}{=} \frac{1}{\alpha_{t_k}} \left( \frac{H_\Sigma^{k-1}}{2} \|\overline{z}_{t_k+1}^k - \hat{z}\|^2 + \sum_{i=1}^{n} \left( r_i^{k-1;t_1,\ldots,t_{k-1}}(\overline{z}_{t_k+1}^k) - r_i^{k-1;t_1,\ldots,t_{k-1}}(\hat{z}) \right) \right)$$

$$- \frac{1-\alpha_{t_k}}{\alpha_{t_k}} \left( \frac{H_\Sigma^{k-1}}{2} \|\overline{z}_{t_k}^k - \hat{z}\|^2 + \sum_{i=1}^{n} \left( r_i^{k-1;t_1,\ldots,t_{k-1}}(\overline{z}_{t_k}^k) - r_i^{k-1;t_1,\ldots,t_{k-1}}(\hat{z}) \right) \right)$$

$$- \frac{H_k^{k;t_1,\ldots,t_k}}{2} \|\hat{z} - z_{t_1,\ldots,t_k}^k\|^2 + \frac{H_k^{k;t_1,\ldots,t_k}}{2} \|z_{t_1,\ldots,t_{k-1},t_k+1}^k - \hat{z}\|^2$$

$$+ \sum_{i=k+1}^{n} \left( D_{i;t_k+1}^{k;t_1,\ldots,t_k} - D_{i;t_k}^{k;t_1,\ldots,t_k} \right),$$

where (a) uses the previous inequality and the definition of function $\hat{p}_i^{k;t_1,\ldots,t_k}(z)$ on line 17 of Algorithm 1; (b) uses the definition of $\overline{z}_{t_k+1}^k$ on line 25 of Algorithm 1 and the convexity of functions $p_i^{k-1;t_1,\ldots,t_{k-1}}(z)$; (c) uses the definition of $\overline{z}_{t_k+1}^k$ on line 25 of Algorithm 1; (d) uses the convexity of function $p_i^{k-1;t_1,\ldots,t_{k-1}}(z)$ and $\|\cdot\|^2$; (e) uses the definition of functions $r_i^{k-1;t_1,\ldots,t_{k-1}}(z)$ in equation (171). Next, we divide both sides of the inequality by $\alpha_{t_k}$ and obtain the following:

$$0 \geq \frac{1}{\alpha_{t_k}^2} \left( \frac{H_\Sigma^{k-1}}{2} \|\overline{z}_{t_k+1}^k - \hat{z}\|^2 + \sum_{i=1}^{n} \left( r_i^{k-1;t_1,\ldots,t_{k-1}}(\overline{z}_{t_k+1}^k) - r_i^{k-1;t_1,\ldots,t_{k-1}}(\hat{z}) \right) \right)$$

$$- \frac{1-\alpha_{t_k}}{\alpha_{t_k}^2} \left( \frac{H_\Sigma^{k-1}}{2} \|\overline{z}_{t_k}^k - \hat{z}\|^2 + \sum_{i=1}^{n} \left( r_i^{k-1;t_1,\ldots,t_{k-1}}(\overline{z}_{t_k}^k) - r_i^{k-1;t_1,\ldots,t_{k-1}}(\hat{z}) \right) \right)$$

$$- \frac{H_k^{k;t_1,\ldots,t_k}}{2\alpha_{t_k}} \|\hat{z} - z_{t_1,\ldots,t_k}^k\|^2 + \frac{H_k^{k;t_1,\ldots,t_k}}{2\alpha_{t_k}} \|z_{t_1,\ldots,t_{k-1},t_k+1}^k - \hat{z}\|^2$$

$$+ \sum_{i=k+1}^{n} \frac{1}{\alpha_{t_k}} \left( D_{i;t_k+1}^{k;t_1,\ldots,t_k} - D_{i;t_k}^{k;t_1,\ldots,t_k} \right)$$

$$\overset{(a)}{=} \frac{1}{\alpha_{t_k}^2} \left( \frac{H_\Sigma^{k-1}}{2} \|\overline{z}_{t_k+1}^k - \hat{z}\|^2 + \sum_{i=1}^{n} \left( r_i^{k-1;t_1,\ldots,t_{k-1}}(\overline{z}_{t_k+1}^k) - r_i^{k-1;t_1,\ldots,t_{k-1}}(\hat{z}) \right) \right)$$

$$- \frac{1-\alpha_{t_k}}{\alpha_{t_k}^2} \left( \frac{H_\Sigma^{k-1}}{2} \|\overline{z}_{t_k}^k - \hat{z}\|^2 + \sum_{i=1}^{n} \left( r_i^{k-1;t_1,\ldots,t_{k-1}}(\overline{z}_{t_k}^k) - r_i^{k-1;t_1,\ldots,t_{k-1}}(\hat{z}) \right) \right)$$

$$- \frac{L_k^{k;t_1,\ldots,t_k} + M_k^{k;t_1,\ldots,t_k}}{2\alpha_{t_k}} \|\hat{z} - z_{t_1,\ldots,t_k}^k\|^2 + \frac{L_k^{k;t_1,\ldots,t_k} + M_k^{k;t_1,\ldots,t_k}}{2\alpha_{t_k}} \|z_{t_1,\ldots,t_{k-1},t_k+1}^k - \hat{z}\|^2$$

$$+ \sum_{i=k+1}^{n} \frac{1}{\alpha_{t_k}} \left( D_{i;t_k+1}^{k;t_1,\ldots,t_k} - D_{i;t_k}^{k;t_1,\ldots,t_k} \right)$$

$$\overset{(b)}{=} \frac{1}{\alpha_{t_k}^2} \left( \frac{H_{\Sigma}^{k-1}}{2} \|\overline{z}_{t_k+1}^k - \hat{z}\|^2 + \sum_{i=1}^{n} \left( r_i^{k-1;t_1,\ldots,t_{k-1}}(\overline{z}_{t_k+1}^k) - r_i^{k-1;t_1,\ldots,t_{k-1}}(\hat{z}) \right) \right)$$

$$- \frac{1-\alpha_{t_k}}{\alpha_{t_k}^2} \left( \frac{H_{\Sigma}^{k-1}}{2} \|\overline{z}_{t_k}^k - \hat{z}\|^2 + \sum_{i=1}^{n} \left( r_i^{k-1;t_1,\ldots,t_{k-1}}(\overline{z}_{t_k}^k) - r_i^{k-1;t_1,\ldots,t_{k-1}}(\hat{z}) \right) \right)$$

$$+ \frac{L_k^{k-1;t_1,\ldots,t_{k-1}}}{2} \left( \|z_{t_1,\ldots,t_{k-1},t_k+1}^k - \hat{z}\|^2 - \|z_{t_1,\ldots,t_k}^k - \hat{z}\|^2 \right)$$

$$+ \frac{M_k^{k-1;t_1,\ldots,t_{k-1}}}{2\alpha_{T_k-1}} \left( \|z_{t_1,\ldots,t_{k-1},t_k+1}^k - \hat{z}\|^2 - \|z_{t_1,\ldots,t_k}^k - \hat{z}\|^2 \right)$$

$$+ \sum_{i=k+1}^{n} \frac{1}{\alpha_{t_k}} \left( D_{i;t_k+1}^{k;t_1,\ldots,t_k} - D_{i;t_k}^{k;t_1,\ldots,t_k} \right),$$

where (a) uses the definition of $H_k^{k;t_1,\ldots,t_k}$ on line 20 of Algorithm 1; (b) uses the definitions of $L_k^{k;t_1,\ldots,t_k}$ and $M_k^{k;t_1,\ldots,t_k}$ on lines 18 and 19 of Algorithm 1. Using the definition of $D_{i;t}^{k;t_1,\ldots,t_k}$ in equation (173), one can verify that $\frac{1}{\alpha_{t_k}} D_{i;t}^{k;t_1,\ldots,t_k}$ does not depend on $t_k$. Moreover, using the definition of $\alpha_{t_k}$ in equation (20), one can verify that $\alpha_0 = 1$ and $\frac{1-\alpha_{t_k}}{\alpha_{t_k}^2} = \frac{1}{\alpha_{t_k-1}^2}$ for $t_k \geq 1$. Hence, we can do the telescoping and obtain for arbitrary $t_k \in \{0,\ldots,T_k-1\}$ the following:

$$0 \geq \frac{1}{\alpha_{T_k-1}^2} \left( \frac{H_{\Sigma}^{k-1}}{2} \|\overline{z}_{T_k}^k - \hat{z}\|^2 + \sum_{i=1}^{n} \left( r_i^{k-1;t_1,\ldots,t_{k-1}}(\overline{z}_{T_k}^k) - r_i^{k-1;t_1,\ldots,t_{k-1}}(\hat{z}) \right) \right)$$

$$+ \frac{L_k^{k-1;t_1,\ldots,t_{k-1}}}{2} \left( \|z_{t_1,\ldots,t_{k-1},T_k}^k - \hat{z}\|^2 - \|z_{t_1,\ldots,t_{k-1},0}^k - \hat{z}\|^2 \right)$$

$$+ \frac{M_k^{k-1;t_1,\ldots,t_{k-1}}}{2\alpha_{T_k-1}} \left( \|z_{t_1,\ldots,t_{k-1},T_k}^k - \hat{z}\|^2 - \|z_{t_1,\ldots,t_{k-1},0}^k - \hat{z}\|^2 \right)$$

$$+ \sum_{i=k+1}^{n} \frac{1}{\alpha_{t_k}} \left( D_{i;T_k}^{k;t_1,\ldots,t_k} - D_{i;0}^{k;t_1,\ldots,t_k} \right).$$

After multiplying both sides of the inequality by $\alpha_{T_k-1}^2$, we obtain for arbitrary $t_k \in \{0,\ldots,T_k-1\}$ the following:

$$0 \geq \sum_{i=1}^{n} \left( r_i^{k-1;t_1,\ldots,t_{k-1}}(\overline{z}_{T_k}^k) - r_i^{k-1;t_1,\ldots,t_{k-1}}(\hat{z}) \right) + \frac{H_{\Sigma}^{k-1}}{2} \|\overline{z}_{T_k}^k - \hat{z}\|^2$$

$$+ \frac{L_k^{k-1;t_1,\ldots,t_{k-1}} \cdot \alpha_{T_k-1}^2}{2} \left( \|z_{t_1,\ldots,t_{k-1},T_k}^k - \hat{z}\|^2 - \|z_{t_1,\ldots,t_{k-1},0}^k - \hat{z}\|^2 \right)$$

$$+ \frac{M_k^{k-1;t_1,\ldots,t_{k-1}} \cdot \alpha_{T_k-1}}{2} \left( \|z_{t_1,\ldots,t_{k-1},T_k}^k - \hat{z}\|^2 - \|z_{t_1,\ldots,t_{k-1},0}^k - \hat{z}\|^2 \right)$$

$$+ \sum_{i=k+1}^{n} \frac{\alpha_{T_k-1}^2}{\alpha_{t_k}} \left( D_{i;T_k}^{k;t_1,\ldots,t_k} - D_{i;0}^{k;t_1,\ldots,t_k} \right)$$

$$\overset{(a)}{=} \sum_{i=1}^{n} \left( r_i^{k-1;t_1,\ldots,t_{k-1}}(\overline{z}_{T_k}^k) - r_i^{k-1;t_1,\ldots,t_{k-1}}(\hat{z}) \right) + \frac{H_{\Sigma}^{k-1}}{2} \|\overline{z}_{T_k}^k - \hat{z}\|^2$$

$$+ \frac{L_k^{k-1;t_1,\ldots,t_{k-1}} \cdot \alpha_{T_k-1}^2}{2} \left( \|z_{t_1,\ldots,t_{k-1},T_k}^k - \hat{z}\|^2 - \|z_{t_1,\ldots,t_{k-1},0}^k - \hat{z}\|^2 \right)$$

$$+ \frac{M_k^{k-1;t_1,\ldots,t_{k-1}} \cdot \alpha_{T_k-1}}{2} \left( \|z_{t_1,\ldots,t_{k-1},T_k}^k - \hat{z}\|^2 - \|z_{t_1,\ldots,t_{k-1},0}^k - \hat{z}\|^2 \right)$$

$$+ \frac{\alpha_{T_k-1}^2}{\alpha_{t_k}} \sum_{i=k+1}^n \frac{L_i^{k;t_1\ldots,t_k} \prod_{j=k+1}^i \alpha_{T_j-1}^2 + M_i^{k;t_1\ldots,t_k} \prod_{j=k+1}^i \alpha_{T_j-1}}{2} \|z_{t_1,\ldots,t_{k-1},T_k,0\ldots0}^i - \hat{z}\|^2$$

$$- \frac{\alpha_{T_k-1}^2}{\alpha_{t_k}} \sum_{i=k+1}^n \frac{L_i^{k;t_1\ldots,t_k} \prod_{j=k+1}^i \alpha_{T_j-1}^2 + M_i^{k;t_1\ldots,t_k} \prod_{j=k+1}^i \alpha_{T_j-1}}{2} \|z_{t_1,\ldots,t_{k-1},0\ldots0}^i - \hat{z}\|^2$$

$$\overset{(b)}{=} \sum_{i=1}^n \left( r_i^{k-1;t_1,\ldots,t_{k-1}}(\overline{z}_{T_k}^k) - r_i^{k-1;t_1,\ldots,t_{k-1}}(\hat{z}) \right) + \frac{H_\Sigma^{k-1}}{2} \|\overline{z}_{T_k}^k - \hat{z}\|^2$$

$$+ \frac{L_k^{k-1;t_1,\ldots,t_{k-1}} \cdot \alpha_{T_k-1}^2}{2} \left( \|z_{t_1,\ldots,t_{k-1},T_k}^k - \hat{z}\|^2 - \|z_{t_1,\ldots,t_{k-1},0}^k - \hat{z}\|^2 \right)$$

$$+ \frac{M_k^{k-1;t_1,\ldots,t_{k-1}} \cdot \alpha_{T_k-1}}{2} \left( \|z_{t_1,\ldots,t_{k-1},T_k}^k - \hat{z}\|^2 - \|z_{t_1,\ldots,t_{k-1},0}^k - \hat{z}\|^2 \right)$$

$$+ \sum_{i=k+1}^n \frac{L_i^{k-1;t_1\ldots,t_{k-1}} \prod_{j=k}^i \alpha_{T_j-1}^2 + M_i^{k-1;t_1\ldots,t_{k-1}} \prod_{j=k}^i \alpha_{T_j-1}}{2} \|z_{t_1,\ldots,t_{k-1},T_k,0\ldots0}^i - \hat{z}\|^2$$

$$- \sum_{i=k+1}^n \frac{L_i^{k-1;t_1\ldots,t_{k-1}} \prod_{j=k}^i \alpha_{T_j-1}^2 + M_i^{k-1;t_1\ldots,t_{k-1}} \prod_{j=k}^i \alpha_{T_j-1}}{2} \|z_{t_1,\ldots,t_{k-1},0\ldots0}^i - \hat{z}\|^2$$

$$\overset{(c)}{=} \sum_{i=1}^n \left( r_i^{k-1;t_1,\ldots,t_{k-1}}(\overline{z}_{T_k}^k) - r_i^{k-1;t_1,\ldots,t_{k-1}}(\hat{z}) \right) + \frac{H_\Sigma^{k-1}}{2} \|\overline{z}_{T_k}^k - \hat{z}\|^2$$

$$+ \sum_{i=k}^n \frac{L_i^{k-1;t_1\ldots,t_{k-1}} \prod_{j=k}^i \alpha_{T_j-1}^2 + M_i^{k-1;t_1\ldots,t_{k-1}} \prod_{j=k}^i \alpha_{T_j-1}}{2} \|z_{t_1,\ldots,t_{k-2},t_{k-1}+1,0\ldots0}^i - \hat{z}\|^2$$

$$- \sum_{i=k}^n \frac{L_i^{k-1;t_1\ldots,t_{k-1}} \prod_{j=k}^i \alpha_{T_j-1}^2 + M_i^{k-1;t_1\ldots,t_{k-1}} \prod_{j=k}^i \alpha_{T_j-1}}{2} \|z_{t_1,\ldots,t_{k-1},0\ldots0}^i - \hat{z}\|^2$$

$$\overset{(d)}{=} \sum_{i=1}^n \left( r_i^{k-1;t_1,\ldots,t_{k-1}}(\overline{z}_{T_k}^k) - r_i^{k-1;t_1,\ldots,t_{k-1}}(\hat{z}) \right) + \frac{H_\Sigma^{k-1}}{2} \|\overline{z}_{T_k}^k - \hat{z}\|^2$$

$$+ \sum_{i=k}^n \left( D_{i;t_{k-1}+1}^{k-1;t_1,\ldots,t_{k-1}} - D_{i;t_{k-1}}^{k-1;t_1,\ldots,t_{k-1}} \right)$$

$$\overset{(e)}{=} \sum_{i=1}^n \left( r_i^{k-1;t_1,\ldots,t_{k-1}}(z_{t_1,\ldots,t_{k-2},t_{k-1}+1/2}^{k-1}) - r_i^{k-1;t_1,\ldots,t_{k-1}}(\hat{z}) \right)$$

$$+ \frac{H_\Sigma^{k-1}}{2} \|z_{t_1,\ldots,t_{k-2},t_{k-1}+1/2}^{k-1} - \hat{z}\|^2 + \sum_{i=k}^n \left( D_{i;t_{k-1}+1}^{k-1;t_1,\ldots,t_{k-1}} - D_{i;t_{k-1}}^{k-1;t_1,\ldots,t_{k-1}} \right),$$

where (a) and (d) use the definition of $D_{i;t}^{k;t_1,\ldots,t_k}$ in equation (173); (b) uses the definitions of $L_i^{k;t_1,\ldots,t_k}$ and $M_i^{k;t_1,\ldots,t_k}$ on lines 18 and 19 of Algorithm 1; (c) uses line 29 of Algorithm 1; (e) uses lines 24 and 32 of Algorithm 1. The latter inequality is nothing else but the desired equation (170) for $k-1$, which concludes the proof. $\square$

K.1.2. PROOF OF LEMMA K.5

We prove by induction that for all $1 \leq k \leq n$, the following inclusion holds:

$$z^k_{t_1, \ldots, t_{k-1}, t_k + 1/2} \in \mathcal{C}_z. \tag{179}$$

Indeed, in the base case $k = n$, equation (179) holds due to lines 12 and 24 of Algorithm 1. Next, we assume the inclusion (179) for a fixed $k$ satisfying $1 \leq k \leq n$. We have $\overline{z}^k_1 = z^k_{t_1, \ldots, t_{k-1}, 1/2} \in \mathcal{C}_z$ due to the definition on line 25 of Algorithm 1 and the fact that $\alpha_0 = 1$, which is implied by equation (20). Furthermore, for $t_k \geq 1$, we have $\overline{z}^k_{t_k+1} \in \mathcal{C}_z$ due to the definition on line 25 of Algorithm 1 and the inclusions $\overline{z}^k_{t_k}, z^k_{t_1, \ldots, t_{k-1}, t_k + 1/2} \in \mathcal{C}_z$, and $\alpha_{t_k} \in (0, 1)$. Hence, we obtain $\overline{z}^k_{T_k} \in \mathcal{C}_z$. This, together with lines 24 and 32 of Algorithm 1, implies equation (179) for $k - 1$, if $k \geq 2$, or $z_{\text{out}} \in \mathcal{C}_z$, if $k = 1$, which concludes the proof. $\square$

# L. Proof of Corollary 4.6

Without loss of generality, we can assume that

$$\max\left\{ \sqrt{L_{i+1}/\epsilon}, M_{i+1}/\epsilon, 1 \right\} \geq \max\left\{ \sqrt{L_i/\epsilon}, M_i/\epsilon, 1 \right\} \quad \text{for all} \quad i \in \{1, \ldots, n-1\}. \tag{180}$$

Otherwise, we can simply reshuffle the pairs $(p_i(z), Q_i(z))$. To ensure the desired inequality (22), it is sufficient to choose $\{T_i\}_{i=1}^n$ as follows:

$$T_1 = \left\lceil 2 \cdot \max\left\{ \sqrt{L_1/\epsilon}, M_1/\epsilon, 1 \right\} \right\rceil,$$

$$T_{i+1} = \left\lceil 2 \cdot \frac{\max\left\{ \sqrt{L_{i+1}/\epsilon}, M_{i+1}/\epsilon, 1 \right\}}{\max\left\{ \sqrt{L_i/\epsilon}, M_i/\epsilon, 1 \right\}} \right\rceil \quad \text{for} \quad i \in \{1, \ldots, n-1\}. \tag{181}$$

Indeed, we can lower-bound $\prod_{j=1}^i T_j$ as follows:

$$\prod_{j=1}^i T_j = T_1 \prod_{j=2}^i T_j \geq 2 \max\left\{ \sqrt{L_1/\epsilon}, M_1/\epsilon, 1 \right\} \prod_{j=1}^{i-1} \frac{2 \max\left\{ \sqrt{L_{j+1}/\epsilon}, M_{j+1}/\epsilon, 1 \right\}}{\max\left\{ \sqrt{L_j/\epsilon}, M_j/\epsilon, 1 \right\}}$$

$$= 2^i \max\left\{ \sqrt{L_i/\epsilon}, M_i/\epsilon, 1 \right\},$$

which, together with equation (21), implies equation (22). Similarly, we can upper-bound $\prod_{j=1}^i T_j$ as follows:

$$\prod_{j=1}^i T_j = T_1 \prod_{j=2}^i T_j$$

$$\leq \left( 2 \max\left\{ \sqrt{L_1/\epsilon}, M_1/\epsilon, 1 \right\} + 1 \right) \prod_{j=1}^{i-1} \left( \frac{2 \max\left\{ \sqrt{L_{j+1}/\epsilon}, M_{j+1}/\epsilon, 1 \right\}}{\max\left\{ \sqrt{L_j/\epsilon}, M_j/\epsilon, 1 \right\}} + 1 \right)$$

$$\leq 3 \max\left\{ \sqrt{L_1/\epsilon}, M_1/\epsilon, 1 \right\} \prod_{j=1}^{i-1} \frac{3 \max\left\{ \sqrt{L_{j+1}/\epsilon}, M_{j+1}/\epsilon, 1 \right\}}{\max\left\{ \sqrt{L_j/\epsilon}, M_j/\epsilon, 1 \right\}}$$

$$= 3^i \max\left\{ \sqrt{L_i/\epsilon}, M_i/\epsilon, 1 \right\}$$

$$\leq 3^n \max\left\{ \sqrt{L_i/\epsilon}, M_i/\epsilon, 1 \right\},$$

Finally, it is easy to verify that Algorithm 1 performs $\prod_{j=1}^i T_j$ computations of the gradient $\nabla p_i(z)$ on line 21 and $2^i \cdot \prod_{j=1}^i T_j$ computations of operator $Q_i(z)$ on lines 21 and 26, which concludes the proof. $\square$

## M. Proof of Theorem 4.7

Let $z = (x, y) \in \mathcal{Z} = \mathcal{X} \times \mathcal{Y}$ and $z' = (x', y') \in \mathcal{Z} = \mathcal{X} \times \mathcal{Y}$. Then we can upper-bound $\|\nabla p_1(z) - \nabla p_1(z')\|_{\mathbf{P}^{-1}}$ as follows:

$$\|\nabla p_1(z) - \nabla p_1(z')\|_{\mathbf{P}^{-1}} \overset{(a)}{=} \delta_x^{-1/2} \cdot \|\nabla f(x) - \nabla f(x')\|$$
$$\overset{(b)}{\leq} \delta_x^{-1/2} \cdot L_x \|x - x'\| \overset{(c)}{=} \frac{L_x}{\delta_x} \|x - x'\|_{\mathbf{P}} \overset{(d)}{=} \kappa_x \|x - x'\|_{\mathbf{P}},$$

where (a) uses the definition of function $p_1(z)$ in equation (26) and the definition of matrix $\mathbf{P}$ in equation (24); (b) uses the smoothness property in Assumption 2.3; (c) uses the definition of matrix $\mathbf{P}$ in equation (24); (d) uses the definition of $\kappa_x$ in equation (6). Hence, we can choose $L_1 = \kappa_x$, and similarly, we can choose $L_2 = \kappa_y$. In addition, we can upper-bound $\|\nabla p_3(z) - \nabla p_3(z')\|_{\mathbf{P}^{-1}}$ as follows:

$$\|\nabla p_3(z) - \nabla p_3(z')\|_{\mathbf{P}^{-1}} \overset{(a)}{=} \sqrt{\frac{\beta_x^2}{\delta_x}\|\mathbf{B}^\top \mathbf{B}(x - x')\|^2 + \frac{\beta_y^2}{\delta_y}\|\mathbf{B}\mathbf{B}^\top(y - y')\|^2}$$
$$\overset{(b)}{\leq} \sqrt{\frac{\beta_x^2 L_{xy}^4}{\delta_x}\|x - x'\|^2 + \frac{\beta_y^2 L_{xy}^4}{\delta_y}\|y - y'\|^2}$$
$$\overset{(c)}{\leq} \max\left\{\frac{\beta_x L_{xy}^2}{\delta_x}, \frac{\beta_y L_{xy}^2}{\delta_y}\right\} \|z - z'\|_{\mathbf{P}}$$
$$\overset{(d)}{=} \kappa_{xy} \cdot \max\{\beta_x \delta_y, \beta_y \delta_x\} \|z - z'\|_{\mathbf{P}}$$
$$\overset{(e)}{\leq} \kappa_{xy} \cdot \max\left\{\frac{\delta_y}{4L_y}, \frac{\delta_x}{4L_x}\right\} \|z - z'\|_{\mathbf{P}}$$
$$\overset{(f)}{=} \kappa_{xy} \cdot \max\left\{\frac{1}{4\kappa_x}, \frac{1}{4\kappa_y}\right\} \|z - z'\|_{\mathbf{P}}$$
$$\overset{(g)}{\leq} \tfrac{1}{4}\kappa_{xy} \|z - z'\|_{\mathbf{P}},$$

where (a) uses the definition of function $p_3(z)$ in equation (26) and the definition of matrix $\mathbf{P}$ in equation (24); (b) uses Assumption 2.5; (c) uses the definition of matrix $\mathbf{P}$ in equation (24); (d) uses the definition of $\kappa_{xy}$ in equation (6); (e) uses the definitions of $\beta_x$ and $\beta_y$ in equation (27); (f) uses the definitions of $\kappa_x$ and $\kappa_y$ in equation (6); (g) uses the fact that $\kappa_x, \kappa_y \geq 1$, which is implied by Assumption 2.7. Hence, we can choose $L_3 = \kappa_{xy}$. Furthermore, we can upper-bound $\|Q_3(z) - Q_3(z')\|_{\mathbf{P}^{-1}}$ as follows:

$$\|Q_3(z) - Q_3(z')\|_{\mathbf{P}^{-1}} \overset{(a)}{=} \left\|\begin{bmatrix} \mathbf{O}_{d_x} & \mathbf{B}^\top \\ -\mathbf{B} & \mathbf{O}_{d_y} \end{bmatrix}\begin{bmatrix} x - x' \\ y - y' \end{bmatrix}\right\|_{\mathbf{P}^{-1}}$$
$$\overset{(b)}{=} \sqrt{\delta_x^{-1}\|\mathbf{B}^\top(y - y')\|^2 + \delta_y^{-1}\|\mathbf{B}(x - x')\|^2}$$
$$\overset{(c)}{\leq} \sqrt{\frac{L_{xy}^2}{\delta_x}\|y - y'\|^2 + \frac{L_{xy}^2}{\delta_y}\|x - x'\|^2}$$
$$\overset{(d)}{=} \sqrt{\kappa_{xy}\delta_y\|y - y'\|^2 + \kappa_{xy}\delta_x\|x - x'\|^2}$$
$$\overset{(e)}{=} \sqrt{\kappa_{xy}}\|z - z'\|_{\mathbf{P}},$$

where (a) uses the definition of operator $Q_3(z)$ in equation (25); (b) and (e) use the definition of matrix $\mathbf{P}$ in equation (24); (c) uses Assumption 2.5; (d) uses the definition of $\kappa_{xy}$ in equation (6). Hence, we can choose $M_3 = \sqrt{\kappa_{xy}}$.

Next, we can obtain the following:

$$\epsilon n\|z_{\text{in}} - z^*\|_{\mathbf{P}}^2 \overset{(a)}{\geq} p(z_{\text{out}}) - p(z^*) + \langle Q(z^*), z_{\text{out}} - z^* \rangle$$
$$\overset{(b)}{=} f(x_{\text{out}}) - f(x^*) + g(y_{\text{out}}) - g(y^*) + \langle \mathbf{B}^\top y^*, x_{\text{out}} - x^* \rangle - \langle \mathbf{B}x^*, y_{\text{out}} - y^* \rangle$$

$$+ \frac{\beta_x}{2}\|\mathbf{B}x_{\text{out}} - \nabla g(y_{\text{in}})\|^2 + \frac{\beta_y}{2}\|\mathbf{B}^\top y_{\text{out}} + \nabla f(x_{\text{in}})\|^2$$

$$- \frac{\beta_x}{2}\|\mathbf{B}x^* - \nabla g(y_{\text{in}})\|^2 - \frac{\beta_y}{2}\|\mathbf{B}^\top y^* + \nabla f(x_{\text{in}})\|^2$$

$$\overset{\text{(c)}}{\geq} f(x_{\text{out}}) - f(x^*) + g(y_{\text{out}}) - g(y^*) + \langle \mathbf{B}^\top y^*, x_{\text{out}} - x^*\rangle - \langle \mathbf{B}x^*, y_{\text{out}} - y^*\rangle$$

$$+ \frac{\beta_x}{4}\|\mathbf{B}(x_{\text{out}} - x^*)\|^2 + \frac{\beta_y}{4}\|\mathbf{B}^\top(y_{\text{out}} - y^*)\|^2$$

$$- \beta_x\|\mathbf{B}x^* - \nabla g(y_{\text{in}})\|^2 - \beta_y\|\mathbf{B}^\top y^* + \nabla f(x_{\text{in}})\|^2$$

$$\overset{\text{(d)}}{=} D_f(x_{\text{out}}, x^*) + D_g(y_{\text{out}}, y^*) + \frac{\beta_x}{4}\|\mathbf{B}(x_{\text{out}} - x^*)\|^2 + \frac{\beta_y}{4}\|\mathbf{B}^\top(y_{\text{out}} - y^*)\|^2$$

$$- \beta_x\|\nabla g(y^*) - \nabla g(y_{\text{in}})\|^2 - \beta_y\|\nabla f(x^*) - \nabla f(x_{\text{in}})\|^2$$

$$\overset{\text{(e)}}{\geq} D_f(x_{\text{out}}, x^*) + D_g(y_{\text{out}}, y^*) + \frac{\beta_x}{4}\|\mathbf{B}(x_{\text{out}} - x^*)\|^2 + \frac{\beta_y}{4}\|\mathbf{B}^\top(y_{\text{out}} - y^*)\|^2$$

$$- 2\beta_x L_y D_g(y_{\text{in}}, y^*) - 2\beta_y L_x D_f(x_{\text{in}}, x^*)$$

$$\overset{\text{(f)}}{\geq} D_f(x_{\text{out}}, x^*) + D_g(y_{\text{out}}, y^*) + \frac{\beta_x}{4}\|\mathbf{B}(x_{\text{out}} - x^*)\|^2 + \frac{\beta_y}{4}\|\mathbf{B}^\top(y_{\text{out}} - y^*)\|^2$$

$$- \tfrac{1}{2}D_f(x_{\text{in}}, x^*) - \tfrac{1}{2}D_g(y_{\text{in}}, y^*),$$

where (a) uses Corollary 4.6; (b) the definitions of functions $p_i(z)$ and operators $Q_i(z)$ in equations (25) and (26); (c) uses Young's inequality; (d) uses the optimality conditions (7); (e) uses the smoothness properties in Assumptions 2.3 and 2.4; (f) uses the definitions of $\beta_x$ and $\beta_y$ in equation (27).

Using the definitions of functions $p_i(z)$ and operators $Q_i(z)$ in equations (25) and (26), Assumptions 2.5 and 2.6, and Lemma 2.8, we can conclude that $\text{proj}_{\mathcal{S}}(z_{\text{in}}) = \text{proj}_{\mathcal{S}}(z_{\text{out}})$. Hence, we get the following:

$$\beta_x\|\mathbf{B}(x_{\text{out}} - x^*)\|^2 + \beta_y\|\mathbf{B}^\top(y_{\text{out}} - y^*)\|^2 \geq \beta_x\mu_{xy}^2\|x_{\text{out}} - x^*\|^2 + \beta_y\mu_{yx}^2\|y_{\text{out}} - y^*\|^2, \tag{182}$$

which implies the following:

$$\epsilon n\|z_{\text{in}} - z^*\|_{\mathbf{P}}^2 \geq D_f(x_{\text{out}}, x^*) + D_g(y_{\text{out}}, y^*) - \tfrac{1}{2}D_f(x_{\text{in}}, x^*) - \tfrac{1}{2}D_g(y_{\text{in}}, y^*)$$

$$+ \tfrac{1}{4}\beta_x\mu_{xy}^2\|x_{\text{out}} - x^*\|^2 + \tfrac{1}{4}\beta_y\mu_{yx}^2\|y_{\text{out}} - y^*\|^2$$

$$\overset{\text{(a)}}{\geq} \tfrac{3}{4}D_f(x_{\text{out}}, x^*) + \tfrac{3}{4}D_g(y_{\text{out}}, y^*) - \tfrac{1}{2}D_f(x_{\text{in}}, x^*) - \tfrac{1}{2}D_g(y_{\text{in}}, y^*)$$

$$+ \left(\tfrac{1}{8}\mu_x + \tfrac{1}{4}\beta_x\mu_{xy}^2\right)\|x_{\text{out}} - x^*\|^2 + \left(\tfrac{1}{8}\mu_y + \tfrac{1}{4}\beta_y\mu_{yx}^2\right)\|y_{\text{out}} - y^*\|^2$$

$$\overset{\text{(b)}}{\geq} \tfrac{3}{4}D_f(x_{\text{out}}, x^*) + \tfrac{3}{4}D_g(y_{\text{out}}, y^*) - \tfrac{1}{2}D_f(x_{\text{in}}, x^*) - \tfrac{1}{2}D_g(y_{\text{in}}, y^*)$$

$$+ \tfrac{1}{16}\delta_x\|x_{\text{out}} - x^*\|^2 + \tfrac{1}{16}\delta_y\|y_{\text{out}} - y^*\|^2$$

$$\overset{\text{(c)}}{=} \tfrac{3}{4}D_f(x_{\text{out}}, x^*) + \tfrac{3}{4}D_g(y_{\text{out}}, y^*) - \tfrac{1}{2}D_f(x_{\text{in}}, x^*) - \tfrac{1}{2}D_g(y_{\text{in}}, y^*)$$

$$+ \tfrac{1}{16}\|z_{\text{out}} - z^*\|_{\mathbf{P}}^2,$$

where (a) uses the strong convexity properties in Assumptions 2.3 and 2.4; (b) uses the definitions of $\beta_x$ and $\beta_y$ in equation (27) and the definitions of $\delta_x$ and $\delta_y$ in equation (4); (c) uses the definition of $\mathbf{P}$ in equation (24). Furthermore, we have $\|z_{\text{in}} - z^*\|_{\mathbf{P}}^2 = \mathcal{R}_{\delta_x\delta_y}^2(x_{\text{in}}, y_{\text{in}})$ and $\|z_{\text{out}} - z^*\|_{\mathbf{P}}^2 = \mathcal{R}_{\delta_x\delta_y}^2(x_{\text{out}}, y_{\text{out}})$ due to the definition of $z^*$. Hence, we obtain the following inequality:

$$\mathcal{R}_{\delta_x\delta_y}^2(x_{\text{out}}, y_{\text{out}}) + 12D_f(x_{\text{out}}, x^*) + 12D_g(y_{\text{out}}, y^*)$$
$$\leq 16\epsilon n\mathcal{R}_{\delta_x\delta_y}^2(x_{\text{in}}, y_{\text{in}}) + 8D_f(x_{\text{in}}, x^*) + 8D_g(y_{\text{in}}, y^*). \tag{183}$$

Choosing $\epsilon = \frac{1}{24n} = \frac{1}{72}$ concludes the proof. $\qquad\square$

## N. Proof of Corollary 4.8

We use a restarted version of Algorithm 1. That is, we apply Algorithm 1 $T$ times and use the output at each run as the input for the next run. Formally, by $z_{\text{in}}^t = (x_{\text{in}}^t, y_{\text{in}}^t)$ and $z_{\text{out}}^t = (x_{\text{out}}^t, y_{\text{out}}^t)$ we denote the input and the output of Algorithm 1 at $t$-th run, where $t \in \{0, \ldots, T-1\}$. Then we have $z_{\text{in}}^0 = 0$ and $z_{\text{in}}^{t+1} = z_{\text{out}}^t$ for all $t \in \{0, \ldots, T-1\}$. Hence, we can upper-bound $\mathcal{R}_{\delta_x \delta_y}^2(x_{\text{in}}^T, y_{\text{in}}^T)$ as follows:

$$
\begin{aligned}
\mathcal{R}_{\delta_x \delta_y}^2(x_{\text{in}}^T, y_{\text{in}}^T) &\overset{(a)}{\leq} \Psi(z_{\text{in}}^T) \\
&\overset{(b)}{\leq} \left(\tfrac{2}{3}\right)^T \Psi(z_{\text{in}}^0) \\
&\overset{(c)}{=} \left(\tfrac{2}{3}\right)^T \left(\mathcal{R}_{\delta_x \delta_y}^2(x_{\text{in}}^0, y_{\text{in}}^0) + 12\mathrm{D}_f(x_{\text{in}}^0, x^*) + 12\mathrm{D}_g(y_{\text{in}}^0, y^*)\right) \\
&\overset{(d)}{\leq} \left(\tfrac{2}{3}\right)^T \left(\mathcal{R}_{\delta_x \delta_y}^2(x_{\text{in}}^0, y_{\text{in}}^0) + 12L_x\|x_{\text{in}}^0 - x^*\|^2 + 12L_y\|y_{\text{in}}^0 - y^*\|^2\right) \\
&\overset{(e)}{\leq} \left(\tfrac{2}{3}\right)^T (1 + 12\kappa_x + 12\kappa_y) \mathcal{R}_{\delta_x \delta_y}^2(x_{\text{in}}^0, y_{\text{in}}^0) \\
&\overset{(f)}{=} \left(\tfrac{2}{3}\right)^T cR^2,
\end{aligned}
$$

where (a) use the definition of $\Psi(z)$ in equation (29); (b) uses Theorem 4.7; (c) use the definition of $\Psi(z)$ in equation (29), where $(x^*, y^*) = \mathrm{proj}_{\mathcal{S}}(z_{\text{in}}^0)$; (d) uses the smoothness properties in Assumptions 2.3 and 2.4; (e) uses the definitions of $\kappa_x$ and $\kappa_y$ in equation (6) and the definition of $\mathcal{R}_{\delta_x \delta_y}^2$ in equation (9); (f) uses the definitions $z_{\text{in}}^0 = 0$, $R^2 = \mathcal{R}_{\delta_x \delta_y}^2(0, 0)$ and $c = 1 + 12\kappa_x + 12\kappa_y$. Next, we choose $T$ as follows:

$$
T = \left\lceil \frac{\log(cR^2/\epsilon)}{\log(3/2)} \right\rceil, \tag{184}
$$

which implies $\mathcal{R}_{\delta_x \delta_y}^2(x_{\text{in}}^T, y_{\text{in}}^T) \leq \epsilon$. Note that $T \geq 0$ due to the fact that $\epsilon \leq R^2$ and $c \geq 1$. In addition, we can upper-bound $T$ as follows:

$$
\begin{aligned}
T &\overset{(a)}{\leq} \frac{\log(cR^2/\epsilon)}{\log(3/2)} + 1 \\
&= \left(\frac{1}{\log(3/2)} + \frac{1}{\log(cR^2/\epsilon)}\right) \log(cR^2/\epsilon) \\
&\overset{(b)}{\leq} \left(\frac{1}{\log(3/2)} + \frac{1}{\log(c)}\right) \log(cR^2/\epsilon) \\
&\overset{(c)}{=} \left(\frac{1}{\log(3/2)} + \frac{1}{\log(1 + 12\kappa_x + 12\kappa_y)}\right) \log(cR^2/\epsilon) \\
&\overset{(d)}{\leq} \left(\frac{1}{\log(3/2)} + \frac{1}{\log(25)}\right) \log(cR^2/\epsilon) \\
&= \mathcal{O}\left(\log \frac{cR^2}{\epsilon}\right),
\end{aligned}
$$

where (a) uses the properties of $\lceil \cdot \rceil$; (b) uses the assumption $\epsilon \leq R^2$; (c) uses the definition $c = 1 + 12\kappa_x + 12\kappa_y$; (d) uses the fact that $\kappa_x, \kappa_y \geq 1$, which is implied by Assumption 2.7.

It remains to combine equation (28) in Theorem 4.7 and multiply $T$ by the appropriate number of computations of the gradients $\nabla p_i(z)$ and operators $Q_i(z)$, which are provided by Corollary 4.6. Note that the computation of the gradients $\nabla p_1(z)$ and $\nabla p_2(z)$ is equivalent to the computation of the gradients $\nabla f(x)$ and $\nabla g(y)$, respectively. The computation of the gradient $\nabla p_3(z)$ and operator $Q_3(z)$ requires $\mathcal{O}(1)$ matrix-vector multiplications with the matrices $\mathbf{B}$ and $\mathbf{B}^\top$, as well as a single computation of the gradients $\nabla f(x_{\text{in}})$ and $\nabla g(y_{\text{in}})$ at the beginning of the algorithm, which concludes the proof. $\qquad\square$

