# OpenReview forum: "On Linear Convergence in Smooth Convex-Concave Bilinearly-Coupled Saddle-Point Optimization: Lower Bounds and Optimal Algorithms"
_ICML.cc/2025/Conference — ICML 2025 poster_

### Official Review · Reviewer_FzFW · 2025-03-11

**Overall Recommendation:** 1

**Summary:**

This paper studies the first-order methods for solving smooth convex-concave saddle-point problems with bilinear coupling i.e. $ \min_x \max_y f(x) + \langle y, Bx \rangle - g(y)$. It establishes the first lower bounds on the number of gradient evaluations $\nabla f(x), \nabla g(y)$ and matrix-vector multiplications with $B$ and $B^{\top}$ needed for solving these saddle-point problems. Moreover, it develops algorithm which matches this lower bound.

**Claims And Evidence:**

This paper claims to develop an optimal algorithm for solving saddle point optimization problem with bilinear coupling. However, the paper doesn't provide any experiment to show it's performance and comparison to other existing algorithms in practice.

**Essential References Not Discussed:**

I think the references discussed in the paper are sufficient.

**Experimental Designs Or Analyses:**

This paper provides no experiments to evaluate the performance of the proposed algorithm. Therefore, there are no experimental designs or analyses to assess for soundness or validity.

**Methods And Evaluation Criteria:**

This paper provides no experiments to evaluate the performance of the proposed algorithm. The analysis is entirely theoretical, focusing on deriving lower complexity bounds and developing an optimal algorithm that matches these bounds. No empirical validation is provided to demonstrate its practical performance on real-world problems or benchmark datasets.

**Other Comments Or Suggestions:**

1. Please write down the pseudocode of the algorithm which solves the problem 1.
2. Add a discussion on how problem 14 generalizes problem 1.

**Other Strengths And Weaknesses:**

Strength:
1. Provides lower bound for saddle point optimization problems with bilinear coupling.

Weakness:
1. This paper claims they provide the first algorithm which attains the optimal complexity for strongly convex strongly concave. To my understanding, Kovalev et al 2022b already attains the optimal complexity with respect to each of the gradient evaluations (they just didn't mention that explicitly). I think the contribution of this paper is incremental.
2. The iteration complexity of this paper is given in terms of **weighted square distance** $\mathcal{R}^2$ which is completely different from Kovalev et al 2022b. **This is an unfair comparison.**
3. The paper provides no experiment to show the performance of proposed algorithm in practice.
4. The presentation is poor.
5. The main problem under consideration is problem (1). However, authors do not provide any pseudocode to solve this problem class. Only pseudocode is for algorithm 1, which solves problem 14.
6. It is hard to follow how problem 14 generalizes problem 1. Authors should add a detailed discussion on this in the Appendix.

**Questions For Authors:**

1. What is the number of gradient $\nabla f(x), \nabla g(y)$ evaluations required for the algorithm in Kovalev et al 2022b.
2. Assumptions 2.6 and 2.7 are not really assumptions right? They follow from assumptions 2.3-2.5.

**Relation To Broader Scientific Literature:**

The saddle-point optimization problem with bilinear coupling is a fundamental problem class that arises in various machine learning applications, including game theory, reinforcement learning, computer vision, robust optimization, and distributed optimization. Developing an optimal algorithm for solving such problems has the potential to significantly benefit the machine learning community.

**Theoretical Claims:**

The paper claims they develop the first optimal algorithm that matches the lower bound. However, they do not provide explicit pseudocode of the algorithm for solving the saddle point problems. Moreover, the iteration complexity of this paper is given in terms of **weighted square distance** $\mathcal{R}^2$ which is completely different from Kovalev et al 2022b. **This is an unfair comparison.**

---

> ### Author Rebuttal · Authors · 2025-03-31
>
> We thank the reviewer for the time and effort. Unfortunately, the development of the *optimal algorithm* for solving problem (1), which is one of the key contributions of our paper acknowledged by other reviewers, is missing from the "strengths" list. Moreover, the criticism of our paper is based on claims that are either **factually false** or *highly questionable*. We address these below.
>
> 1. ```Kovalev et al 2022b already attains...```
>     Unfortunately, this claim is false and unjustified:
>      - *This claim is false*. The algorithm of Kovalev et al. (2022b) cannot reach the lower bounds because it does not implement *the separation of complexities*. Refer to lines 60-72 in Section 1.2 for an explanation. This also clearly follows from Table 1.
>      - *This claim is unjustified*. You should support this claim with evidence, including an explanation with references to Kovalev et al. (2022b), such as precise theorems, equations, paragraphs, etc.
> 2. ```This is an unfair comparison.``` Unfortunately, *this statement is false.* The exact opposite is true: it is correct to compare $\mathcal{R}$ with any *norm*, including the one used by Kovalev et al. (2022b, eq. (14)), due to the *norm equivalence theorem*. The cost of the transition between the norms is negligible due to linear convergence: it only results in extra *additive factors* in the complexity. Moreover:
>      - It is standard practice in the field to ignore additive constants in linear complexities. This includes *all* SOTA papers in Table 1, most of which are A*/Q1.
>      - On lines 162 and 403, we explicitly mention that we ignore additive constants. This is also acknowledged by **Reviewer ydoA**.
> 3. ```No experiments.``` It is a common standard in the field that papers with strong theoretical results are not required to include any experiments, just as strong experimental papers are not required to include any theory. Our paper contains strong theoretical results, which are acknowledged by other reviewers. Hence, the absence of experiments is justified.
> 4. ```Presentation is poor.``` Unfortunately, you have not provided any arguments to support this claim. On the other hand, other reviewers found our paper "well-written" and "clearly written". Regrettably, our only option is to disregard this claim.
> 5. ```Authors do not provide pseudocode.``` In Section 4.3, we provide a clear and comprehensive explanation of how to apply Algorithm 1 with restarting to solve problem (1), including the definitions of functions $p_i$ and operators $Q_i$ in eqs. (25) and (26). This is more than enough for anyone with at least some expertise in mathematical optimization to use the algorithm. It is also important to highlight that, according to lines 351-367 in Section 4.2, it is necessary to reorder functions $p_i$ and operators $Q_i$, depending on the values of constants $L_i$ and $M_i$ from Theorem 4.7. This would lead to $3!=6$ explicit variants of the algorithm for each order. Hence, a practical implementation would still use Algorithm 1 in combination with separate first-order oracle implementations for functions $p_i$ and operators $Q_i$, which can be easily done in any modern programming language/framework.
> 6. ```How problem 14 generalizes problem 1.``` A brief explanation is available on lines 351-255, Section 4.3. Some explanation is also available in (Lan and Ouyang, 2021; Gidel et al., 2018; Nesterov, 2007). We are strongly convinced that a more detailed explanation is unnecessary because the reduction of convex-concave saddle-point optimization problems to monotone variational inequalities is one of the most basic facts in optimization theory. For a detailed explanation, please refer to books like "Finite-Dimensional Variational Inequalities and Complementarity Problems".
>
> ### Questions
>
> 1. ```What is the number...``` It is given in Table 1 of our paper or Table 1 of Kovalev et al. (2022b).
> 2. ```Assumptions 2.6...``` This is not true. Assumption 2.6 requires $\delta_x,\delta_y>0$, which is not implied by Assumptions 2.3-2.5. Moreover, Assumption 2.6 is the "line" that separates the settings of linear convergence and sublinear convergence. This is clearly mentioned numerous times in the paper, for instance, on lines 160-188 (Section 2.2), and lines 228-262 (Section 3.2). Similarly, it is easy to verify that Assumption 2.7 is not implied by Assumptions 2.3-2.5.

---

> > ### Comment · Reviewer_FzFW · 2025-04-02
> >
> > Thank you for your reply.
> >
> > 1. Kovalev et al 2022b haven't mentioned the gradient evaluations separately. But check the terms inside max of equation 25 from Kovalev et al, 2022b. Each of these terms corresponds to $\nabla f(x)$, $\nabla g(y)$ computations and exactly matches what you have.
> >
> > 2. The definition of R^2 contain terms like $\delta_x$ which in turn depend on $\mu_x, L_y, \mu_{xy}$. So showing $\|x_k - x_* \|^2 \leq \epsilon$ will have another $\frac{1}{\delta_x}$ term in the number of gradient evaluations.
> >
> > 3. No, the lack of experiments is not justified. The authors do not provide any pseudocode or discuss how to implement the algorithm in practice, which raises the question of whether the proposed algorithm benefits from the existing ones.
> >
> > 4. I did provide arguments on why I think the presentation is poor. a. There is pseudocode for the main problem (equation 1), b. There is no discussion on why problem (14) generalizes (1).
> >
> > I will keep my score.

---

> > > ### Author Response · Authors · 2025-04-03
> > >
> > > Thank you for the reply. Unfortunately, all four of your arguments are false:
> > >
> > > - >Kovalev et al 2022b haven't mentioned the gradient evaluations separately. But check the terms inside max of equation 25 from Kovalev et al, 2022b. Each of these terms corresponds to $\\nabla f(x)$, $\\nabla g(y)$ computations and exactly matches what you have.
> > >
> > >      Kovalev et al. (2022b) reported the **iteration complexity** $\\tilde{\\mathcal{O}}\\left(\\max\\left\\{\\sqrt{\\frac{L_x}{\\mu_x}},\\sqrt{\\frac{L_y}{\\mu_y}}, \\frac{L_{xy}}{\\sqrt{\\mu_x\\mu_y}}\\right\\}\\right)$. The gradients $\\nabla f(x)$ and $\\nabla g(y)$ are computed **exactly once at each iteration** of their algorithm. Hence, Kovalev et al. (2022b) require $\\tilde{\\mathcal{O}}\\left(\\max\\left\\{\\sqrt{\\frac{L_x}{\\mu_x}},\\sqrt{\\frac{L_y}{\\mu_y}}, \\frac{L_{xy}}{\\sqrt{\\mu_x\\mu_y}}\\right\\}\\right)$ computations of $\\nabla f(x)$ and $\\nabla g(y)$. This **does not coincide** with $\\tilde{\\mathcal{O}}\\left(\\sqrt{\\frac{L_x}{\\mu_x}}\\right)$ and $\\tilde{\\mathcal{O}}\\left(\\sqrt{\\frac{L_y}{\\mu_y}}\\right)$ from our paper.
> > > - >The definition of $R^2$ contain terms like $\\delta_x$ which in turn depend on $\\mu_x, L_y,\\mu_{xy}$. So showing $\\|x_k - x_*\\|^2 \\leq \\epsilon$ will have another $\\frac{1}{\\delta_x}$ term in the number of gradient evaluations.
> > >
> > >      Our algorithm requires $\\mathcal{O}(\\sqrt{\\kappa_x} \\log 1/\\epsilon)$ computations of the gradient $\\nabla f(x)$ to reach the precision $\\mathcal{R}^2_{\\delta_x\\delta_y}(x^k,y^k) \\leq \\epsilon$. Hence, to reach the precision $\\|x^k - x^*\\|^2 \\leq \\epsilon$, our algorithm requires $\\mathcal{O}(\\sqrt{\\kappa_x} \\log 1/(\\epsilon\\delta_x)) = \\mathcal{O}(\\sqrt{\\kappa_x} \\log 1/\\epsilon + \\sqrt{\\kappa_x} \\log 1/\\delta_x) = \\mathcal{O}(\\sqrt{\\kappa_x} \\log 1/\\epsilon)$ computations of the gradient $\\nabla f(x)$. As you can see, these linear convergence rates **coincide**.
> > >
> > > - >No, the lack of experiments is not justified. The authors do not provide any pseudocode or discuss how to implement the algorithm in practice, which raises the question of whether the proposed algorithm benefits from the existing ones.
> > >
> > >      We **do provide** a pseudocode for solving problem (1), which is a special instance of problem (14), in Algorithm 1, along with clear and comprehensive instructions on how to implement this algorithm in Section 4.3.
> > >
> > > - >I did provide arguments on why I think the presentation is poor. a. There is no pseudocode for the main problem (equation 1), b. There is no discussion on why problem (14) generalizes (1).
> > >
> > >      Unfortunately, neither argument supports the "poor presentation" claim:
> > >
> > >      - (a) This is false. Please refer to the information above and our original rebuttal.
> > >      - (b) There is no place for such a discussion in a scientific paper aimed at an audience with expertise in optimization, beyond the references that we provide in the paper, which are highlighted in our original rebuttal. In particular, the question of equivalence between problems (1) and (14) is so basic that it is literally asked during the undergraduate optimization course exam at most universities around the world, including ours.

---

### Official Review · Reviewer_PYLp · 2025-03-13

**Overall Recommendation:** 4

**Summary:**

This paper considers deterministic convex-concave minimax optimization problems. In particular, the main focus is on the case where we can obtain linear convergence, as characterized in Assumption 2.6.
* First, the authors establish fine-grained lower bounds by separately counting oracle calls for the gradients and the coupling matrix multiplications, and the results (1) recover [ZHZ'22] for the SCSC (strongly-convex-strongly-concave) case and (2) also cover some other cases (including strongly-convex-concave or bilinear functions).
* Second, the authors propose Algorithm 1, which is based on the idea of solving a more general finite-sum variational inequality. An application of this to minimax optimization problems yield tight convergence upper bounds for the cases described in the lower bound results.

[ZHZ'22] Zhang, J., Hong, M., and Zhang, S. On lower iteration complexity bounds for the convex concave saddle point problems. Mathematical Programming, 2022.

**Claims And Evidence:**

(The main results are theoretical.) I have not checked the proofs line by line, but the details seem to have no fatal errors.

**Essential References Not Discussed:**

I am unaware of any particular closely related work that has not been cited in the paper.

**Ethical Review Concerns:**

(None)

**Experimental Designs Or Analyses:**

The main results are theoretical.

**Methods And Evaluation Criteria:**

The main results are theoretical.

**Other Comments Or Suggestions:**

* The running title says 'Submission and Formatting Instructions for ICML 2025.' (I have 3 such papers in my review batch, and I don't know why!)
* I think it would have been better if there were more discussions on the lower bound instance constructions, which I think is one of the most interesting parts of the paper.
* (TYPO) Table 1, Footnote (5) symetric → symmetric

**Other Strengths And Weaknesses:**

**Strengths**
* The paper is well-written (in my opinion), and I really enjoyed reading it.
* The paper suggests both novel lower bounds and matching upper bounds that, altogether, closes the case. The proposed results make solid contributions, especially those for the new tight rates for all of the non-SCSC cases.
* The paper also contains (upper bound) results that consider finite-sum variational inequalities of the form $(14)$.

**Weaknesses**
* See **Questions** for details.

**Questions For Authors:**

* Can the authors elaborate on any iteration complexities or number of oracle calls induced in the $\arg \min$ part of Algorithm 1 in the innermost loop $k = n+1$?
* In the lower bounds, using matrices with $1$'s in the diagonals and $-1$'s in the super-diagonals is quite standard (as in [ZHZ'22]), but I wonder what the high-level motivations of the instances in Lemmas G.2 and G.3 (for the $\sqrt{\kappa_{xy}}$ parts) were. Could the authors give a bit of an illustration on this, or is this just a magical lower bound?
* I am also curious if previous results by [ZHZ'22] can recover the same results considering separate counts for the gradient and coupling matrix oracles for the SCSC case. (I am aware that [ZHZ'22] deals *only* with the SCSC case while this paper covers several more cases.)
* This might be a bit out of scope because the proposed problem $(1)$ is already an important, classical problem, but one weakness one could point out (for the upper bound) is that everything applies only when we have the bilinear-coupled structure. Personally, I also have been curious about whether we can construct an intuitive minimax optimization algorithm for SCSC (or maybe more general cases where we can have linear convergence) *without* the bilinear coupling structure in the objective function. I know that the existence of the coupling matrix is essential for the proposed results, but have the authors ever thought of extending Algorithm 1 to the general convex-concave case by, for instance, replacing the $\boldsymbol{B}$'s with the Hessians $\nabla_{xy} f$, or else?

**Relation To Broader Scientific Literature:**

The paper can contribute to theoretical guarantees and understandings on convex-concave minimax optimization algorithms and variational inequalities.

**Theoretical Claims:**

I have not checked the proofs line by line, but I read the appendix, and the details seem to have no fatal errors.
(See **Strengths & Weaknesses** and **Questions** for details.)

---

> ### Author Rebuttal · Authors · 2025-03-31
>
> We thank the reviewer for the detailed comments, valuable feedback, and high evaluation of our work. Below, we provide our detailed response to the review.
>
> ### Other Comments Or Suggestions
> - Fixed.
> - Please refer to the separate paragraph below.
> - Fixed.
>
> ### Questions For Authors
> - All functions $p_i^{n;t_1,\\ldots,t_n}(z)$ are just quadratic functions that contain previously computed gradients $\\nabla p_i(z)$ and operators $Q_i(z)$. This can be shown by analyzing lines 17 and 22 at recursion levels $k = 1,\\ldots,n$. Hence, line 12 is a simple (possibly constrained) quadratic optimization problem, which does not require any oracle calls.
> - Please refer to the separate paragraph below.
> - Indeed, the result of Zhang et al. (2022b) recovers the $\\sqrt{\\kappa_{xy}}$ part in the SCSC case. In particular, the proof of Lemma G.3 is inspired by the result of Ibrahim et al. (2020), which is a slightly generalized version of the result by Zhang et al. (2022b). Combining this with the lower bound for smooth and strongly convex optimization by Nesterov (2013), we recover the full result in the SCSC case.
> - Some results for general min-max problems beyond SCSC were developed by Kovalev et al. (2022b). However, these results are far from reaching our lower bounds. It is worth highlighting that obtaining accelerated rates is much more difficult for general min-max problems (some results were developed by Alkousa et al. (2020), [2] mentioned by **Reviewer 1jnp**, or in arXiv:2002.02417, arXiv:2205.05653). Overall, this is indeed an interesting question that we are currently starting to examine.
>
> ### Lower bounds construction.
> Here, we attempt to provide an intuition behind the construction of our lower bounds. We start with the basic example of Nesterov (considering an infinite-dimensional space $\\ell^2$ for simplicity). Consider the following linear system:
> $$\\mathbf{A}x = e_1,$$
> where
> $$\\mathbf{A} = \\begin{bmatrix}1\\\\\\alpha-1&1\\\\&\\alpha-1&1\\\\&&&\\ddots\\end{bmatrix},$$
> and $\\alpha \\in (0,1)$. It has a unique solution $x^* = (1,1-\\alpha,(1-\\alpha)^2,\\ldots)$. We can construct a minimization problem with the same solution:
> $$\\min_x \\|\\mathbf{A}x - e_1\\|^2.$$
> One can show that after $k$ computations of the gradient, no more than $\\mathcal{O}(k)$ coordinates of the current iterate $x^k$ are nonzero (due to the linear span assumption). Hence, the distance to the solution is lower bounded by the remainder of the geometric series $\\sum_{i=k+1}^\\infty(1-\\alpha)^i = \\frac{(1-\\alpha)^{k+1}}{\\alpha}$. Moreover, the condition number $\\kappa_x$ is proportional to $1/\\alpha^2$, which gives the lower bound $\\tilde{\\Omega}(\\sqrt{\\kappa_x})$ of Nesterov.
>
> The minimization problem above has the following min-max reformulation:
> $$\\min_x\\max_y 2 \\langle y, \\mathbf{A}x - e_1\\rangle - \\|y\\|^2.$$
> Here, we have $\\kappa_{xy}$ proportional to $1/\\alpha^2$, and we can apply arguments similar to the ones above to obtain the lower bound $\\tilde{\\Omega}(\\sqrt{\\kappa_{xy}})$. We can also add a regularizer of the form $\\|x\\|^2$, which, subject to some additional details, leads to the results of Zhang et al. (2022b) and Ibrahim et al. (2020). Our Lemma G.3 can be seen as a finite-dimensional variant of the result by Ibrahim et al. (2020).
>
> Now, we discuss the most challenging case of our lower bounds, which is Lemma G.2. The starting point is the lower bounds of Scaman et al. (2018) for smooth and strongly convex decentralized minimization, which allows for the min-max reformulation of the form (1). It is based on splitting the hard function of Nesterov (see above) into two functions and placing them on the first and the last $n/3$ nodes of the path consisting of $n$ nodes (refer to Scaman et al. (2018) for details). To obtain our lower bounds, we need to make the following substantial changes:
> - We add dual regularization of the form $-\\mu_y \\|y\\|^2$. This allows us to obtain the desired result for $\\mu_y > 0$, but it makes the problem much more difficult to work with.
> - We replace the $n$-node path with an $n/3$-node path and attach two $n/3$-node star-topology networks to its ends, such that Nesterov functions are stored only on the "leaves" of these stars. This step introduces some sort of symmetry, which simplifies finding the solution to the problem.
> - We also introduce an extra dual variable, modify matrix $\\mathbf{B}$, and add an extra regularizer of the form $-L_y\\|y\\|^2$ with respect to the new variable to account for nontrivial values of $L_y \\gg \\mu_y$. Additionally, we analyze the spectral properties of matrix $\\mathbf{B}$ in Lemma I.1.
>
> We apply a series of reformulations to the resulting min-max problem and obtain a simple minimization problem similar to the example above but with a different value of $\\alpha$. It remains to carefully analyze this value, subject to additional details which we, unfortunately, are unable to discuss here due to the 5000 character limitation.

---

### Official Review · Reviewer_ydoA · 2025-03-13

**Overall Recommendation:** 4

**Summary:**

This paper develops tight lower complexity bounds and matching optimal algorithms for smooth saddle-point problems with bilinear coupling. The work unifies existing results in different regimes (strongly-convex-strongly-concave, bilinear saddle-point, strongly convex with affine constraints) as well as gives new results in the convex-concave setting.

---
 ## Update after rebuttal

The authors have addressed my concerns regarding the proof of Theorem 3.3 in their rebuttal. I keep my current evaluation of the paper.

**Claims And Evidence:**

The claims are clear and supported by convincing evidence.

**Essential References Not Discussed:**

The related publications are adequately discussed throughout the paper.

**Experimental Designs Or Analyses:**

There are no experiments in this paper.

**Methods And Evaluation Criteria:**

The intuition behind the used methodology is lacking and could be made more clear. The evaluation criteria make sense for the problem at hand.

**Other Comments Or Suggestions:**

List of typos and minor errors:
1. The page titles are still “Submission and Formatting Instructions for ICML 2025”
2. Line 140: would it be possible to write $0$ instead of the minimum eigenvalues in the “otherwise” cases, or is this not valid?
3. Line 165: Assumption 2.6 is a bit poorly worded, it might give the impression that Assumptions 2.3-2.5 implies the inequality.
4. Line 200: Equation (9): missing squares for the distances
5. Line 233: fix first set of quotation marks around “hard”
6. Line 245: Theorem 3.2: does not hold for any time $\tau>0$, in Nesterov’s book there is an upper bound for the iteration number which should translate into an upper bound for $\tau$
7. Line 327: Footnote 5: it is better to specify the proposition/page number instead of the whole book
8. Line 345: Theorem 4.5: should also include a reference to Algorithm 1 as well as the problem to be solved
9. Line 402: “numbers” should be “number”
10. Line 448: capitalisation Polyak-Łojasiewicz (also add en-dash names between instead of hyphen)
11. Capitalisation of conference names in references is not consistent (e.g. “International Conference on Machine Learning” vs “international conference on machine learning” or “Advances in Neural Information Processing Systems” vs “Advances in neural information processing systems”
12. Line 561: “eigenvalues of a matrix” should be “eigenvalues of a symmetric matrix” since otherwise they might not be real
13. Line 566: “argmin” should be “min”
14. Line 699: “symetric” should be “symmetric”
15. Line 718: “this” should be “these”
16. Line 738: Equation (38) does not follow from putting in the values from (37)
17. Line 782: A reference to Assumption 2.6 in addition to Assumption 2.5 should be added. Moreover, the implication stated should be explained more thoroughly
18. Each time a line number from Algorithm 1 is referenced, it comes out as “algorithm 1 of Algorithm 1” instead of “line x of Algorithm 1”
19. Line 1430: “wuch” should be “such”
20. Line 1527: $(n+i)\times(n+i)$ should be in the superscript
21. Line 2290: Missing an identity matrix in the upper bound
22. Line 2430: Lemma K.2: does this hold for $k \neq i$?
23. Line 2460: Equation (174): missing superscript $0$ on $r_i(\hat{z})$; the last $z$ should be $\hat{z}$; $z^0$ should be $z_{\rm in}$
24. Line 2489: “which is implied by which is implied by”
25. Line 2495: “Introduction step” should be “Induction step”
26. Line 2495: also have to assume that it holds for $k=n$ or the induction is invalid
27. Line 2518: $\nabla \hat{p}_i^k$ should be $\nabla \hat{p}_k^k$
28. Line 2596: $\hat{p}_i^k$ should be $\hat{p}_k^k$
29. Line 2835, 2846, 2850: subscript inside the last product should be $j$ instead of $i$ (resp. $j+1$ instead of $i+1$)
30. Line 2880: step (c) should be an inequality and $\|z-z’\|_P^2$ should be $\|z-z’\|_P$
31. Line 2911: Q(z) should be Q(z^*)
32. Line 2930: The Lipschitz constants are missing
33. Line 2983: the coefficients 12 could be 6?
34. Line 2985: step (e) equality should be an inequality
35. Line 3001-3009 could be removed, the statement is obvious since big O does not care about additive constants
36. Line 3013: deifnition should be definition

**Other Strengths And Weaknesses:**

Strengths:
- The paper is clearly written and provides tight theoretical results in the linearly-converging setting.
- The results of the paper are relevant to numerous machine learning applications.

Weaknesses:
- There is no intuition provided for some constants such as the ones in Assumption 2.7 or the Lyapunov function.
- Similarly, the proofs are not very enlightening, being only a series of algebraic manipulations.

**Questions For Authors:**

Could you elaborate on the proof of Theorem 3.3 in the cases that $\mu_x=\mu_y=0$ or when one of the two strong convexity constants is $0$?

**Relation To Broader Scientific Literature:**

The paper studies saddle-point problems which have applications in various fields such as economics, game theory and statistics. A comparison with existing state-of-the-art linearly-converging algorithms is given in Appendix C. Moreover, optimal algorithms and theoretical lower bounds are given, which builds further upon the work of Nesterov.

**Theoretical Claims:**

I checked the claims and proofs in all the Appendices except Appendix I. The main issue I have is in the proof of Theorem 3.3 (Appendix G), where the case $\mu_x=\mu_y=0$ is proved by assuming that $\mu_y>0$. The authors should provide more explanation why this is justified.

---

> ### Author Rebuttal · Authors · 2025-03-31
>
> We thank the reviewer for the detailed comments, valueable feedback, and high evaluation of our work. Below, we provide our detailed response to the review.
>
> ### Question about the proof of Theorem 3.3 in the case $\\mu_x = 0$ or $\\mu_y = 0$
>
> Thank you for the question! This indeed may need additional explanation. This question is related to cases **(i)**, **(ii.b)**, and **(iii.b)**. For instance, consider case **(i)** (other cases are similar). It is easy to observe that if the function $g(y)$ is $\\mu_y$-strongly convex with $\\mu_y > 0$, then it is also $0$-strongly convex or simply convex (see Definition 2.1 and Assumption 2.4). Hence, the class of problems (1) satisfying Assumptions 2.3-2.5 with parameters $\\pi = (\\ldots,\\mu_y,\\ldots) \\in \\Pi$ is contained in the class of problems (1) satisfying Assumptions 2.3-2.5 with parameters $\\pi = (\\ldots,0,\\ldots) \\in \\Pi$. Thus, for case **(i)**, we can choose our hard instance of problem (1) with a $\\mu_y$-strongly convex function $g(y)$ with an arbitrary $0 < \\mu_y < L_y/4$. In particular, we choose the hard problem instance according to Lemma G.2, which gives the following lower bound:
> $$
> \\tilde{\\Omega}\\left(\\frac{L_{xy}\\sqrt{L_y}}{\\mu_{xy}\\sqrt{\\mu_y + \\mu_{yx}^2/L_x}}\\right).
> $$
> We can choose $\\mu_y =  \\mu_{yx}^2/L_x$ and obtain
> $$
> \\tilde{\\Omega}\\left(\\frac{L_{xy}\\sqrt{L_xL_y}}{\\mu_{xy}^2}\\right),
> $$
> which is the desired lower bound for case **(i)**.
>
> ### Intuition behind the numerical constants
>
> - There is no particular intuition behind the actual values of the numerical constants in Assumption 2.7 (4 and 18), except that we chose these constants to simplify our calculations in the proof of Theorem 3.3 and make them less ugly. It is likely that these numerical constants can be reduced, but it would not make much sense because Assumption 2.7 is only used to avoid covering uninteresting corner cases with small, i.e., $\\mathcal{O}(1)$, condition numbers, as mentioned in the paper.
> - The numerical constants in the Lyapunov function in eq. (29) hold little intuition. The values of these constants are mostly driven by the proof and can likely be improved as well, but it would not make much sense as this would only result in logarithmic or additive improvements in complexity.
>
> ### Intuition behind the proofs
>
> For the intuition behind the construction and the proof of lower bounds, please refer to our response to **Reviewer PYLp**, who raised a similar question. Unfortunately, we were unable to provide a more detailed intuition behind the convergence proof beyond what we have in Section 4.2 (lines 303-345) due to the 5000 character limit (we really tried). We will include a more detailed explanation in the revised version of the paper.
>
> ### Typos and minor errors
>
> Thank you for the list of typos and minor errors! We fix them all as follows:
> - (1) Fixed.
> - (2) Yes, indeed. For instance, in the definition of $\\mu_{xy}$, if $\\lambda_{\\min}(\\mathbf{B}^\\top \\mathbf{B}) > 0$, we have $\\mathrm{range} \\mathbf{B}^\\top = \\mathcal{X}$ and fall into the first option.
> - (3) Thanks for the suggestion. We have removed the references to Assumptions 2.3-2.5 and added the words "Parameters $\\pi \\in \\Pi$ satisfy the inequality...".
> - (4-5) Fixed.
> - (6) Indeed, speaking rigorously, we need to mention the transition from the lower bound on the number of iterations in Nesterov's book to the lower bound on $\\tau$, even though it is straightforward. We will add an appropriate comment to the proof in the revised version of the paper.
> - (7) Added reference to Lan (2020, proof of Theorem 3.3).
> - (8-15) Fixed.
> - (16) Indeed, lines 732-740 are worded inaccurately because we cannot choose $L_x=L_y=0$ due to Assumption 2.7. This is also discussed on lines 741-745 ("Note that strictly speaking..."). We will rewrite line 732 and eq. (37) in a more accurate way, i.e., something like "the class of bilinear saddle-point optimization problems falls under Assumptions 2.3-2.5 with parameters..."
> - (17) Indeed, the case $\\mu_x  > 0$ is trivial. The case $\\mu_x = 0$ implies $\\mu_{xy} > 0$ due to Assumption 2.6, and $\\nabla f(x) \\in \\mathrm{range}\\mathbf{B}^\\top = (\\ker \\mathbf{B})^\\perp$ due to Assumption 2.5 and point #2 from your list.
> - (18-21) Fixed.
> - (22) Yes, it does. For $i = k$, it holds due to line 22 (first option). For $k > i$, we have $p\_{i}^{k,t_1,\\ldots,t\_k} \\equiv \\hat{p}\_{i}^{k,t\_1,\\ldots,t\_k} \\equiv p\_{i}^{k-1,t\_1,\\ldots,t\_{k-1}}$ due to line 22 (second option) and line 17 (second option). Hence, we can prove the desired statement by induction.
> - (23-32) Fixed.
> - (33) Yes, indeed, they could.
> - (34) Fixed.
> - (35) Yes, indeed.
> - (36) Fixed.

---

> > ### Comment · Reviewer_ydoA · 2025-04-04
> >
> > I would like to thank the authors for their reply. The authors have addressed my concerns regarding the proof of Theorem 3.3.
> >
> > I keep my current evaluation of the paper.

---

### Official Review · Reviewer_1jnp · 2025-03-25

**Overall Recommendation:** 4

**Summary:**

This work studied smooth (strongly)-convex-(strongly)-concave bilinearly-coupled saddle-point problem, and provided lower complexity bounds in terms of the computation time, and achieved the separation of complexities. And they further proposed an optimal algorithm which matches with the lower bound.

**Claims And Evidence:**

The results are supported by convincing evidence, generally I am satisfied with the results. Here are some questions:

1. Missing literature. Line 83, you mentioned for strongly-convex-concave case, "To the best of our knowledge, there are no lower complexity bounds that would cover these cases", I think the work [1] should have solved it, check Theorem 2 therein.
2. Another missing literature should be [2], whose upper bound result is different from your Table 1.
3. Echo on [1], they also extended the oracle class to proximal mapping, which is broader than your setting, which may be an extension that you can consider.

[1] Xie, Guangzeng, et al. "Lower complexity bounds for finite-sum convex-concave minimax optimization problems." ICML 2020.

[2] Wang, Yuanhao, and Jian Li. "Improved algorithms for convex-concave minimax optimization." NeurIPS 2020

**Essential References Not Discussed:**

See above

**Experimental Designs Or Analyses:**

/

**Methods And Evaluation Criteria:**

/

**Other Comments Or Suggestions:**

/

**Other Strengths And Weaknesses:**

/

**Questions For Authors:**

/

**Relation To Broader Scientific Literature:**

This work advanced the understanding of optimal complexities for bilinear min-max optimization problems.

**Theoretical Claims:**

/

---

> ### Author Rebuttal · Authors · 2025-03-31
>
> We thank the reviewer for the high evaluation of our work and the useful references. We provide our answers to the questions below.
>
> 1. Thank you for pointing this out. The statement "to the best of our knowledge, there are no lower complexity bounds that would cover these cases" is indeed a bit inaccurate. Our point was that there are no *linear* lower bounds, i.e., lower bounds for the linear convergence setting. On the other hand, [1, Theorem 2] provides a *sublinear* lower bound. We will fix the inaccuracy and cite [1] in the revised version of the paper.
> 2. As far as we understand, the main limitation of [2] is that it can achieve the linear SOTA rate $\tilde{\mathcal{O}}\left(\sqrt{\frac{L_x}{\mu_x}+\frac{L_y}{\mu_y} + \frac{L_{xy}}{\mu_x\mu_y}}\right)$ in the SCSC regime only for quadratic problems. On the other hand, [2] achieves SOTA rates for general min-max problems without bilinear coupling. We will discuss this in the revised version of the paper.
> 3. Thank you for the suggestion. This is an interesting question to consider for future work. In particular, we think that it is possible to remove the terms $\sqrt{\kappa_x}$ and/or $\sqrt{\kappa_y}$ from the lower and upper complexity bounds by assuming access to proximal mappings associated with functions $f(x)$ and $g(y)$, respectively.

---

> > ### Comment · Reviewer_1jnp · 2025-04-07
> >
> > Thank you for the clarification.
> >
> > A further question, can you clarify whether the dist() in your Equation 9 comes with square? I did not find the detailed formal definition. BTW, Equation 32 should be wrong I think, it should be min, rather than argmin (also questionable on whether to apply square). From the proof in Appendix I, the dist() should be defined in terms of squared norm I think.

---

> > > ### Author Response · Authors · 2025-04-08
> > >
> > > Thank you for your reply.
> > >
> > > There is a typo in equation (9); the squared distance $\\mathcal{R}\_{\\delta_x\\delta_y}^2$ should be defined using squared distances $\\mathrm{dist}^2$ as follows:
> > > $$
> > > \\mathcal{R}\_{\\delta_x\\delta_y}^2(x,y) = \\delta\_x \\mathrm{dist}^2 (x;\\mathcal{S}_x) + \\delta_y \\mathrm{dist}^2 (y;\\mathcal{S}_y).
> > > $$
> > > There is indeed also a typo in equation (32); $\\arg\\min$ should be replaced with $\\min$, but no square this time:
> > > $$
> > > \\mathrm{dist}(x;\\mathcal{A}) = \\min\_{x' \\in \\mathcal{A}} \\|x-x'\\|.
> > > $$
> > > The proof in Appendix I indeed uses the squared distances $\\mathcal{R}\_{\\delta_x\\delta_y}^2$, according to the corrected definitions above. Thank you for pointing out the typos; we have fixed them.

---

### Decision · Program_Chairs · 2025-05-01

**Decision:**

Accept (poster)

**Comment:**

This paper studies bilinearly coupled saddle-point problems with general smooth (non-strongly) convex $f(x)$ and $g(y)$. The main contributions of the work include tight upper and lower complexity bounds that establish linear convergence rates. The paper is well written and provides insightful discussion into the nuances that arise in assessing optimal convergence complexity for saddle-point settings. The reviewers were overall in agreement that the paper provides important results for saddle-point optimization, and would therefore be a valuable contribution to the conference.